# MME-UNIFY: A COMPREHENSIVE BENCHMARK FOR UNIFIED MULTIMODAL UNDERSTANDING AND GENERATION MODELS

**Wulin Xie**[1,*], **Yi-Fan Zhang**[1,5,*,†], **Chaoyou Fu**[2,5], **Yang Shi**[3], **Bingyan Nie**[1], **Hongkai Chen**[4], **Zhang Zhang**[1], **Liang Wang**[1]

[1]CASIA, [2]NJU, [3]PKU, [4]Vivo, [5]M-M-E
[*]Equal Contribution  [†]Project leader
https://mme-unify.github.io/

## ABSTRACT

Unified Multimodal Large Language Models (U-MLLMs) have garnered considerable interest for their ability to seamlessly integrate generation and comprehension tasks. However, existing research lacks a unified evaluation standard, often relying on isolated benchmarks to assess these capabilities. Moreover, current work highlights the potential of "mixed-modality generation capabilities" through case studies—such as generating auxiliary lines in images to solve geometric problems, or reasoning through a problem before generating a corresponding image. Despite this, there is no standardized benchmark to assess models on such unified tasks. To address this gap, we introduce MME-Unify, also termed as MME-U, the first open and reproducible benchmark designed to evaluate multimodal comprehension, generation, and mixed-modality generation capabilities. For comprehension and generation tasks, we curate a diverse set of tasks from 12 datasets, aligning their formats and metrics to develop a standardized evaluation framework. For unified tasks, we design five subtasks to rigorously assess how models' understanding and generation capabilities can mutually enhance each other. Evaluation of 17 U-MLLMs, including Janus-Pro, Bagel, and Gemini2-Flash, reveals significant room for improvement, particularly in areas such as instruction following and image generation quality.

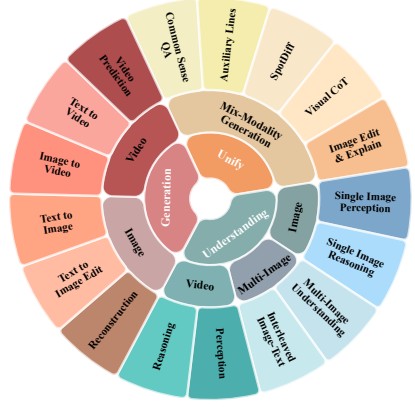

| Rank | MME-U Score | |
|:---:|:---:|:---:|
| ① | Gemini2.5-flash | 50.04 |
| ② | Gemini2.0-flash | 45.57 |
| ③ | RecA | 42.60 |
| 4 | GPT-4o-Image | 41.06 |
| 5 | Bagel | 40.35 |
| 6 | MIO-Instruct | 37.17 |
| 7 | SEED-LLaMA | 28.45 |
| 8 | Anole | 18.59 |

(a) MME-U tasks.        (b) Leaderboard.

Figure 1: **A comprehensive visualization of the diverse tasks in MME-U and the leaderboard.** The figure (a) illustrates the wide-ranging nature of the tasks covered in our benchmark, which spans from traditional understanding tasks to complex mixed-modality generation challenges. The leaderboard (b) highlights the performance rankings of various U-MLLMs in our benchmark.

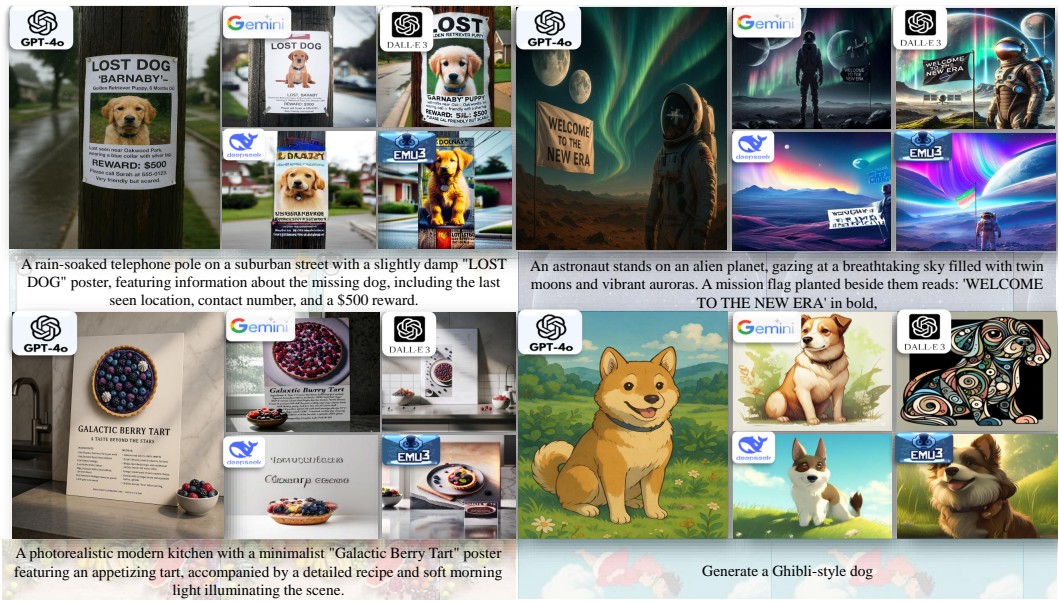

Figure 2: **Complex instruction-based image generation comparison** of results from open-source U-MLLMs (DeepSeek-Janus Flow, EMU3), closed-source U-MLLMs (GPT-4o, Gemini-2), and proprietary models (DALLE-3). The closed-source U-MLLMs have demonstrated abilities surpassing proprietary generation models, with a significantly larger gap compared to open-source models.

## 1 INTRODUCTION

Unlike traditional MLLMs (e.g., GPT-4V) and purely generative models (e.g., DALL-E 3), U-MLLMs Xie et al. (2025b); Wang et al. (2024b); Chen et al. (2025c); Ma et al. (2024) excel in processing mixed-modal inputs and outputs, providing enhanced flexibility and the ability to address a broader spectrum of complex tasks. Recently, closed-source U-MLLMs, such as Gemini 2.0 Flash, have demonstrated exceptional generative capabilities, impressing in both instruction comprehension and image creation, as shown in Figure 2. These models exhibit an extraordinary grasp of image details, even surpassing proprietary generative models. However, this versatility also introduces considerable challenges in comprehensively evaluating their capabilities under unified evaluation criteria, primarily due to two key issues:

- **Lack of Standardized Benchmarks for Traditional Tasks.** Existing works typically evaluate traditional generation and understanding tasks separately, using various benchmarks. However, the benchmarks chosen across studies are inconsistent, and often incompatible, leading to unfair comparisons. Moreover, the evaluation methods differ significantly—multimodal understanding tasks may involve varied formats such as multiple-choice questions, GPT-4 scoring, or binary classification, while multimodal generation tasks may rely on metrics like CLIP score or FID. This diversity in evaluation makes it difficult to derive an intuitive and unified performance score.

- **Absence of Benchmarks for Mixed-Modality Generation[1].** The most distinctive feature of U-MLLMs is their mixed-modality generation capabilities, which demonstrate the synergistic interaction between multiple modalities Shi et al. (2025); Yang et al. (2025b). For instance, image editing requires accurately understanding textual instructions and identifying objects to be modified, while solving geometry problems involves comprehending the problem, drawing auxiliary lines, and performing logical reasoning. Despite these advanced capabilities, most methods only showcase simple cases, lacking a standardized benchmark to rigorously assess these complex mixed-modality tasks.

To address these challenges, we propose a comprehensive evaluation framework for U-MLLMs, which is shown in Figure 1. For **traditional generation and understanding tasks**, we sample data

---

[1]also termed as unify tasks

Table 1: **Comparison of MME-U and other Benchmark. SIPU:** Single Image Perception & Understanding; **MITIU:** Multiple & Interleaved Image-Text Understanding; **VPU:** Video Perception & Understanding; **CIVG:** Conditional Image-to-Video Generation; **FIR:** Fine-grained Image Reconstruction; **TIE:** Text-Guided Image Editing; **TIG:** Text-to-Image Generation; **TVG:** Text-to-Video Generation; **VP:** Video Prediction; **UT:** Unified Task.

| Benchmark | Question | Year | Understanding | | | Generation | | | | | | Unify |
|---|---|---|---|---|---|---|---|---|---|---|---|---|
| | | | SIPU | MITIU | VPU | FIR | TIE | TIG | CIVG | TVG | VP | UT |
| MSR-VTT Xu et al. (2016) | 10,000 | CVPR 2016 | × | × | ✓ | × | × | × | × | × | × | × |
| MMBench Liu et al. (2024) | 3,217 | arXiv 2023 | ✓ | × | × | × | × | × | × | × | × | × |
| GenEval Ghosh et al. (2023) | 1,200 | arXiv 2023 | × | × | × | × | × | ✓ | × | × | × | × |
| MagicBrush Zhang et al. (2023) | 10,338 | NeurIPS 2023 | × | × | × | ✓ | ✓ | × | × | × | × | × |
| VBench Huang et al. (2024) | 1,600 | CVPR 2024 | × | × | × | × | × | × | × | ✓ | × | × |
| SEED-Bench2 Li et al. (2023) | 19,242 | arXiv 2024 | ✓ | ✓ | ✓ | ✓ | ✓ | ✓ | × | × | × | × |
| Emu-Edit Sheynin et al. (2024) | 5,611 | CVPR 2024 | × | × | × | ✓ | ✓ | × | × | × | × | × |
| TIP-I2V Wang & Yang (2024) | 500,000 | arXiv 2024 | × | × | × | × | × | × | ✓ | × | ✓ | × |
| MMBench-Video Fang et al. (2025) | 2,000 | NeurIPS 2024 | × | × | ✓ | × | × | × | × | × | × | × |
| MME Fu et al. (2023) | 2,374 | arXiv 2023 | ✓ | × | × | × | × | × | × | × | × | × |
| Video-MME Fu et al. (2024a) | 2,700 | CVPR 2025 | × | × | ✓ | × | × | × | × | × | × | × |
| MME-RealWorld Zhang et al. (2025d) | 29,429 | ICLR 2025 | ✓ | × | × | × | × | × | × | × | × | × |
| Wise Niu et al. (2025) | 1,000 | arXiv 2025 | × | × | × | × | × | × | × | × | × | ✓ |
| **MME-Unify (ours)** | **4,100** | **2025** | ✓ | ✓ | ✓ | ✓ | ✓ | ✓ | ✓ | ✓ | ✓ | ✓ |

from 12 existing datasets, resulting in 10 tasks with 30 subtasks. On the understanding side, these tasks encompass single-image, multi-image, and video-based perception and reasoning tasks, covering a wide range of difficulties—from simple visual question-answering (VQA) to high-resolution VQA in real-world scenarios and long-video understanding. On the generation side, we include tasks such as image/video generation and editing, as well as more complex conditional image generation and image-to-video generation, aiming to cover the full spectrum of existing generative tasks. To simplify evaluation and provide a unified score, we manually reformat all understanding tasks into multiple-choice questions, reporting accuracy as the primary metric. For generation tasks, we standardize the evaluation scores and normalize them to provide a consistent metric. This approach reduces the difficulty of benchmark collection and mitigates the issue of inconsistent evaluation metrics across studies.

**For the Unified Tasks**, we constructed five tasks: (1) Image Editing and Explaining, where the model first understands complex editing instructions and edits an image; (2) Common Sense Question Answering, where the model answers a question and generates the corresponding image; (3) Auxiliary Lines, where the model draws auxiliary lines for geometry problems and then solves them; (4) SpotDiff, where the model identifies and draws the differences between two images; and (5) Visual CoT, where the model generates step-by-step strategies for navigating a maze and visualizes the next state. These tasks evaluate a model's ability to perform sequential reasoning and generate corresponding multimodal outputs at each step. All tasks are carefully formatted as multiple-choice questions to facilitate consistent, fair, and objective evaluation.

We evaluate 17 existing U-MLLMs, including Janus-Pro, Bagel, VILA-U, and MiniGPT-5. To provide context for their performance, we also compare them with specialized understanding models (e.g., Claude-3.5 Sonnet, Qwen2.5-VL) and generative models (e.g., DALL-E-2, DALL-E-3). This comprehensive evaluation not only underscores the strengths and weaknesses of U-MLLMs but also establishes a standardized benchmark for future research in this rapidly evolving field. For example, we uncover several key experimental findings, as illustrated in Figure 1. Currently, U-MLLMs exhibit significant variance in rankings across three dimensions, and no single model has emerged as the best performer across multiple capabilities. Moreover, the performance gap between models is substantial. Finally, the current open-sourced U-MLLMs still exhibit a significant gap in performance compared to specialized models in both understanding and generation tasks. Additionally, while many works claim to handle mixed-modality generation, our unify task tests demonstrate that most of existing U-MLLMs struggle to consistently and effectively process these types of tasks.

## 2 MME-UNIFY

This section outlines the data collection, question annotation, and evaluation strategy for MME-Unify. Figures 1 and 3 provide visual representations of subtasks and samples across three domains, while Table 1 compares MME-U with existing benchmarks. MME-U categorizes U-MLLM capa-

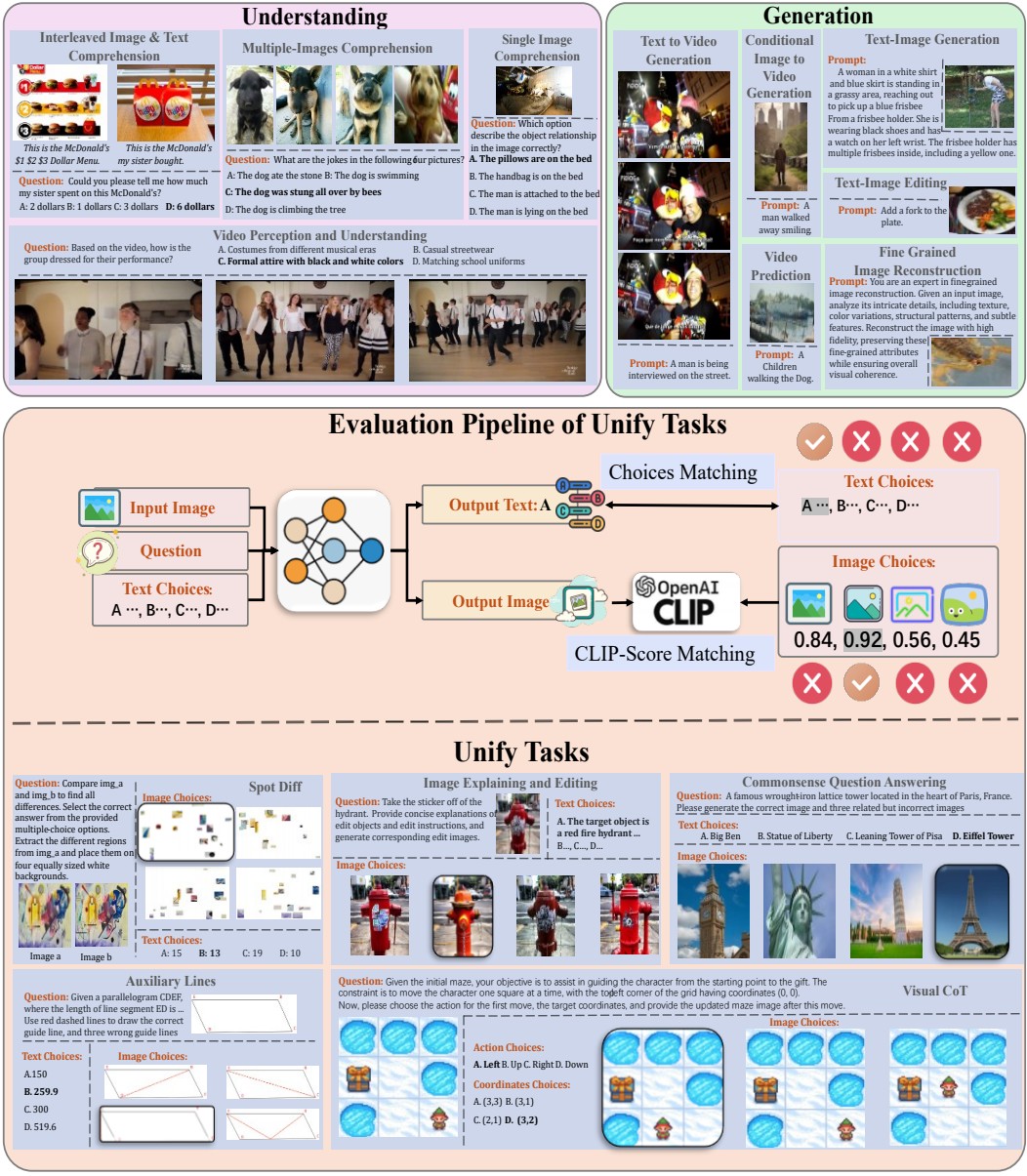

Figure 3: **Diagram of our MME-Unify.** Our benchmark consists of 3 main domains, encompassing 15 subtasks to comprehensively evaluate U-MLLMs' understanding, generation, and unified capabilities. Specifically, each unify task includes at least one question, an input image, multiple text choices, and image choices. The image choices consist of a correct answer image and a set of manually crafted negative samples. During the evaluation process, we input the image, question, and text options, and the U-MLLMs are required to select the correct text answer and generate an image. The text answer is evaluated by matching it with the correct answer, while the generated image is compared with the constructed image choices. If the CLIP score between the generated image and the correct answer image is the highest, it is considered correct; otherwise, it is deemed incorrect.

bilities into three areas: (1) Multimodal Understanding, (2) Multimodal Generation, and (3) Unify Capability, highlighting the diverse aspects of model performance.

## 2.1 MULTI-MODAL UNDERSTANDING

**Data Collection.** Multimodal understanding tasks are divided into three subcategories based on visual input type:

- Single-Image Perception and Understanding (SIPU). This task is designed to evaluate image-text pair comprehension.
- Multi-Image & Interleaved Text-Image Understanding (MITIU). Assesses the model's ability to handle and process multi-image and interleaved text-image inputs.
- Video Perception and Understanding (VPU). Measures video comprehension capability.

To ensure comprehensive coverage of various image and video understanding scenarios, we collect 1,900 samples from 5 benchmarks such as MME and MMBench, encompassing over 24 tasks. This includes 1,600 perception tasks, such as OCR, diagram and table understanding, and spatial perception, along with 300 reasoning tasks, including attribute reasoning and action reasoning, with at least 50 QA pairs per sub-task. Additional details can be found in Appendix Figure 7 and Appendix Table 13. More visualization examples can be found in Appendix Figure 5.

**QA Pairs Reformulation.** To standardize the evaluation of the understanding task, we convert all the collcted data into multiple-choice QA pairs, with one correct option and the remaining options carefully designed to be closely related to it. For models that can accept only single-image input, we use the first image from the multi-image input or the first frame from the video input. For models that cannot process video files (e.g., MP4 files), we uniformly sample six key frames from the video to serve as the visual input.

**Evaluation Strategy.** To fairly evaluate MLLM outputs, we apply rule-based filtering to match model responses with answer options, similar to MME-Realworld Zhang et al. (2025d); Fu et al. (2024a). Furthermore, to eliminate positional bias inherent in multiple-choice questions, the correct answer is randomly shuffled among the four available options. We then calculate the average accuracy across all sub-tasks and derive the overall understanding score, providing a fair, robust, and unbiased evaluation of the model's performance.

## 2.2 MULTI-MODAL GENERATION

Multimodal generation involves various tasks for image and video modalities, which can be further subdivided based on application, as shown in Figure 3: 1. *Fine-grained Image Reconstruction (FIR)*. Given an original image, the model is required to restore detailed features and local textures. 2. *Text-guided Image Editing (TIE)*. Edit or modify an image based on textual instructions. 3. *Text-guided Image Generation (TIG)*. Given a text description, the model needs to generate an image that matches it. 4. *Conditional Image-to-Video Generation (CIVG)*. Generate a dynamic video sequence based a given image and text prompt. 5. *Text-guided Video Generation (TVG)*. Generate a video sequence based on a textual description. 6. *Video Prediction (VP)*. Predict subsequent frames or the complete video sequence based on the information from the first frame.

**Data Collection.** Data is collected from benchmark datasets, such as COCO Lin et al. (2014a), Emu-Edit Sheynin et al. (2024), MSR-VTT Xu et al. (2016), ensuring at least 200 samples for each task. For video prediction, videos are sourced from the Pexel Video website[2] and the first frame is used for prediction. Detailed data sources and sample sizes are in Appendix Table 13. More visualization examples can be found in Figure 6.

**QA Pairs Reformulation.** Due to the diversity of generation tasks and their varied data sources, the collected samples contain redundant attributes and inconsistent number of images, videos, and other multimodal data. We aim to provide a streamlined, unified evaluation framework. To achieve this, we contribute the following:

---

[2]https://www.pexels.com/videos/

- **Attribute Unification Pipeline.** First, we summarize all attributes appearing in the data, which exceed 30 types, creating significant complexity. We then manually eliminate task-irrelevant attributes and merge similar attributes across different tasks. For example, text attributes are represented as *Text Prompt*, image attributes as *Src Image* and *Ref Image* based on their input/output roles, and video attributes as *Video*. For any task where an attribute is not required, its corresponding value remains empty.

- **Task-Specific Prompt Engineering.** To ensure that the model can effectively generate outputs that meet the task requirements, we establish specific system prompts for each subtask. Each sample's text prompt or src image serves as the input, while the reference image or video acts as the ground truth answer. Through standardizing attribute values and constructing tailored prompts, we convert diverse samples from different tasks into a unified format for evaluating multimodal generation tasks.

**Evaluation Strategy.** Evaluating multimodal generation tasks with a unified metric is challenging due to the diversity of subdomains and their distinct metrics (e.g., CLIP-I, CLIP-T, FVD, FID). To address this, we: (1) Perform domain-specific preliminary evaluations using standard metrics; (2) Standardize all metrics to a consistent (0, 100) scale, converting non-positive indicators into positive ones; and (3) Compute the average of standardized scores to derive the final generation score. This approach ensures cross-task comparability while maintaining domain-specific evaluation rigor. Detailed metrics and standardization methods are provided in Appendix H.

## 2.3 UNIFY CAPABILITY

MME-U contains five unified subtasks: (1) Common Sense Question Answering (CSQ), (2) Image Editing and Explaining (IEE), (3) SpotDiff (SD), (4) Auxiliary Lines (AL), and (5) Visual CoT (VCoT). Each subtask includes at least 50 manually constructed samples and is structured with task-specific instructions and question templates that require mixed-modality input-to-output generation.

**Common Sense Question Answering.** This task evaluates U-MLLMs' ability to associate commonsense descriptions with visual features, such as linking "the tomb of an ancient Egyptian pharaoh" to a pyramid or "China's national treasure" to a panda. Our approach involves: 1. *Question Construction.* Using GPT-4o, we generate riddle-like questions based on commonsense concepts, with similar but incorrect words as negative options. For example, when the answer is "panda," we select "brown bear" or "polar bear" as negative options to increase difficulty. 2. *Image Collection.* We manually gather images from the internet corresponding to the correct and their negative options. 3. *Task Execution.* U-MLLMs are prompted to select the correct textual option and generate the corresponding image. Detailed procedures and the prompt are in Figure 9(a) and 20.

**Image Editing and Explanation.** This task evaluates U-MLLMs' ability to understand complex editing instructions and generate accurate modifications. Our methodology includes: 1. *Data Collection.* We source data (source images, editing instructions, and reference images) from the Emu-Edit dataset. 2. *Textual QA Construction.* Using GPT-4o, we generate accurate interpretations of editing targets and three incorrect interpretations for textual multiple-choice questions. 3. *Visual QA Construction.* The correct instruction corresponds to the target image in Emu-Edit. For incorrect instructions, we input them into InstructPix2Pix Brooks et al. (2023) to generate negatively edited images, forming image-based multiple-choice questions. 4. *Task Execution.* Given the corresponding prompt, source image, and editing instructions, the model must first produce a correct understanding of the editing target and instructions, and then generate an edited image based on that understanding. Detailed procedures and the system prompt are in Figure 9(b) and 21.

**SpotDiff.** When identifying differences between two similar images, humans typically need to recall the exact locations of these differences to accurately count them. This task evaluates U-MLLMs' ability to identify and recall differences between similar images, simulating human visual reasoning. Our approach involves: 1.*Data Collection:* We sample image pairs with annotated differences from the SpotDiff website[3]. 2. *Textual QA Construction.* Using the annotated difference count, we create textual multiple-choice questions with three incorrect counts (±10 from the true value). 3. *Visual QA Construction.* We place the annotated difference regions from the image pair onto a white background as the correct answer, and randomly crop other areas to place them on the background

---

[3]https://www.allstarpuzzles.com/spotdiff

as incorrect answers. 4. *Task Execution.* U-MLLMs must identify the difference regions between the two images and draw them onto the white background, while also selecting the correct difference count. Detailed procedures in Figure 9(c), and the system prompt is provided in Figure 22.

**Auxiliary Lines.** This task evaluates U-MLLMs' ability to integrate understanding and generation by solving geometric problems requiring auxiliary lines. Our methodology includes: 1. *Data Selection.* We filter the Geometry3K dataset for problems requiring auxiliary lines, extracting logical forms (e.g., "Triangle(A, B, C)"), choices, and answers. 2. *Textual QA Construction.* Using GPT-4o, we generate natural language QA pairs (Question, Choices, Answer) for textual multiple-choice questions. 3. *Visual QA Construction.* We manually solve each sampled geometric problem by drawing the correct auxiliary lines on its diagram, and we construct three additional diagrams with erroneous auxiliary lines. 4. *Task Execution.* U-MLLMs must first generate a geometric diagram with auxiliary lines, and then, based on that diagram, solve the problem by selecting the correct answer. Detailed procedures appear in Figure 9(d), and the prompt is provided in Figure 23.

**Visual CoT.** This task evaluates U-MLLMs' step-by-step decision-making in maze navigation, simulating real-world problem-solving. Our approach involves: 1. *Maze Generation.* Using the OpenAI API, we create maze configurations of varying sizes (3×3, 4×4, 5×5) and layouts. 2. *Action Specification.* For each step, we manually define actions (Up, Right, Down, Left, Finish) and coordinates, updating the maze layout via the API. 3. *QA Construction. - Action Questions.* Options are uniformly set as Up, Right, Down, and Left, with the correct answer manually determined. - *Coordinate/Image Questions.* The correct answers for each step's coordinates and state image are manually defined, and negative samples are also manually specified. 4. *Task Execution.* U-MLLMs receive the initial maze state and task definition, then are prompted to generate actions, coordinates, and maze images iteratively. After the first step, we add the action, coordinate, and image from the previous decision into the system prompt as history information. The model iterates, outputting each step's decision until the target is reached[4]. Detailed procedures appear in Figure 9(e), and the prompts are in Figure 24 and 25.

**Evaluation Strategy.** The unified tasks evaluation combines text-based and image-based multiple-choice questions across all subtasks. Our evaluation framework includes:

1. *Textual QA Evaluation.* For image explanation and editing, we compute CLIP-T similarity between the generated explanation and each option, selecting the one with the highest similarity as the correct answer. For other tasks, U-MLLMs directly select the correct option from the provided multiple-choice set.

2. *Image-Based QA Evaluation.* We compute CLIP-I similarity between the generated image and each candidate image option, selecting the image option with the highest CLIP-I similarity as the model's prediction.

3. *Task-Specific Rules.* For each task we calculate two accuracy metrics—**acc** and **acc+**—where **acc** is defined as the average of the text option accuracy and the image accuracy, and **acc+** represents the accuracy for samples where both the textual and image-based answers are correct. Specifically, for the Visual CoT task, each step is treated as a multiple-choice question, and the accuracy of action, accordinate and image are calculated separately, and the average of these three accuracies is calculated as **acc**, while the accuracy of successfully completing the maze is used to calculate **acc+**. The detailed calculation process can be found in the Appendix H

We then calculate the average **acc** of all subtasks as the unified score, and the overall MME-U score is the average of the understanding, generation, and unified scores.

## 3  EXPERIMENT

We evaluate a total of 31 MLLMs and U-MLLMs, including DeepSeek-Janus-Pro Chen et al. (2025c), DeepSeek-Janus-Flow Ma et al. (2024), SliME Zhang et al. (2024), VITA-1.5 Fu et al. (2025), Gemini2.0-flash DeepMind (2024), Gemini2.5-pro DeepMind (2024), Gemini2.0-flash-exp DeepMind (2024), Gemini2.5-flash-image DeepMind (2024), Claude-3.5sonnet Anthropic

---

[4]task requires an average of 3.5 steps per sample, with a minimum of two and a maximum of seven steps (as shown in Figure 8).

(2024), Emu3 Wang et al. (2024b), GPT-4o OpenAI (2024c), GPT-4o-Image OpenAI (2024c), OmniGen Xiao et al. (2024), DALL-E-2 OpenAI (2024a), DALL-E-3 OpenAI (2024b), Qwen-Image Wu et al. (2025), Qwen-Image-Edit Wu et al. (2025), CogVideoXYang et al. (2025c), HermesFlow Yang et al. (2025a), Qwen2.5-VL-Instruct Wang et al. (2024a), Intern-VL-3 Zhu et al. (2025), Show-o Xie et al. (2025b), Show-o2 Xie et al. (2025c), VILA-U Wu et al. (2024), GILL Koh et al. (2023), Anole Chern et al. (2024), MIO-Instruct Wang et al. (2024c), SEED-LLaMA Ge et al. (2024), MiniGPT-5 Zheng et al. (2023), Bagel Deng et al. (2025), RecA Xie et al. (2025a). Among the baselines, SliME, VITA-1.5, Qwen2.5-VL, Intern-VL-3, Gemini2.0-flash, Gemini2.5-Pro, Claude-3.5-sonnet, OmniGen, DALL-E-2, DALL-E-3, Qwen-Image, Qwen-Image-Edit are specialized understanding models or generative models. Notably, GILL, Anole, MIO-Instruct, SEED-LLaMA, MiniGPT-5, Gemini2.0-flash-exp, Bagel, RecA, GPT-4o-Image, Gemini2.5-flash-image can generate interleaved images and texts. Some MLLMs also can generate arbitrarily interlaced modalities, but they are not available as open-source code or model weights yet, such as PUMA Fang et al. (2024), VITRON Fei et al. (2024) and TextHarmony Zhao et al. (2024).

| Method | LLM | Understanding | | | | Generation | | | | | | | Unify | | | | | | MME-U Score |
|---|---|---|---|---|---|---|---|---|---|---|---|---|---|---|---|---|---|---|---|
| Task Split | | SIPU | MITIU | VPU | Avg | CIVG | FIR | TIE | TIG | TVG | VP | Avg | IEE | CSQ | AL | SD | VCoT | Avg | Avg |
| QA pairs | | 1200 | 400 | 364 | 1964 | 600 | 200 | 200 | 200 | 200 | 194 | 1594 | 200 | 101 | 52 | 104 | 90 | 546 | 4104 |
| *Understanding Models* | | | | | | | | | | | | | | | | | | | |
| SliME-7B | Vicuna-7B | 58.50 | 43.53 | 36.02 | 46.02 | - | - | - | - | - | - | - | - | - | - | - | - | - | 15.34 |
| VITA-1.5 | Qwen-7B | 70.67 | 56.00 | 56.04 | 60.89 | - | - | - | - | - | - | - | - | - | - | - | - | - | 20.30 |
| Qwen2.5-VL-Instruct | Qwen-7B | 75.08 | 53.50 | 57.14 | 61.91 | - | - | - | - | - | - | - | - | - | - | - | - | - | 20.64 |
| Intern-VL3 | Qwen2.5-7B | 75.58 | 52.75 | 58.24 | 62.19 | - | - | - | - | - | - | - | - | - | - | - | - | - | 20.73 |
| Claude-3.5-sonnet | - | 75.83 | 53.25 | 58.52 | 62.53 | - | - | - | - | - | - | - | - | - | - | - | - | - | 20.84 |
| GPT-4o | - | 74.01 | 54.50 | 59.34 | 62.62 | - | - | - | - | - | - | - | - | - | - | - | - | - | 20.87 |
| Gemini2.0-flash | - | 80.92 | 61.75 | 64.64 | 69.10 | - | - | - | - | - | - | - | - | - | - | - | - | - | 23.03 |
| Gemini2.5-Pro | - | 87.00 | 69.00 | 66.21 | 74.07 | - | - | - | - | - | - | - | - | - | - | - | - | - | 24.69 |
| *Generative Models* | | | | | | | | | | | | | | | | | | | |
| DALL-E-2 | - | - | - | - | - | - | - | - | 50.62 | - | - | 8.44 | - | | | | | - | 2.81 |
| DALL-E-3 | - | - | - | - | - | - | - | - | 51.40 | - | - | 8.57 | | | | | | | 2.86 |
| Qwen-Image | - | - | - | - | - | - | - | - | 72.43 | - | - | 12.07 | - | - | - | - | - | - | 4.02 |
| OmniGen | - | - | - | - | - | - | 48.82 | 43.82 | 51.05 | - | - | 23.95 | - | - | - | - | - | - | 7.98 |
| Qwen-Image-Edit | - | - | - | - | - | - | - | - | - | 58.81 | 88.86 | 24.61 | | | | | | | 8.21 |
| CogVideoX | - | - | - | - | - | 68.05 | - | - | 69.62 | 87.61 | - | 37.54 | - | - | - | - | - | - | 12.51 |
| *Unified Models* | | | | | | | | | | | | | | | | | | | |
| Show-o | Phi-1.5 | 32.47 | 34.75 | 25.66 | 30.96 | - | - | - | 43.54 | - | - | 7.26 | - | - | - | - | - | - | 12.74 |
| Emu3 | LLama-8B | 45.75 | 30.50 | 23.32 | 33.19 | - | - | - | 49.08 | - | - | 8.18 | - | - | - | - | - | - | 13.79 |
| HermesFlow | Phi-1.5 | 41.49 | 33.00 | 28.32 | 34.27 | - | - | - | 46.48 | - | - | 7.75 | - | - | - | - | - | - | 14.01 |
| GILL* | OPT-6-7B | 22.18 | 6.00 | 3.56 | 10.58 | - | 50.67 | 35.71 | 46.60 | - | - | 22.16 | 24.25 | 21.29 | 8.66 | 6.67 | 1.90 | 12.55 | 15.10 |
| Janus-Flow | DeepSeek-LLM-1.5b-base | 63.17 | 32.00 | 35.16 | 43.44 | - | - | - | 32.88 | - | - | 5.48 | - | - | - | - | - | - | 16.31 |
| MiniGPT-5* | Vicuna-7B | 19.25 | 10.92 | 15.93 | 15.37 | - | 38.96 | 35.04 | 35.48 | - | - | 18.25 | 22.80 | 34.13 | 14.37 | 5.00 | 2.08 | 15.67 | 16.43 |
| Janus-Pro | DeepSeek-LLM-7b-base | 59.56 | 43.50 | 42.22 | 48.43 | - | - | - | 35.29 | - | - | 5.88 | - | - | - | - | - | - | 18.10 |
| VILA-U | LLama-7B | 51.04 | 32.25 | 36.54 | 39.95 | - | - | - | 45.10 | 49.64 | - | 15.79 | - | - | - | - | - | - | 18.58 |
| Anole* | - | 17.17 | 14.50 | 9.00 | 13.56 | - | 36.64 | 43.42 | 41.52 | - | - | 19.91 | 18.55 | 59.65 | 14.42 | 15.00 | 3.89 | 22.30 | 18.59 |
| Show-o2 | Qwen2.5-7B | 68.33 | 47.00 | 50.00 | 55.11 | - | - | - | 50.18 | - | - | 8.36 | - | 66.34 | - | - | - | 13.27 | 23.94 |
| SEED-LLaMA* | LLaMA2-Chat-13B | 49.17 | 33.00 | 36.26 | 39.48 | - | 57.00 | 42.26 | 41.96 | - | - | 23.54 | 22.00 | 51.49 | 12.50 | 22.00 | 3.61 | 22.32 | 28.45 |
| MIO-Instruct* | MIO-7B | 52.00 | 33.50 | 39.01 | 41.50 | 51.24 | 59.29 | 43.66 | 48.23 | 51.88 | 66.37 | 53.45 | 24.16 | 38.50 | 8.66 | 11.50 | 0 | 16.56 | 37.17 |
| Bagel* | Qwen2.5-7B | 76.67 | 53.00 | 51.10 | 60.26 | - | - | - | 44.51 | 45.46 | 59.91 | 24.98 | 33.34 | 7.23 | 31.73 | 25.50 | 11.20 | 35.80 | 40.35 |
| GPT-4o-Image* | - | 65.50 | 49.50 | 45.05 | 53.35 | - | - | - | 60.07 | 46.58 | 65.65 | 28.72 | 43.00 | 86.64 | 42.31 | 22.50 | 11.02 | 41.10 | 41.06 |
| RecA* | Qwen2.5-7B | 76.00 | 57.00 | 56.04 | 63.01 | - | - | - | 46.30 | 46.87 | 72.88 | 27.36 | 35.67 | 79.21 | 33.66 | 26.50 | 12.22 | 37.45 | 42.60 |
| Gemini2.0-flash-exp* | - | 72.58 | 68.25 | 54.90 | 65.24 | - | 77.61 | 43.54 | 57.56 | - | - | 29.79 | 38.42 | 74.75 | 47.12 | 26.00 | 12.41 | 40.74 | 45.57 |
| Gemini2.5-flash-image* | - | 80.25 | 70.75 | 58.79 | 69.93 | - | - | - | 66.29 | 52.90 | 85.32 | 34.09 | 48.75 | 86.02 | 59.62 | 25.50 | 15.23 | 47.02 | 50.04 |

Table 2: **Comparison of MLLMs on understanding, generation, unifying tasks, and overall MME-U Score. SIPU:** Single Image Perception & Understanding; **MITIU:** Multiple & Interleaved Image-Text Understanding; **VPU:** Video Perception & Understanding; **CIVG:** Conditional Image-to-Video Generation; **FIR:** Fine-grained Image Reconstruction; **TIE:** Text-Guided Image Editing; **TIG:** Text-to-Image Generation; **TVG:** Text-to-Video Generation; **VP:** Video Prediction; **IEE:** Image Editing and Explaining; **CSQ:** Common Sense Question Answering; **AL:** Auxiliary Lines; **SD:** SpotDiff; **VCoT:** Visual CoT. * denotes U-MLLMs with the ability to generate interleaved images and texts, while '-' indicates that the model is unable to finish the corresponding task and underlined content signifies the best performance within a single model across all methods on this task.

## 3.1 RESULTS

The evaluation results of various MLLMs in MME-U, as shown in Table 2, indicate that Gemini2.5-flash-image achieves the highest MME-U score at 50.04. Although compared to MIO-Instruct it does not encompass all subtasks, it demonstrates very balanced performance across understanding, generation, and unify tasks, unlike other models that may exhibit deficiencies in certain test dimensions.

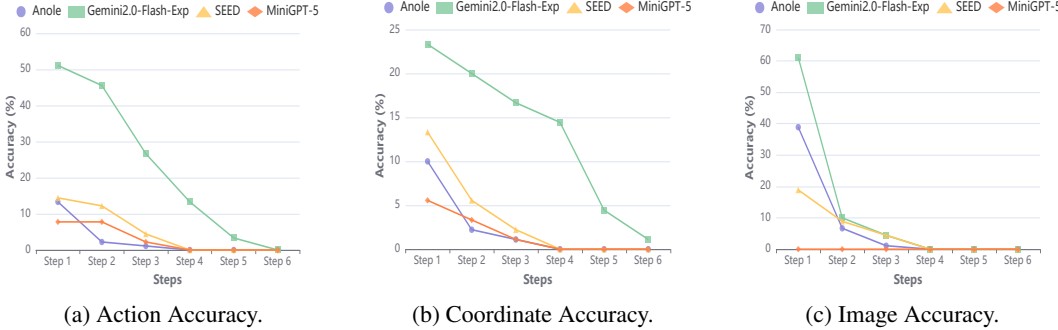

(a) Action Accuracy.    (b) Coordinate Accuracy.    (c) Image Accuracy.

Figure 4: **Accuracy distribution across different dimensions on visual cot task.** (a) action, (b) location, and (c) image.

It is evident that, compared to traditional MLLMs or generative models, U-MLLMs are capable of handling a wider range of tasks, including more complex image-text interleaved reasoning. However, overall, the development of U-MLLMs is still in its early stages, and even the best-performing models only achieve scores of around 50 on MME-U. Next, we will provide a separate analysis of understanding, generation, and unify tasks.

**Understanding.** It is evident that Gemini2.5-flash-image DeepMind (2024) demonstrates the best understanding capability among U-MLLMs, while also being a closed-source model. For open-source models, the two U-MLLMs with the best understanding capabilities are RecA Xie et al. (2025a) and Bagel Deng et al. (2025). These models utilize two separate vision encoders to handle generation and understanding tasks independently, thus overcoming the limitations of tokenizers like VQGAN Yu et al. (2022), which are not well-suited for extracting semantic features. In contrast, models like Emu3 Wang et al. (2024b) and Show-o Xie et al. (2025b), which use a single tokenizer for all image tasks, perform poorly on understanding tasks and still show a significant performance gap compared to currently available open-source MLLMs of similar size. However, our experiments also show that models like Janus-Pro perform poorly on generation tasks. They even fail to support multimodal generation, scoring zero on unified tasks. Therefore, how to strike a balance between understanding and generation capabilities, or whether the two capabilities can indeed complement each other, remains an open question. We also see potential in bridging this gap in understanding capabilities by leveraging existing U-MLLMs alongside strong MLLM baselines. For instance, MIO-instruct Wang et al. (2024c) achieves impressive understanding results through extensive training data, including video, audio, image-text pairs, and a complex three-stage training pipeline. This suggests that U-MLLMs may require a broader variety or larger volume of data for training.

**Generation.** We compare the performance gap between various U-MLLMs and current state-of-the-art generative models such as DALLE-3. It is evident that, compared to understanding capabilities, the gap in generation tasks is not as significant. For the simplest TIG task, Gemini-2.0-flash-exp even outperforms the best generative model DALLE-3 by six points, while U-MLLMs such as EMU3, HermesFlow, and GILL all achieve an average score above 48. However, it is clear that most U-MLLMs still do not perform well on video generation tasks. Notably, although the original paper for Emu3 mentions its capability for video generation, the corresponding checkpoints have not been released. It's clear that the open-source community still has a long way to go before U-MLLMs that support video generation become widely available. Detailed results on the generation tasks can be found in Table 9. In Figure 10, we showcase the generation results from various models using the following text prompt: "A man is standing in a park with a 'Run for Rights' banner in the background. He is wearing a white t-shirt with the number 28 on it, grey shorts, and grey socks with black shoes. The park is filled with people, some sitting on benches, and there is a bicycle leaning against a tree." It is evident that most generated images, such as those from VILA-U, Show-O, and Janus-Pro, fail to capture key details from the caption, such as the number on the jersey or specific text. In contrast, the results from EMU3 more closely resemble the textual description, while MIO-Instruct's outputs are more aligned with realistic scenes (we hypothesize this is because MIO-Instruct was trained on a large amount of real-world data, enhancing its ability to generate lifelike images). However, when it comes to image detail, current open-source U-MLLMs still lag significantly behind dedicated generative models.

**Unify Capability.** Our systematic unify task testing shows that, while U-MLLMs have indeed expanded the potential for such tasks compared to traditional understanding/generation models, their performance remains insufficient. For each unify task in Table 2, we require the models to generate the correct image and perform correct reasoning. Under these conditions, even for simple tasks such as answering common questions and generating images, the best open-sourced model (Anole) only achieves an accuracy of 59.65% and accuracy-plus of 38% (Table 3). In other tasks, no open-sourced model is able to surpass the 30% accuracy. It is worth noting that models perform even worse on tasks like Visual CoT, which require multi-step image generation and reasoning. No model is able to successfully complete tasks involving multiple steps. This finding underscores the importance of our MME-U, as relying solely on case studies to demonstrate a model's mix-modality generation capabilities is clearly insufficient. We will further analyze these models' performance, weaknesses, and provide examples in the analysis section.

Due to space limitations, we have included additional in-depth analyses in Appendix F and G, which contain expanded analyses of current U-MLLMs' weakness, detailed visualizations of the U-MLLMs' generation results, as well as specific examples from the unify tasks.

| Method | IEE | | | | CSQ | | | | AL | | | | SD | | | | VCoT | | | | | Unify Score | |
|---|---|---|---|---|---|---|---|---|---|---|---|---|---|---|---|---|---|---|---|---|---|---|---|
| Metric | Text Acc | Image Acc | Acc | Acc+ | Text Acc | Image Acc | Acc | Acc+ | Text Acc | Image Acc | Acc | Acc+ | Text Acc | Image Acc | Acc | Acc+ | Action Acc | Coordinate Acc | Image Acc | Acc | Acc+ | Acc | Acc+ |
| GILL | 21.00 | 27.50 | 24.25 | 8.00 | 14.75 | 27.82 | 21.29 | 4.95 | 7.69 | 9.62 | 8.66 | 1.92 | 0 | 13.50 | 6.75 | 0 | 0.63 | 0 | 5.08 | 1.90 | 0 | 12.55 | 2.98 |
| Show-o2 | - | - | - | - | 79.21 | 53.47 | 66.34 | 56.44 | - | - | - | - | - | - | - | - | - | - | - | - | - | 13.27 | 11.29 |
| MiniGPT-5 | 21.50 | 24.00 | 22.75 | 5.00 | 29.70 | 38.56 | 34.13 | 15.81 | 5.66 | 23.08 | 14.37 | 3.84 | 4.00 | 6.00 | 5.00 | 2.00 | 2.22 | 1.27 | 2.92 | 2.13 | 0 | 15.67 | 5.33 |
| MIO-Instruct | 24.00 | 24.00 | 24.00 | 7.00 | 77.24 | 0 | 38.62 | 0 | 17.31 | 0 | 8.66 | 0 | 23.00 | 0 | 11.50 | 0 | 0 | 0 | 0 | 0 | 0 | 16.56 | 1.40 |
| Anole | 17.00 | 20.00 | 18.50 | 3.00 | 70.30 | 48.52 | 59.41 | 38.00 | 15.38 | 13.46 | 14.42 | 3.84 | 17.00 | 13.00 | 15.00 | 2.00 | 3.49 | 0.64 | 7.62 | 3.91 | 0 | 22.30 | 9.17 |
| SEED-LLaMA | 19.00 | 25.00 | 22.00 | 4.50 | 56.44 | 46.53 | 51.49 | 37.62 | 13.46 | 11.54 | 12.50 | 3.84 | 23.00 | 21.00 | 22.00 | 4.00 | 4.13 | 2.85 | 3.81 | 3.61 | 0 | 22.32 | 9.99 |
| Bagel | 22.78 | 43.89 | 33.34 | 11.67 | 90.10 | 64.36 | 77.23 | 63.37 | 42.31 | 21.15 | 31.73 | 9.62 | 27.00 | 24.00 | 25.50 | 7.00 | 10.83 | 12.08 | 10.69 | 11.20 | 0 | 35.80 | 18.33 |
| RecA | 23.39 | 47.95 | 35.67 | 10.53 | 90.10 | 68.32 | 79.21 | 59.41 | 44.23 | 23.08 | 33.66 | 13.46 | 26.00 | 27.00 | 26.50 | 9.00 | 11.53 | 14.03 | 11.11 | 12.22 | 0 | 37.45 | 18.48 |
| Gemini2.0-flash-exp | 33.00 | 43.50 | 38.25 | 10.00 | 83.17 | 66.33 | 74.75 | 63.37 | 59.61 | 34.62 | 47.12 | 30.77 | 28.00 | 24.00 | 26.00 | 5.00 | 17.77 | 10.14 | 9.44 | 12.41 | 0 | 40.74 | 21.05 |
| GPT-4o-Image | 25.50 | 60.50 | 43.00 | 17.00 | 97.03 | 76.24 | 86.64 | 74.26 | 42.31 | 42.31 | 42.31 | 15.38 | 29.00 | 21.00 | 25.00 | 9.00 | 11.53 | 8.47 | 13.06 | 11.02 | 0 | 41.10 | 23.13 |
| Gemini2.5-flash-image | 32.00 | 65.50 | 48.75 | 19.00 | 98.92 | 73.12 | 86.02 | 73.12 | 67.31 | 51.92 | 59.62 | 32.69 | 27.00 | 24.00 | 25.50 | 9.00 | 20.97 | 11.25 | 13.47 | 15.23 | 0 | 47.02 | 26.76 |

Table 3: **Comparison of U-MLLMs on various unify tasks and overall unify Score.**

## 4 CONCLUSION AND LIMITATION

The MME-U benchmark framework presented here serves as a foundational step towards evaluating U-MLLMs on a diverse array of tasks encompassing multimodal understanding, generation, and their integration. This benchmark reveals the current landscape of U-MLLMs, highlighting their capabilities and areas for improvement. While these models demonstrate proficiency in handling various multimodal tasks, they struggle with balancing understanding and generation, handling complex instructions, and performing well on unify tasks. Moreover, current U-MLLMs exhibit significant inconsistencies in aligning textual instructions with their visual outputs, highlighting the need for further research to improve multimodal reasoning and generation integration. However, this study simplifies the evaluation of unify tasks by framing image generation as multiple-choice questions, which may allow model "hacking". For instance, SEED-generated images may not meet style standards but achieve high similarity scores, inflating accuracy metrics. Future work will incorporate MLLM or CLIP scores for stricter evaluation.

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

# Appendix

**Ethics statement.** This work involves no human subjects or sensitive data, uses only publicly available, properly licensed datasets without identifiable information, and presents no foreseeable safety, privacy, bias/discrimination, misuse, conflict-of-interest, or legal concerns.

**Reproducibility statement.** To ensure the full reproducibility of our benchmark, we have provided comprehensive implementation details throughout the paper. The construction and statistics of our MME-Unify are detailed in Sec 2 and Figure 7, Figure 9.

**The Use of Large Language Models (LLMs).** The authors employ the LLM exclusively for grammar and style refinement. The LLM was not used to generate or substantively revise scientific content, design experiments, analyze data, write code, or select references.

## A    RELATED WORKS

**Unified Multimodal Large Language Models.** Building on the success of MLLMs Diao et al. (2025b); Wang et al. (2024a); Fu et al. (2025); Yu et al. (2025); Zhang et al. (2024; 2025a;b); Chen et al. (2025b;a); Diao et al. (2026; 2025a), recent studies U-MLLMs, which can understand and generate multiple modalities in an end-to-end manner. Some approaches have adopted a unified training objective, projecting both text and images into a discrete token space and employing a next-token prediction loss function for training Wang et al. (2024c); Wu et al. (2024); Team (2024). This training method and framework are notably straightforward. However, using discrete image tokens (e.g., extracted from VQVAE image features) may not be optimal for image understanding tasks. Therefore, works like Janus-Flow Chen et al. (2025c), Janus-Pro Chen et al. (2025c), among others, have employed different vision encoders such as VQVAE for image generation and SigLIP for image comprehension, significantly enhancing the understanding capabilities of U-MLLMs. Additionally, other methods have found that diffusion training is more suitable for image generation. Thus, adopting diffusion-based training for image generation and next-token prediction for text generation aims to strengthen the image generation capabilities further Xie et al. (2025b); Zhou et al. (2025). Recent research has also explored fine-tuning U-MLLMs to further enhance their performance on unified tasks Li et al. (2025). However, despite the rapid advancements of U-MLLMs, there remains a lack of comprehensive benchmarks for systematically and fairly evaluating their capabilities in understanding, generation, and multimodal synthesis tasks.

**Benchmarks for Understanding.** With the rapid development of MLLMs, several concurrent works Fu et al. (2024b) have proposed various benchmarks to evaluate the models' capabilities in multimodal comprehension tasks, such as single-image perception and understanding Fu et al. (2023); Zhang et al. (2025d) (e.g., MME series), interleaved image & text understanding, and video understanding Fang et al. (2025) (e.g., MMBench-Video, Video-MME). Additionally, some benchmarks focus on multimodal safety Zhang et al. (2025c) or mathematical reasoning Yan et al. (2024). These benchmarks differ in coverage and metrics.

**Benchmarks for Generation.** Various benchmarks have been proposed to assess multi-modal generation capabilities Wang & Yang (2024); Sheynin et al. (2024); Deng et al. (2009); Xu et al. (2016); Ku et al. (2024); Li et al. (2023), including tasks like image reconstruction Deng et al. (2009), image editing Sheynin et al. (2024); Ku et al. (2024), and conditional image & video generation Wang & Yang (2024); Lin et al. (2014b). However, these benchmarks mainly focus on individual tasks within single modalities, failing to capture the full scope of multi-modal comprehension and generation. While some benchmarks, such as SEED-Bench-2 Li et al. (2023) and MMIE Xia et al. (2025), provide hierarchical evaluation for both understanding and generation, they do not assess unified tasks, and the range of tasks is limited.

## B    STABILITY ANALYSIS OF MME-UNIFY

For the multimodal understanding task, MME-Unify collects 1,900 multiple-choice questions from five public benchmarks spanning 24 subtasks (SIPU, MITIU, VPU), with at least 50 QA pairs per subtask. For the multimodal generation task, we evaluate six subtasks—FIR, TIE, TIG, CIVG, TVG,

and VP—with at least 200 samples per subtask, yielding over 1,200 evaluated instances. These are scored using standard domain-specific metrics and subsequently normalized to a [0, 100] scale. For the Unify tasks, four subtasks (IEE, CSQ, AL, SD) each contain at least 50 samples, and each sample induces two multiple-choice questions (text and image), resulting in over 100 questions per subtask. In particular, the VCoT subtask includes 90 mazes with an average of 3.5 steps; each step requires three decisions (action, coordinate, image), yielding approximately 945 step-level questions (90 × 3.5 × 3). In total, MME-Unify comprises over 3,200 multiple-choice questions alongside more than 1,200 generation instances.

We further assess the stability of MME-U scores and rankings. Specifically, we halve the benchmark samples and re-evaluate the four U-MLLMs. As shown in Table 4, While a few subtask rankings (e.g., for Anole and SEED-LLaMA) exhibit minor fluctuations due to the reduced sample size, the overall MME-U score ranking remains consistent with the full-dataset evaluation. These results indicate that the benchmark's sample sizes and scoring strategy can assess stable scores and consistent model rankings, supporting the reliability of MME-U for comparative assessment.

| Split | Method | Understanding | Generation | Unify | MME-U |
|---|---|---|---|---|---|
| **Split A** | MiniGPT5 | 14.08 | 18.60 | 14.65 | 15.78 |
| | Anole | 12.75 | 19.50 | 20.24 | 17.53 |
| | SEED-LLaMA | 24.50 | 23.92 | 21.57 | 23.33 |
| | MIO-Instruct | 40.88 | 52.68 | 16.34 | 36.63 |
| **Split B** | MiniGPT5 | 16.67 | 19.40 | 15.01 | 17.03 |
| | Anole | 14.33 | 19.73 | 23.40 | 19.15 |
| | SEED-LLaMA | 27.67 | 23.73 | 23.15 | 24.85 |
| | MIO-Instruct | 42.15 | 53.98 | 16.78 | 37.64 |
| **Overall** | MiniGPT5 | 15.37 | 18.25 | 15.67 | 16.43 |
| | Anole | 13.56 | 19.91 | 22.30 | 18.59 |
| | SEED-LLaMA | 26.19 | 23.54 | 22.32 | 28.45 |
| | MIO-Instruct | 41.50 | 53.45 | 16.56 | 37.17 |

Table 4: Split-half evaluation on four U-MLLMs.

We aslo supplement the split-half experimental results for each sub-task in the unified tasks. As shown in Table 5, The ranking of the four representative models remains stable basically across different half-sample divisions for the unify tasks.

| Method | IEE | | | CSQ | | | AL | | | SD | | | VCoT | | | | Avg |
|---|---|---|---|---|---|---|---|---|---|---|---|---|---|---|---|---|---|
| **Split A** | | | | | | | | | | | | | | | | | |
| MiniGPT5 | 19 | 24 | 21.5 | 23.8 | 35.6 | 29.6 | 7.7 | 23.1 | 15.4 | 4 | 8 | 6 | 3.2 | 1.3 | 3.8 | 2.8 | 14.7 |
| Anole | 15 | 17 | 16 | 70.3 | 51.4 | 60.8 | 19.2 | 11.5 | 15.4 | 20 | 12 | 16 | 4.4 | 0 | 6.4 | 3.6 | 20.2 |
| SEED-LLAMA | 17 | 22 | 19.5 | 54.1 | 43.2 | 48.7 | 11.5 | 11.5 | 11.5 | 26 | 24 | 25 | 3.2 | 2.5 | 3.8 | 3.2 | 21.6 |
| MIO-Instruct | 22 | 21 | 21.5 | 73.3 | 0 | 36.6 | 23.1 | 0 | 11.5 | 24 | 0 | 12 | 0 | 0 | 0 | 0 | 16.3 |
| **Split B** | | | | | | | | | | | | | | | | | |
| MiniGPT5 | 23 | 24 | 23.5 | 35.6 | 41.6 | 38.6 | 3.9 | 23.1 | 13.5 | 4 | 4 | 4 | 1.3 | 1.3 | 1.9 | 1.5 | 15 |
| Anole | 19 | 23 | 21.5 | 81 | 47.6 | 64.3 | 11.5 | 15.4 | 13.5 | 14 | 14 | 14 | 2.5 | 1.3 | 8.9 | 4.2 | 23.4 |
| SEED-LLAMA | 21 | 28 | 24.5 | 58.7 | 50.8 | 54.8 | 15.4 | 11.5 | 13.5 | 20 | 18 | 19 | 5.1 | 3.2 | 3.8 | 4 | 23.2 |
| MIO-Instruct | 26 | 27 | 26.5 | 79.2 | 0 | 39.6 | 11.5 | 0 | 5.8 | 22 | 0 | 11 | 0 | 0 | 0 | 0 | 16.8 |
| **Overall** | | | | | | | | | | | | | | | | | |
| MiniGPT5 | 21.5 | 24 | 22.8 | 29.7 | 38.6 | 34.1 | 5.7 | 23.1 | 14.4 | 4 | 6 | 5 | 2.2 | 1.3 | 2.9 | 2.1 | 15.7 |
| Anole | 17 | 20 | 18.5 | 70.3 | 49 | 59.7 | 15.4 | 13.5 | 14.4 | 17 | 13 | 15 | 3.5 | 0.6 | 7.6 | 3.9 | 22.3 |
| SEED-LLAMA | 19 | 25 | 22 | 56.4 | 46.5 | 51.5 | 13.5 | 11.5 | 12.5 | 23 | 21 | 22 | 4.1 | 2.9 | 3.8 | 3.6 | 22.3 |
| MIO-Instruct | 24 | 24 | 24 | 77.2 | 0 | 38.5 | 17.3 | 0 | 8.7 | 23 | 0 | 11.5 | 0 | 0 | 0 | 0 | 16.6 |

Table 5: **Performance comparison on unified tasks across different data splits. T**: Text Accuracy; **I**: Image Accuracy; **A**: Average Accuracy; **Act**: Action Accuracy; **Crd**: Coordinate Accuracy. **IEE**: Image Editing and Explaining; **CSQ**: Common Sense Question Answering; **AL**: Auxiliary Lines; **SD**: SpotDiff; **VCoT**: Visual Chain-of-Thought.

## C    DEFINITION OF TRADITIONAL MLLM AND UNIFY MLLM

We use the term "**Traditional MLLMs**"to denote understanding-centric multimodal models that accept visual inputs (images/videos) but do not natively generate visual outputs at inference time, such as Qwen2.5-VL, and their outputs are textual only. In contrast, "**Generative MLLMs**" specialize in visual generation, without general multimodal understanding capability, and "**U-MLLMs**" support both understanding and generation, including interleaved image–text generation.

## D    COMPARISON OF EVALUATION STRATEGIES FOR THE UNIFY TASKS

We compare two evaluation strategies for assessing Unify Tasks: (i) a generation-based strategy, which we term CLIP-Choice, and (ii) an option-selection strategy, which we term Select-Choice. In the CLIP-Choice strategy, the model first generates an image; we then compute its CLIP similarity to four candidate references and take the argmax as the model's implicit choice. In contrast, Select-Choice feeds all image options to the model and asks it to select the correct one directly. We condect experiments based on **Gemini 2.0-flash-exp**, which is shown in Table 6, Compared with the CLIP-Choice strategy, the Select-Choice strategy achieves higher accuracy on text and image multiple-choice questions evaluated on all unified tasks except the IEE task. This may be because choosing the correct option is easier than generating the correct image. Moreover, after introducing the reference image into the model input, the model is more likely to rely on the given reference image answer rather than the image generated by the model itself. Therefore, compared with the CLIP-Similarity evaluation strategy, Select-Choice deviates from our original intention of evaluating the unify capability of U-MLLM on the unify task.

| Strategy | IEE | | | | CSQ | | | | AL | | | | SD | | | | VCoT | | | | | Unify | |
|---|---|---|---|---|---|---|---|---|---|---|---|---|---|---|---|---|---|---|---|---|---|---|---|
| | Text | Img | Acc | Acc+ | Text | Img | Acc | Acc+ | Text | Img | Acc | Acc+ | Text | Img | Acc | Acc+ | Act | Coord | Img | Acc | Acc+ | Acc | Acc+ |
| Select-Choice | 25.0 | 17.5 | 21.3 | 14.0 | 97.0 | 92.1 | 94.6 | 91.1 | 65.4 | 78.9 | 72.1 | 52.0 | 33.0 | 45.0 | 39.0 | 14.0 | 31.1 | 19.1 | 27.1 | 25.8 | 0 | 50.5 | 34.6 |
| CLIP-Choice | 33.0 | 43.5 | 38.3 | 10.0 | 83.2 | 66.3 | 74.8 | 63.4 | 59.6 | 34.6 | 47.1 | 30.8 | 28.0 | 24.0 | 26.0 | 5.0 | 17.8 | 10.1 | 9.4 | 12.4 | 0 | 40.7 | 21.1 |

Table 6: Comparison of two evaluation strategies on the Unify task with detailed metrics per subtask.

## E    DISSCUSSION FOR THE EFFECTIVENESS OF OUR EVALUATION STRATEGY

In MME-U, our goal is not to independently evaluate the capability of understanding and generation, but rather to quantify the *synergistic capability* of U-MLLMs to understand an input, reason and then generate the correct visual output. To this end, we adopt a CLIP-I based multiple-choice image evaluation strategy that is aligned with the text multiple-choice format used in other parts of our benchmark. Below we explain and provide experimental evidence supporting its effectiveness.

### E.1    WHY CLIP-I MULTIPLE CHOICE FOR UNIFIED EVALUATION

**Unified representation and metric design.** Our benchmark aims to measure *unified* understanding–generation synergy with a *single, discrete* metric. Existing continuous metrics like LLM-Judge provide fine-grained and interpretable assessments of visual quality, but they are computationally expensive and difficult to map directly to a discrete "success" signal that is comparable to text multiple-choice accuracy. To achieve the balance, we treat image evaluation similar to text multiple-choice, each unified task instance is associated with several candidate images, and the model-generated image is scored via CLIP-I similarity against each candidate. The candidate with the highest CLIP-I similarity is treated as the model's implicit choice, and we compute accuracy over all instances. This design has two advantages: (i) it maps multimodal synergistic behavior to a discrete accuracy score that is directly comparable across tasks, and (ii) it enables a unified leaderboard over both text and image questions.

**Task design mitigates CLIP-I score hacking.**   A natural concern is that CLIP-based evaluation might be exploited by score hacking. For example, by generating generic but CLIP-friendly images that superficially match the prompt. To this end, our unified tasks are designed explicitly to minimize this risk, since each sub-task requires *prior understanding and reasoning* before correct generation is even possible. For instance, AL requires the model to correctly parse a geometric configuration and

Table 7: Kendall's $\tau_b$ correlation between CLIP-I scores and human ratings on 200 randomly sampled unified-task instances. We observe a consistently strong correlation ($\tau_b \approx 0.69$–$0.73$) across sub-tasks, with an overall correlation of $0.71$, indicating stable alignment between CLIP-I and human perceptual judgments under our task design.

|                    | IEE   | CSQ   | AL    | SD    | VCoT  | Overall |
|--------------------|-------|-------|-------|-------|-------|---------|
| Kendall's $\tau_b$ | 0.712 | 0.732 | 0.717 | 0.707 | 0.692 | 0.709   |

infer the appropriate auxiliary line before drawing; VCoT requires the model to infer the next action and generate the corresponding future state. In such settings, the correct visual output depends on a specific reasoning chain rather than generic visual patterns. Since the candidate options encode these reasoning outcomes (e.g., correct vs. incorrect auxiliary lines, correct vs. incorrect navigation states), an image generated solely from the surface form of the instruction, without solving the underlying reasoning problem, is extremely unlikely to match the correct option under CLIP-I. In other words, the design of unified tasks anchors CLIP-I scores to the correctness of understanding and reasoning of given input, rather than to superficial prompt alignment.

### E.2 Correlation with Human Judgments

To empirically verify that CLIP-I under our task design aligns with human perception, we conduct a correlation study on a subset of the benchmark. We randomly sample 200 instances from the unified tasks and evaluate the generation results of four advanced U-MLLMs (Bagel Deng et al. (2025), GPT-4o-Image OpenAI (2024c), RecA Xie et al. (2025a), Gemini2.5-Flash-Image DeepMind (2024)) using both CLIP-I and human ratings.

Three expert annotators score each model's generated image on a 1–5 scale along three dimensions: (i) *Text-Following*, (ii) *Image Quality*, and (iii) *Reference Similarity*. We average scores across annotators and dimensions to obtain a single human rating per sample. We then compute Kendall's $\tau_b$ correlation coefficient between CLIP-I scores and human ratings for each sub-task and overall.

As shown in Table 7, the Kendall's $\tau_b$ values for all sub-tasks lie in the range of approximately 0.69-0.73, with an overall correlation of 0.71, which falls into the regime of *strong correlation*.

### E.3 Ranking Consistency with LLM-Judge

We further validate the robustness of our evaluation strategy by comparing CLIP-I based rankings with an LLM-Judge (GPT-4o) based ranking that closely mirrors human evaluation.

On the complete unified dataset, we evaluate five U-MLLMs (Show-o2, Bagel, RecA, GPT-4o-Image, Gemini2.5-Flash-Image) under two scoring protocols: (i) CLIP-I multiple-choice accuracy, and (ii) an LLM-Judge that follows the same scoring dimensions as the human evaluation (text-following, image quality, and reference similarity), aggregated into a normalized score. For each model, we compute per-task and overall scores under both evaluation strategies for direct comparison. As shown in Table 8, the relative ordering of models is *highly consistent* between CLIP-I and LLM-Judge across most sub-tasks and in the overall average. In particular, Gemini2.5-Flash-Image and GPT-4o-Image consistently achieve the top two positions under both metrics, while RecA and Bagel form a stable middle tier above Show-o2. Although Bagel and RecA exhibit similar performance on the VCoT task, leading to a local inversion in their per-task ranking for that specific sub-task, the global ordering of the five models remains stable between the two scoring methods.

This consistency in model rankings indicates that CLIP-I based multiple-choice accuracy captures essentially the same relative performance ranking as a much more expensive LLM-Judge strategy. Together with the Kendall's $\tau_b$ analysis, this provides robust evidence that CLIP-I is a reliable and efficient evaluation strategy for human and LLM-based judgments under our Benchmark.

| Model | IEE | CSQ | AL | SD | VCoT | Avg |
|---|---|---|---|---|---|---|
| **CLIP-I** | | | | | | |
| Show-o2 | — | 53.47 | — | — | — | 10.69 |
| Bagel | 43.89 | 64.36 | 21.15 | 24.00 | 10.69 | 32.82 |
| RecA | 47.95 | 68.32 | 23.08 | 27.00 | 11.11 | 35.49 |
| GPT-4o-Image | 60.50 | 76.24 | 42.31 | 21.00 | 13.06 | 42.62 |
| Gemini 2.5-Flash-Image | 65.50 | 73.12 | 51.92 | 24.00 | 13.47 | 45.60 |
| **LLM-Judge** | | | | | | |
| Show-o2 | — | 56.33 | — | — | — | 11.27 |
| Bagel | 49.47 | 66.93 | 37.07 | 47.67 | 52.33 | 50.69 |
| RecA | 49.60 | 68.07 | 43.73 | 48.67 | 52.00 | 52.41 |
| GPT-4o-Image | 68.87 | 78.80 | 58.60 | 45.47 | 52.60 | 60.87 |
| Gemini 2.5-Flash-Image | 80.33 | 72.33 | 65.40 | 53.07 | 58.67 | 65.96 |

Table 8: **Comparison of CLIP-I and LLM-Judge based evaluation on unified image QA tasks.**

| Method | CIVG | | | | FIR | | TIE | | | TIG | | | TVG | | | | VP | | | Generation Score |
|---|---|---|---|---|---|---|---|---|---|---|---|---|---|---|---|---|---|---|---|---|
| Metric | FVD Score | FID Score | CLIPSIM | Avg | 1-LPIPS | Avg | CLIP-I | CLIP-T | Avg | CLIP-I | CLIP-T | Avg | FVD Score | FID Score | CLIPSIM | Avg | FVD Score | FID Score | Avg | Avg |
| *Generative Models* | | | | | | | | | | | | | | | | | | | | |
| DALL-E-2 | - | - | - | - | - | - | - | - | - | 69.33 | 31.91 | 50.62 | - | - | - | - | - | - | - | 8.44 |
| DALL-E-3 | - | - | - | - | - | - | - | - | - | 70.11 | 32.68 | 51.40 | - | - | - | - | - | - | - | 8.57 |
| Qwen-Image | - | - | - | - | - | - | - | - | - | 89.24 | 37.61 | 63.43 | - | - | - | - | - | - | - | 10.57 |
| Qwen-Image-Edit | - | - | - | - | 80.86 | 80.86 | 74.89 | 22.72 | 48.81 | - | - | - | - | - | - | - | - | - | - | 21.61 |
| OmniGen | - | - | - | - | 48.82 | 48.82 | 65.63 | 22.72 | 43.82 | 73.97 | 28.12 | 51.05 | - | - | - | - | - | - | - | 23.95 |
| CogVideoX | 83.91 | 87.02 | 33.23 | 68.05 | - | - | - | - | - | - | - | - | 87.82 | 84.28 | 36.77 | 69.62 | 89.92 | 85.30 | 87.61 | 37.54 |
| *Unified Models* | | | | | | | | | | | | | | | | | | | | |
| DeepSeek-Flow | - | - | - | - | - | - | - | - | - | 52.38 | 13.38 | 32.88 | - | - | - | - | - | - | - | 5.48 |
| DeepSeek-Janus-Pro | - | - | - | - | - | - | - | - | - | 55.46 | 15.11 | 35.29 | - | - | - | - | - | - | - | 5.88 |
| Show-o | - | - | - | - | - | - | - | - | - | 62.10 | 24.97 | 43.54 | - | - | - | - | - | - | - | 7.26 |
| HermesFlow | - | - | - | - | - | - | - | - | - | 65.37 | 27.58 | 46.48 | - | - | - | - | - | - | - | 7.75 |
| Emu3 | - | - | - | - | - | - | - | - | - | 68.54 | 29.62 | 49.08 | - | - | - | - | - | - | - | 8.18 |
| VILA-U | - | - | - | - | - | - | - | - | - | 62.54 | 27.66 | 45.10 | 57.35 | 66.36 | 25.22 | 49.64 | - | - | - | 15.80 |
| MiniGPT-5 | - | - | - | - | 38.96 | 38.96 | 55.86 | 14.21 | 35.04 | 56.33 | 14.62 | 35.48 | - | - | - | - | - | - | - | 18.25 |
| Anole | - | - | - | - | 36.64 | 36.64 | 62.35 | 21.24 | 41.80 | 60.23 | 21.75 | 41.00 | - | - | - | - | - | - | - | 19.91 |
| GILL | - | - | - | - | 50.67 | 50.67 | 54.15 | 17.27 | 35.71 | 67.75 | 25.44 | 46.60 | - | - | - | - | - | - | - | 22.16 |
| SEED-LLaMA | - | - | - | - | 57.00 | 57.00 | 67.12 | 17.39 | 42.26 | 60.57 | 23.34 | 41.96 | - | - | - | - | - | - | - | 23.54 |
| Bagel | - | - | - | - | 59.91 | 59.91 | 71.33 | 19.58 | 45.46 | 66.58 | 22.44 | 44.51 | - | - | - | - | - | - | - | 24.98 |
| RecA | - | - | - | - | 60.97 | 60.97 | 72.88 | 20.85 | 46.87 | 68.94 | 23.66 | 46.30 | - | - | - | - | - | - | - | 27.36 |
| GPT-4o-Image | - | - | - | - | 65.65 | 65.65 | 71.97 | 21.18 | 46.58 | 86.61 | 33.52 | 60.07 | - | - | - | - | - | - | - | 28.72 |
| Gemini-2.0-flash-exp | - | - | - | - | 77.61 | 77.61 | 67.77 | 19.30 | 43.54 | 84.59 | 30.53 | 57.56 | - | - | - | - | - | - | - | 29.79 |
| Gemini-2.5-flash-image | - | - | - | - | 85.32 | 85.32 | 79.24 | 26.56 | 52.90 | 92.47 | 40.11 | 66.29 | - | - | - | - | - | - | - | 34.00 |
| MIO-Instruct | 59.93 | 70.38 | 23.41 | 51.24 | 59.29 | 59.29 | 68.12 | 19.20 | 43.66 | 72.69 | 23.77 | 48.23 | 60.03 | 69.22 | 26.40 | 51.88 | 64.08 | 68.66 | 66.37 | 53.45 |

Table 9: **Comparison of multimodal models on various generation tasks. CIVG:** Conditional Image-to-Video Generation; **FIR:** Fine-grained Image Reconstruction; **TIE:** Text-Guided Image Editing; **TIG:** Text-to-Image Generation; **TVG:** Text-to-Video Generation; **VP:** Video Prediction. * denotes MLLMs with the ability to generate interleaved images and texts, while '-' indicates that the model does not have the ability to achieve the corresponding task and underlined content signifies the best performance within a single model across all methods on this task.

# F   ANALYSIS AND FINDINGS

**Trade-off Between Basic and Unified Capabilities.** The experimental results reveal that current U-MLLMs face a significant challenge in balancing their fundamental abilities—such as understanding and generating performance—with the demands of unified tasks that require integrating multiple modalities. For instance, models like GILL, Anole, and MiniGPT-5 are designed to handle unified tasks but tend to exhibit relatively poor performance on basic tasks, which results in lower overall scores when compared to some non-unified MLLMs. On the other hand, while MIO-Instruct demonstrates high performance in basic understanding and generation, its capability to interleave image and text generation effectively is notably deficient. This imbalance suggests that the current training paradigms may not be adequately aligning the learning objectives for basic and unified capabilities within a single framework.

**Detailed Analysis of Model Performance on Unify Tasks.** In Table 3, we provide a detailed analysis of different models' performance on unify tasks, focusing on text reasoning accuracy and image generation accuracy. It is clear that MIO-Instruct exhibits stronger understanding capabilities

than generation abilities (as confirmed by the results in Table 2). As a result, many of its tasks show high text reasoning performance, particularly in commonsense QA, where its text reasoning accuracy reaches 76.24%. However, it fails to generate a correct image, completely missing the potential for mutual reinforcement between generation and understanding. In contrast, other models show comparable performance in both text reasoning and image reasoning evaluation criteria, but their overall results are not impressive. Notably, for visual CoT tasks, despite our efforts to simplify the questions into multiple-choice format, none of the models have been able to correctly complete multi-step reasoning and generation tasks.

**Poor Instruction Following Ability for Image Generation.** There are two main issues with the current models in image generation: *1. Uncontrolled Style Generation.* In Figure 11, we present the intermediate state images generated by different models in the VCoT task. Only the Anole and Gemini2.0-flash-exp models are able to generate images with a style similar to the initial image. In contrast, other models produce images with a clear style bias, which do not align well with our state diagrams. *2. Difficulty Understanding Complex Instructions.* Many models, such as MIO-Instruct, struggle with following complex instructions, such as generating auxiliary lines based on the original question. These models fail to generate images with auxiliary lines, often requiring multiple attempts to generate a relevant image, and the resulting images often bear little resemblance to the original reference. However, for simpler instructions, like generating an image of a dog, these models are able to execute the task correctly.

**Inadequate Visual CoT Capability in Unified Models.** In Figure 4, we further illustrate the challenges of the Visual CoT task. The accuracy of U-MLLMs declines as the number of steps in the VCoT task increases. Errors made in earlier steps compound over time, making it increasingly difficult for models to generate correct actions, coordinates, and images. This cascading error effect highlights a fundamental limitation in maintaining consistent reasoning across multi-step tasks. At the same time, this example further emphasizes the high requirements of our unify tasks for both generation and understanding capabilities. For instance, although Anole demonstrates relatively strong image accuracy in Figure 4, its weaker understanding abilities result in less effective action selection. This ultimately leads to worse final results compared to the other two baselines.

To further enhance the evaluation value of VCoT task, we additionally provide two complementary analyses: (i) difficulty stratification by maze size, and (ii) step-wise metrics across the full reasoning trajectory for further analysis of the unify capability of current U-MLLMs.

**Difficulty stratification by maze size.** We first stratify VCoT performance by maze size and report average accuracy for each model under different maze configurations (Table 10). This allows us to examine how U-MLLMs behave as the underlying navigation problem becomes more complex.

| Model | 3×3 | | | 4×4 | | | 5×5 | | |
|---|---|---|---|---|---|---|---|---|---|
| | action | coord | image | action | coord | image | action | coord | image |
| Bagel | 21.14 | 23.58 | 21.14 | 19.70 | 23.48 | 19.51 | 12.00 | 18.00 | 14.67 |
| RecA | 28.46 | 28.46 | 20.33 | 19.70 | 28.03 | 20.45 | 14.67 | 19.33 | 16.67 |
| GPT-4o-Image | 21.95 | 18.70 | 25.20 | 20.45 | 17.42 | 20.45 | 19.33 | 10.00 | 16.00 |
| Gemini 2.5-Flash-Image | 43.09 | 29.27 | 31.71 | 38.64 | 20.45 | 23.48 | 31.33 | 12.00 | 18.00 |

Table 10: **VCoT accuracy stratified by maze size.** We report average action, coordinate, and image accuracy (%) on 3×3, 4×4, and 5×5 mazes. For example, Gemini 2.5-Flash-Image achieves 43.09% / 29.27% / 31.71% action/coord/image accuracy on 3×3 mazes, which drops to 31.33% / 12.00% / 18.00% on 5×5 mazes. Other models exhibit the same downward trend, providing an overall perspective on difficulty stratification in VCoT.

From Table 10, we observe that Gemini2.5-Flash-Image's action/coord/image accuracy on the 3×3 maze is 43.09%, 29.27%, and 31.71%, respectively, while these values drop to 31.33%, 12.00%, and 18.00% on the 5×5 maze. Other models exhibit a similar monotonic decline as maze size increases. This confirms that VCoT difficulty scales systematically with problem size and that all current U-MLLMs degrade under more challenging configurations.

**Step-wise metrics and failure modes.** Concurrently, we reconstruct VCoT into a set of *step-wise* metrics and compute average accuracy for four representative U-MLLMs (Bagel, RecA, GPT-

| Model | Step-wise accuracy (%) | | | | | | |
|---|---|---|---|---|---|---|---|
| | Step 1 | Step 2 | Step 3 | Step 4 | Step 5 | Step 6 | Step 7 |
| **Action** | | | | | | | |
| Bagel | 26.67 | 20.00 | 17.78 | 11.11 | 6.67 | 2.22 | 2.22 |
| RecA | 25.56 | 25.56 | 21.11 | 11.11 | 4.44 | 3.33 | 1.11 |
| GPT-4o-Image | 28.89 | 23.33 | 23.33 | 10.00 | 4.44 | 2.22 | 0.00 |
| Gemini 2.5-Flash-Image | 52.22 | 48.89 | 37.78 | 16.67 | 10.00 | 1.11 | 1.11 |
| **Coordinate** | | | | | | | |
| Bagel | 67.78 | 20.00 | 17.78 | 11.11 | 1.11 | 0.00 | 0.00 |
| RecA | 75.56 | 25.56 | 12.22 | 2.22 | 1.11 | 1.11 | 0.00 |
| GPT-4o-Image | 38.89 | 10.00 | 5.56 | 4.44 | 4.44 | 2.22 | 2.22 |
| Gemini 2.5-Flash-Image | 23.33 | 20.00 | 17.78 | 17.78 | 10.00 | 1.11 | 0.00 |
| **Image** | | | | | | | |
| Bagel | 43.33 | 25.56 | 7.78 | 7.78 | 1.11 | 1.11 | 0.00 |
| RecA | 46.67 | 22.22 | 5.56 | 5.56 | 2.22 | 1.11 | 0.00 |
| GPT-4o-Image | 51.11 | 28.89 | 10.00 | 12.22 | 0.00 | 0.00 | 0.00 |
| Gemini 2.5-Flash-Image | 53.33 | 28.89 | 13.33 | 8.89 | 1.11 | 1.11 | 0.00 |

Table 11: **Step-wise VCoT accuracy for different prediction dimensions.** We report step-wise accuracy (%) for four representative U-MLLMs across steps 1–7. The three gray-shaded blocks correspond to action, coordinate, and image accuracy, respectively. All models start well above the random baseline at Step 1, but by Steps 5–7 most entries drop to around 1–4%, especially in the coordinate and image dimensions.

4o-Image, Gemini2.5-Flash-Image) across steps 1–7 in three dimensions: action, coordinate, and image. The results are summarized in Table 11.

From Table 11, we observe that in Step 1 the accuracies of action, location, and image for all models are significantly higher than the random baseline, and remain reasonably good in Step 2. However, starting from Step 2, the coordinate and image dimensions exhibit a near "cliff-like" drop, reaching approximately 1–4% by Steps 5–7, while action accuracy decays more gradually. This pattern indicates that although the models can still correctly identify the current position and generate a roughly correct next frame in the first 1–2 steps, the first capability lost during the multi-step reasoning process is the *consistent visual modeling of the maze state*. Once this internal visual state becomes unreliable, it quickly propagates to degrade both coordinate localization and action prediction.

Therefore, even though the sample-level success rate over complete 7-step trajectories is close to 0%, the stratified analyses by maze size and step index allow us to pinpoint two core bottlenecks of current U-MLLMs on VCoT: (i) **multi-step visual state maintenance**, and (ii) **long-range consistency**. These diagnostics provide essential guidance for targeted improvements to planning and visual memory mechanisms in future unified multimodal models.

## F.1 COMPARE WITH RANDOM AND HUMAN BASELINES

To make the MME-U Score more interpretable, we compare our unified evaluation with two complementary baselines, a *random guessing* baseline and a *human* baseline. Both baselines are evaluated under exactly the same multiple-choice protocol as U-MLLMs across all Unify sub-tasks.

**Random-Select baseline.** For the random baseline (RANDOM SELECT), we uniformly sample one option for every text or image multiple-choice question, and compute all task metrics (Text Acc, Image Acc, Acc, and Acc$^+$; for VCoT we additionally report Action/Coord/Image Acc). As shown in Table 12, the Random-Select metrics on the two high-precision / multi-step reasoning tasks, SpotDiff (SD) and VCoT, *already exceed* the performance of most current U-MLLMs. For example, on VCoT, the random baseline attains 14.24% Acc and 21.73% Image Acc, which are higher than those of Bagel, RecA, and GPT-4o-Image, and only slightly below Gemini2.5-Flash-Image. This indirectly suggests that existing U-MLLMs remain extremely weak in scenarios requiring fine-

| Method | IEE | | | | CSQ | | | | AL | | | | SD | | | | VCoT | | | | | Avg |
|---|---|---|---|---|---|---|---|---|---|---|---|---|---|---|---|---|---|---|---|---|---|---|
| | T | I | A | A$^+$ | T | I | A | A$^+$ | T | I | A | A$^+$ | T | I | A | A$^+$ | Act. | Crd. | Img. | A | A$^+$ | |
| Show-o2 | – | – | – | – | 79.21 | 53.47 | 66.34 | 56.44 | – | – | – | – | – | – | – | – | – | – | – | – | – | 13.27 |
| Random Select | 29.00 | 27.50 | 28.25 | 11.50 | 24.75 | 31.68 | 28.22 | 5.94 | 23.08 | 28.85 | 25.97 | 7.69 | 30.00 | 31.00 | 30.50 | 9.00 | 19.26 | 1.73 | 21.73 | 14.24 | 0.00 | 25.43 |
| Bagel | 22.78 | 43.89 | 33.34 | 11.67 | 90.10 | 64.36 | 77.23 | 63.37 | 42.31 | 21.15 | 31.73 | 9.62 | 27.00 | 24.00 | 25.50 | 7.00 | 10.83 | 12.08 | 10.69 | 11.20 | 0.00 | 35.80 |
| RecA | 23.39 | 47.95 | 35.67 | 10.53 | 90.10 | 68.32 | 79.21 | 59.41 | 44.23 | 23.08 | 33.66 | 13.46 | 26.00 | 27.00 | 26.50 | 9.00 | 11.53 | 14.03 | 11.11 | 12.22 | 0.00 | 37.45 |
| GPT-4o-Image | 25.50 | 60.50 | 43.00 | 17.00 | 97.03 | 76.24 | 86.64 | 74.26 | 42.31 | 42.31 | 42.31 | 15.38 | 29.00 | 21.00 | 25.00 | 9.00 | 11.53 | 8.47 | 13.06 | 11.02 | 0.00 | 41.10 |
| Gemini-2.5-Flash-Image | 32.00 | 65.50 | 48.75 | 19.00 | 98.92 | 73.12 | 86.02 | 73.12 | 67.31 | 51.92 | 59.62 | 32.69 | 27.00 | 24.00 | 25.50 | 9.00 | 20.97 | 11.25 | 13.47 | 15.23 | 0.00 | 47.02 |
| Human | 96.00 | 100.00 | 98.00 | 96.00 | 97.52 | 97.52 | 97.52 | 97.52 | 94.23 | 94.23 | 94.23 | 94.23 | 19.00 | 19.00 | 19.00 | 19.00 | 100.00 | 100.00 | 100.00 | 100.00 | 100.00 | 81.75 |

Table 12: **Calibration of Unify tasks with random and human baselines.** We report unified metrics for each Unify sub-task (IEE, CSQ, AL, SD, VCoT) and overall Avg. Columns: T = Text Acc, I = Image Acc, A = Acc, A$^+$ = joint accuracy (text & image both correct), Act. = action accuracy, Crd. = coordinate accuracy, Img. = image accuracy. RANDOM SELECT randomly chooses one option per question (non-reasoning lower bound), while HUMAN is the average of two experts (upper bound). Current U-MLLMs lie between these two baselines, often much closer to random on SD and VCoT, highlighting substantial headroom for unified multimodal reasoning.

grained difference localization or long-range state tracking, confirming that SD and VCoT are highly challenging items with strong difficulty discrimination.

In contrast, for IEE, CSQ, and AL, although some U-MLLMs' single-modality Text/Image accuracy is only modestly above the random baseline, their Acc$^+$ scores—which require *both* the text and image answers to be correct simultaneously—are substantially higher than the theoretical random upper bound of $\approx 6.25\%$ (i.e., randomly guessing one out of four text options and one out of four image options). For instance, on CSQ, Random-Select achieves only 5.94% Acc$^+$, while Bagel, RecA, GPT-4o-Image, and Gemini2.5-Flash-Image reach 63.37%, 59.41%, 74.26%, and 73.12%, respectively. This indicates that U-MLLMs have learned a non-trivial degree of cross-modal consistency and understanding–generation synergy, and that our Acc / Acc$^+$ metrics effectively separate random strategies from genuine unified capability.

**Human Baseline.** For the human baseline (HUMAN), we asked two human experts with multimodal experience to complete the Unify tasks, selecting the best-matching option for both text and image questions. We report the average score over the two annotators in Table 12. Human performance on IEE, CSQ, AL, and VCoT is substantially higher than that of all current U-MLLMs, e.g., 98.00% / 97.52% / 94.23% Acc on IEE / CSQ / AL, and 100% Acc / Acc$^+$ on VCoT, demonstrating that the human upper bound is well above existing models.

Overall, current U-MLLMs often sit closer to the random baseline on the most challenging tasks (SD, VCoT), yet substantially surpass it on Acc$^+$ for IEE/CSQ/AL, jointly shows that (i) there is still substantial room for improving unified multimodal capability, and (ii) our Unify task design and Acc / Acc$^+$ metrics offer a discriminative and effective measurement of understanding and generation synergy capability.

# G  EXTENDED EXPERIMENTAL RESULTS

## G.1  MOST U-MLLMS EXHIBIT INFERIOR GENERATION CAPABILITIES

While the methods in Table 9 show relatively small differences compared to the current state-of-the-art (SOTA) generation techniques, we found that using CLIP scores for evaluation introduces certain risks of manipulation.

In Figure 12, we present the results on the fine-grained image reconstruction task. For each model, we used a unified prompt: "Reconstruct high-fidelity images from degraded inputs, preserving fine-grained details, textures, and structural integrity with perceptual realism." It is evident that GILL, SEED-LLaMA, and MIO-Instruct effectively capture the structural details of the input images and produce noticeably clearer outputs. In particular, SEED-LLaMA and MIO-Instruct demonstrate strong performance in restoring color fidelity, while Gemini2.0-flash-exp tends to preserve the integrity of the input images. In contrast, MiniGPT-5 and Anole fail to effectively extract the necessary visual information: while MiniGPT-5 does generate an image, its output deviates significantly from the source, and Anole is unable to generate a coherent image at all.

Figure 13 displays the results for the text-guided image editing task, where the editing instruction was "Change this image into a watercolor art." Similar to the reconstruction task, SEED-LLaMA and MIO-Instruct generate images that more closely resemble the source image; however, they fall short in accurately executing the specified editing instruction. Meanwhile, GILL, MiniGPT-5, and Anole show limited capability in capturing and manipulating the requisite visual details for the transformation. Notably, Gemini2.0-flash-exp not only preserves the content of the source image effectively but also accurately implements modifications according to the editing instructions.

Figure 14 illustrates the performance gap between pure video generation models and U-MLLMs on the conditional image-to-video generation task. Using the text prompt "The man is so tired. -camera zoom in," we observe that although MIO-Instruct produces video outputs with richer visual details compared to CogVideoX, it struggles to effectively generate a coherent video sequence that adheres to the given instruction based on the initial image.

In Figure 15, the generation results of CogVideoX and MIO-Instruct in the Text-to-Video Generation task are compared. The results clearly indicate that, in terms of both instruction adherence and video consistency, MIO-Instruct significantly underperforms compared to dedicated video generation models.

Overall, while some U-MLLMs exhibit promising capabilities in capturing visual details and producing high-fidelity reconstructions, challenges remain in faithfully executing complex editing instructions and generating consistent video sequences. These findings highlight critical areas for further improvement in enhancing the generation capabilities of U-MLLM systems.

## G.2 CHALLENGES IN SIMULTANEOUSLY GENERATING HIGH-QUALITY TEXT AND IMAGES IN U-MLLMs

Figures 16, 17, 18, and 19 present the results of U-MLLMs on the Unify tasks. Notably, MIO-Instruct fails to perform any text-image generation across all Unify tasks, GILL is unable to generate multimodal outputs in the SpotDiff task, and SEED-LLaMA does not support text-image generation in the Auxiliary Lines task. Overall, these results indicate that most U-MLLMs struggle to generate images that faithfully adhere to provided instructions or reference images, and their comprehension of the instructions is often flawed.

In the Image Editing and Explanation task, for instance, MiniGPT-5 produced images that bore no relation to the source images. Additionally, the textual outputs from GILL, MiniGPT-5, and SEED-LLaMA were insufficient for accurately describing the editing objects or the instructions. Similarly, in both the Commonsense Question Answering and SpotDiff tasks, although MiniGPT-5 and SEED-LLaMA correctly answered the textual multiple-choice questions, the images they generated were clearly unrelated to the corresponding options. This further emphasizes the difficulty U-MLLMs face in maintaining consistency between textual and visual outputs.

For the Auxiliary Lines task, while Anole managed to generate images that retained some of the visual details of the source images, it failed to correctly draw the required auxiliary lines as per the instructions. GILL and MiniGPT-5, on the other hand, generated content that was completely disconnected from the original images.

These findings suggest several critical limitations in current U-MLLM systems. First, there is a notable gap in their ability to integrate and utilize multimodal cues effectively, as evidenced by the misalignment between textual instructions and visual outputs. Second, while some models can capture certain visual details, they often lack the robust reasoning required to follow complex instructions, especially in tasks demanding precise visual modifications. Finally, the decoupling between text and image generation in these systems underscores the need for further research aimed at improving cross-modal coherence and instruction fidelity.

Overall, the experimental results highlight that, despite progress in individual modalities, existing U-MLLMs have considerable challenges in simultaneously generating high-quality, coherent text and images that align with complex, multimodal instructions.

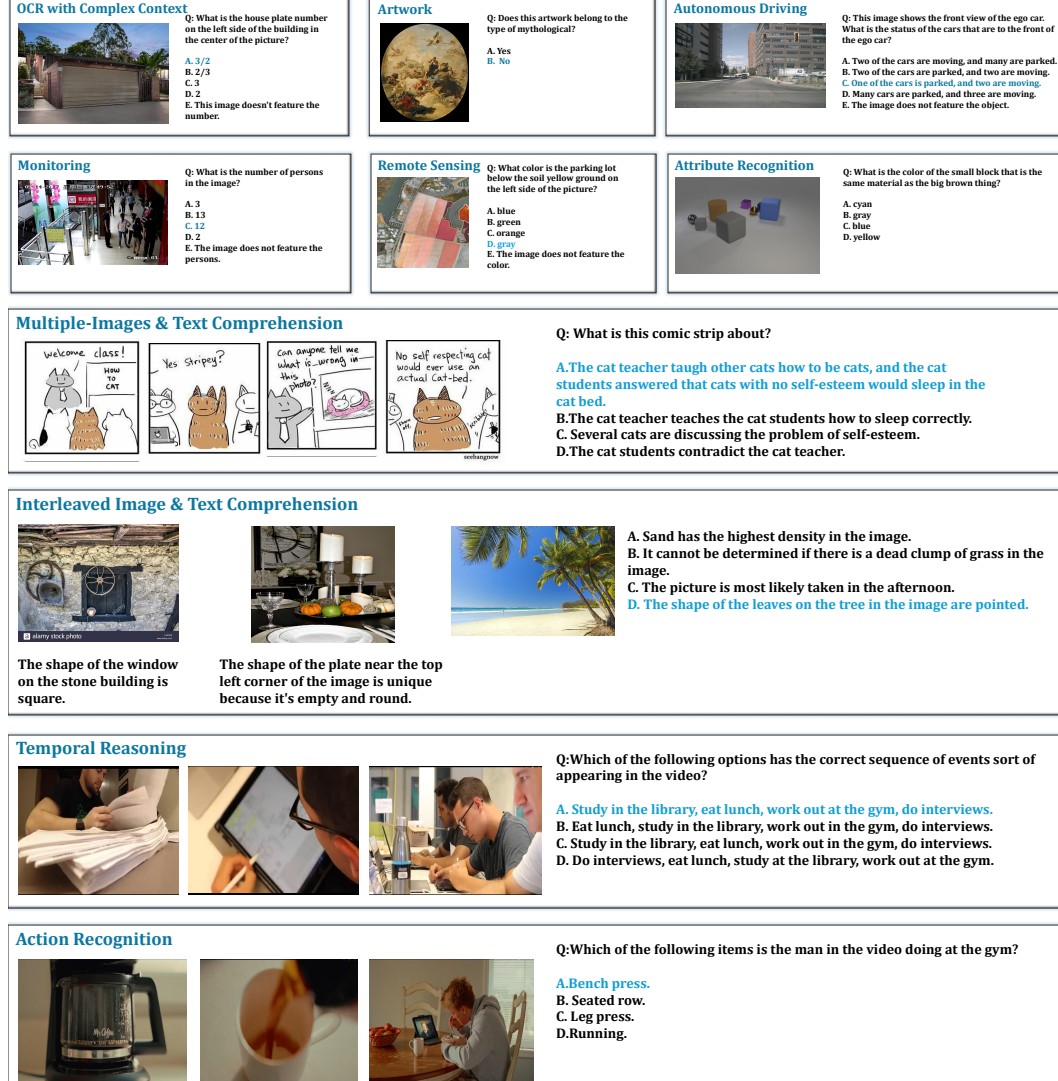

Figure 5: Data samples from understanding task, which includes single-image perception and reasoning, multi-image and image-text interlaced perception and reasoning, video perception and reasoning, etc.

## H EVALUATION METRICS

### H.1 UNDERSTANDING SCORE

Let the three subtasks in the Understanding Task be formally defined as follows:

$$T = \{\text{SIPU}, \text{MITIU}, \text{VPU}\}.$$

For each subtask $t \in T$, let $Q_t$ represent the set of multiple-choice questions, where each question $q \in Q_t$ has exactly one correct answer. To evaluate correctness, we define the indicator function for each question as follows:

$$\mathbb{I}_t(q) = \begin{cases} 1, & \text{if the selected answer for } q \text{ is correct,} \\ 0, & \text{otherwise.} \end{cases}$$

The accuracy for subtask $t$ is given by:

$$\text{acc}_t = \frac{1}{|Q_t|} \sum_{q \in Q_t} \mathbb{I}_t(q).$$

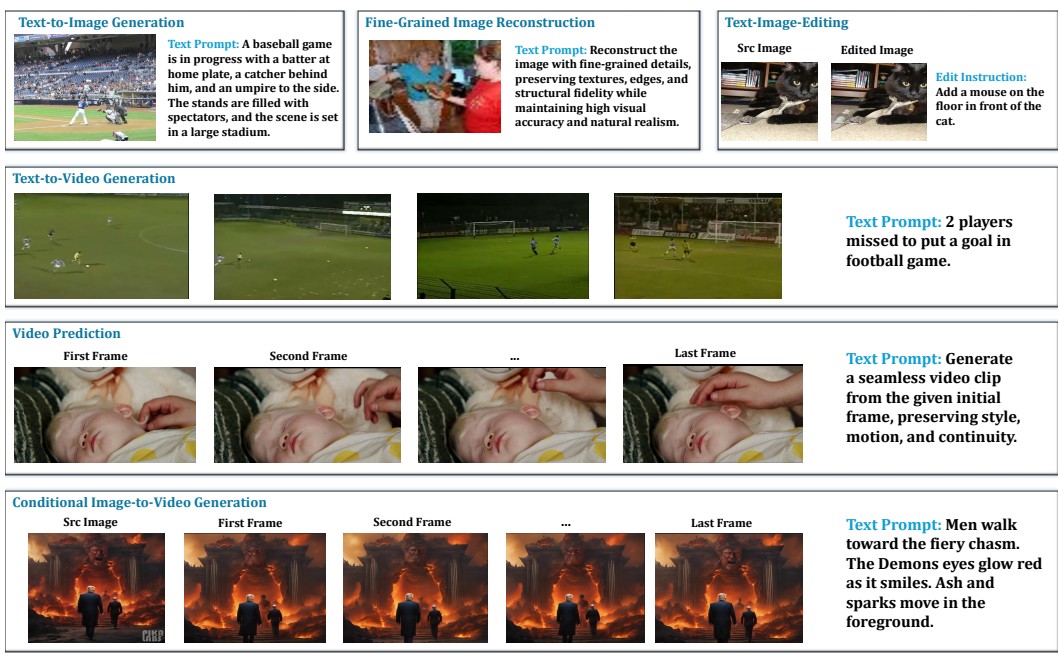

Figure 6: Data samples from generation task. It includes subtasks such as Text-to-Image Generation, Text-to-Image Editing, Fine-Grained Image Reconstruction, Text-to-Video Generation, conditional Image-to-Video Generation, and Video Prediction.

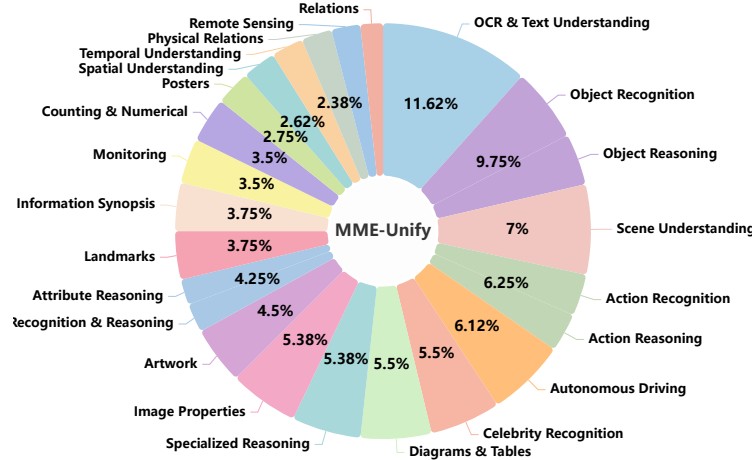

Figure 7: **An overview of real-life scenarios included in the Understanding Task.** The scores in the bars represent the proportion of the number of samples of the corresponding scenario to the total number of samples of the task.

Since equal weights are assigned to each subtask, the Understanding Score (US) is computed as the arithmetic mean of the accuracies across all subtasks:

$$\text{US} = \frac{1}{3} \sum_{t \in T} \text{score}_t, \quad T = \{\text{SIPU}, \text{MITIU}, \text{VPU}\}.$$

## H.2 GENERATION SCORE

The generative task comprises six subtasks:

$$T = \{\text{CIVG}, \text{TVG}, \text{VP}, \text{FIR}, \text{TIE}, \text{TIG}\}.$$

All metric scores are normalized to the range $[0, 100]$.

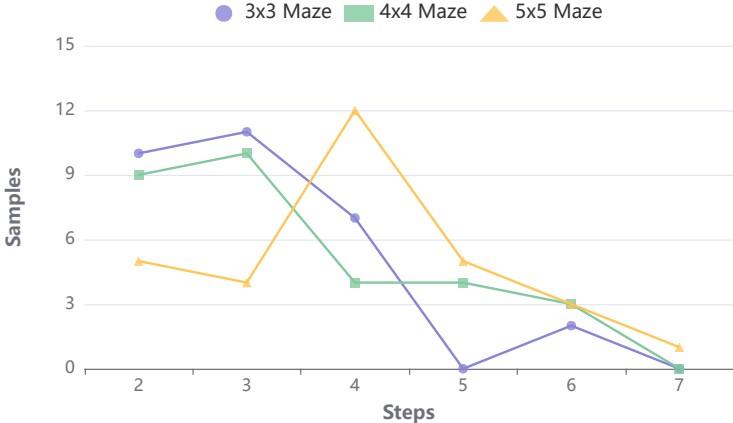

Figure 8: Distribution of steps required for samples of different mazes in the Visual CoT task.

| Task | Dataset | | | | | | | | | | | | | | | Total |
|---|---|---|---|---|---|---|---|---|---|---|---|---|---|---|---|---|
| | MME | MMBench | MME-Realworld | SEED-Bench-2 | Video-MME | Imagen Hub | Emu-Edit | TIP-I2V | COCO | Image Net | MSR-VTT | Pexel Videos | Geometry 3K | Spot Diff | Open AI | Samples |
| *Understanding Task* | | | | | | | | | | | | | | | | |
| SIPU | 400 | 400 | 400 | 0 | 0 | 0 | 0 | 0 | 0 | 0 | 0 | 0 | 0 | 0 | 0 | 1,200 |
| MITIU | 0 | 0 | 0 | 400 | 0 | 0 | 0 | 0 | 0 | 0 | 0 | 0 | 0 | 0 | 0 | 400 |
| VPU | 0 | 0 | 0 | 0 | 364 | 0 | 0 | 0 | 0 | 0 | 0 | 0 | 0 | 0 | 0 | 364 |
| *Generative Task* | | | | | | | | | | | | | | | | |
| CIVG | 0 | 0 | 0 | 0 | 0 | 0 | 0 | 200 | 0 | 0 | 0 | 0 | 0 | 0 | 0 | 200 |
| FIR | 0 | 0 | 0 | 0 | 0 | 0 | 0 | 0 | 0 | 200 | 0 | 0 | 0 | 0 | 0 | 200 |
| TIG | 0 | 0 | 0 | 0 | 0 | 0 | 0 | 0 | 200 | 0 | 0 | 0 | 0 | 0 | 0 | 200 |
| TIE | 0 | 0 | 0 | 0 | 0 | 400 | 200 | 0 | 0 | 0 | 0 | 0 | 0 | 0 | 0 | 600 |
| TVG | 0 | 0 | 0 | 0 | 0 | 0 | 0 | 0 | 0 | 0 | 200 | 0 | 0 | 0 | 0 | 200 |
| VP | 0 | 0 | 0 | 0 | 0 | 0 | 0 | 0 | 0 | 0 | 0 | 194 | 0 | 0 | 0 | 194 |
| *Unify Task* | | | | | | | | | | | | | | | | |
| IEE | 0 | 0 | 0 | 0 | 0 | 0 | 200 | 0 | 0 | 0 | 0 | 0 | 0 | 0 | 0 | 200 |
| CSQ | 0 | 0 | 0 | 0 | 0 | 0 | 0 | 0 | 0 | 0 | 0 | 0 | 0 | 0 | 100 | 101 |
| AL | 0 | 0 | 0 | 0 | 0 | 0 | 0 | 0 | 0 | 0 | 0 | 0 | 52 | 0 | 0 | 52 |
| SD | 0 | 0 | 0 | 0 | 0 | 0 | 0 | 0 | 0 | 0 | 0 | 0 | 0 | 104 | 0 | 100 |
| VCoT | 0 | 0 | 0 | 0 | 0 | 0 | 0 | 0 | 0 | 0 | 0 | 0 | 0 | 0 | 90 | 90 |
| **Dataset Total** | 400 | 400 | 400 | 400 | 364 | 400 | 400 | 200 | 200 | 200 | 200 | 194 | 52 | 104 | 190 | 4100 |
| **Dataset %** | 9.75% | 9.75% | 9.75% | 9.75% | 8.87% | 9.75% | 9.75% | 4.87% | 4.87% | 4.87% | 4.87% | 4.73% | 1.27% | 2.54% | 4.63% | 100% |

Table 13: **Task-Dataset Sampling Statistics.** This table presents the distribution of samples across different multimodal AI tasks and their source datasets. Tasks are categorized into three main groups: Understanding Tasks (SIPU: Single Image Perception and Understanding, MITIU: Multi-Image & Interleaved Text-Image Understanding, VPU: Video Perception and Understanding), Generative Tasks (CIVG: Conditional Image-to-Video Generation, FIR: Fine-grained Image Reconstruction, TIG: Text-to-Image Generation, TIE: Text-Guided Image Editing, TVG: Text-to-Video Generation, VP: Video Prediction), and Unify Tasks (IEE: Image Editing and Explanation, CSQ: Common Sense Question Answring, AL: Auxiliary Lines., SD: SpotDiff, VCoT: Visual CoT). The rightmost column shows the total number of samples used for each task across all datasets. A value of 0 indicates that no samples were drawn from that dataset for the corresponding task.

**Normalization of FVD and FID Scores.** Let $s$ denote the raw FVD or FID value for a sample, where $s \in [1, 1000]$ and lower values indicate better performance. The normalized score $S$ is computed as:

$$S = 100 \left(1 - \frac{s-1}{1000-1}\right) = 100 \left(1 - \frac{s-1}{999}\right).$$

This ensures:

- $S = 100$ when $s = 1$ (best performance),

- $S = 0$ when $s = 1000$ (worst performance).

If all raw scores across models are identical, each normalized score is set to 100 to maintain consistency in evaluation and prevent division by zero in the normalization process.

**Score Calculation for CIVG and TVG.** The subtask score for $t \in \{\text{CIVG}, \text{TVG}\}$ is given by:

$$\text{score}_t = \frac{\text{FVD}_{\text{norm}}^{(t)} + \text{FID}_{\text{norm}}^{(t)} + \text{CLIPSIM}^{(t)}}{3}.$$

**Score Calculation for VP.** The VP subtask score is determined using the following formula:

$$\text{score}_{\text{VP}} = \frac{\text{FVD}_{\text{norm}}^{(\text{VP})} + \text{FID}_{\text{norm}}^{(\text{VP})}}{2}.$$

**Score Calculation for FIR, TIE, and TIG.** For FIR (Fine-Grained Image Reconstruction), the metric is LPIPS. To ensure higher values indicate better performance, the score is defined as:

$$\text{score}_{\text{FIR}} = 1 - \text{LPIPS}.$$

For both TIE (Text Image Editing) and TIG (Text-to-Image Generation), two metrics are used: CLIP-I and CLIP-T. The score for each subtask is computed as the average of these two metrics:

$$\text{score}_{\text{TIE}} = \frac{\text{CLIP-I}_{\text{TIE}} + \text{CLIP-T}_{\text{TIE}}}{2},$$

$$\text{score}_{\text{TIG}} = \frac{\text{CLIP-I}_{\text{TIG}} + \text{CLIP-T}_{\text{TIG}}}{2}.$$

**Overall Generation Score.** The overall Generation Score (GS) is the arithmetic mean of all six subtask scores:

$$\text{GS} = \frac{1}{6} \sum_{t \in T} \text{score}_t, \quad T = \{\text{CIVG}, \text{TVG}, \text{VP}, \text{FIR}, \text{TIE}, \text{TIG}\}.$$

## H.3 UNIFY SCORE

Let the Unify Task consist of the subtasks

$$T = \{\text{IEE}, \text{CSQ}, \text{AL}, \text{SD}, \text{VCoT}\}.$$

For each subtask $t \in T$, denote by $S_t$ the set of samples.

### H.3.1 SUBTASKS IEE, CSQ, AL, SD

For a given subtask $t \in \{\text{IEE}, \text{CSQ}, \text{AL}, \text{SD}\}$ and for each sample $s \in S_t$, there are two questions:

1. A text-based multiple-choice question.
2. An image-based multiple-choice question.

Define the indicator functions for the text and image responses as follow:

$$\mathbb{I}_t^{\text{text}}(s) = \begin{cases} 1, & \text{if the text answer for } s \text{ is correct,} \\ 0, & \text{otherwise,} \end{cases}$$

$$\mathbb{I}_t^{\text{img}}(s) = \begin{cases} 1, & \text{if the image answer for } s \text{ is correct,} \\ 0, & \text{otherwise.} \end{cases}$$

Then, the text accuracy and image accuracy for subtask $t$ are, respectively,

$$\text{acc}_t^{\text{text}} = \frac{1}{|S_t|} \sum_{s \in S_t} \mathbb{I}_t^{\text{text}}(s), \quad \text{acc}_t^{\text{img}} = \frac{1}{|S_t|} \sum_{s \in S_t} \mathbb{I}_t^{\text{img}}(s).$$

The overall accuracy for subtask $t$ is then defined as the average of the two:

$$\text{acc}_t = \frac{\text{acc}_t^{\text{text}} + \text{acc}_t^{\text{img}}}{2}.$$

Additionally, we define $\text{acc}_t^+$ to represent the accuracy for samples where both the textual and image-based answers are correct:

$$\text{acc}_t^+ = \frac{1}{|S_t|} \sum_{s \in S_t} \mathbb{I}_t^{\text{text}}(s) \cdot \mathbb{I}_t^{\text{img}}(s).$$

### H.3.2 SUBTASK VCoT

For the VCoT subtask, each sample $s \in S_{\text{VCoT}}$ represents a maze navigation task composed of $K_s$ sequential steps. For each step $k \in \{1, 2, \ldots, K_s\}$, there are multiple-choice questions evaluating the model's prediction of:

1. An action.
2. A coordinate.
3. An image.

**Calculation of acc$_{\text{VCoT}}$:** Let $N_{\text{steps}} = \sum_{s \in S_{\text{VCoT}}} K_s$ be the total number of steps across all samples in the VCoT subtask. Define the indicator functions for the correctness of action, coordinate, and image predictions for step $k$ of sample $s$ as follow:

$$\mathbb{I}_{\text{VCoT}}^{\text{action}}(s, k) = \begin{cases} 1, & \text{if the action prediction for step } k \\ & \text{of sample } s \text{ is correct,} \\ 0, & \text{otherwise.} \end{cases}$$

$$\mathbb{I}_{\text{VCoT}}^{\text{coord}}(s, k) = \begin{cases} 1, & \text{if the coordinate prediction for step } k \\ & \text{of sample } s \text{ is correct,} \\ 0, & \text{otherwise.} \end{cases}$$

$$\mathbb{I}_{\text{VCoT}}^{\text{img}}(s, k) = \begin{cases} 1, & \text{if the image prediction for step } k \\ & \text{of sample } s \text{ is correct,} \\ 0, & \text{otherwise.} \end{cases}$$

Calculate the average accuracy for each prediction type across all steps:

$$\text{acc}_{\text{VCoT}}^{\text{action}} = \frac{1}{N_{\text{steps}}} \sum_{s \in S_{\text{VCoT}}} \sum_{k=1}^{K_s} \mathbb{I}_{\text{VCoT}}^{\text{action}}(s, k),$$

$$\text{acc}_{\text{VCoT}}^{\text{coord}} = \frac{1}{N_{\text{steps}}} \sum_{s \in S_{\text{VCoT}}} \sum_{k=1}^{K_s} \mathbb{I}_{\text{VCoT}}^{\text{coord}}(s, k),$$

$$\text{acc}_{\text{VCoT}}^{\text{img}} = \frac{1}{N_{\text{steps}}} \sum_{s \in S_{\text{VCoT}}} \sum_{k=1}^{K_s} \mathbb{I}_{\text{VCoT}}^{\text{img}}(s, k).$$

The overall **acc$_{\text{VCoT}}$** metric is the arithmetic mean of these three component accuracies:

$$\mathbf{acc}_{\text{VCoT}} = \frac{\text{acc}_{\text{VCoT}}^{\text{action}} + \text{acc}_{\text{VCoT}}^{\text{coord}} + \text{acc}_{\text{VCoT}}^{\text{img}}}{3}.$$

**Calculation of acc+$_{\text{VCoT}}$:** Define an indicator function for the full correctness of a single step $k$ in sample $s$:

$$\mathbb{I}_{\text{step\_all\_correct}}(s, k) = \mathbb{I}_{\text{VCoT}}^{\text{action}}(s, k) \times \mathbb{I}_{\text{VCoT}}^{\text{coord}}(s, k) \times \mathbb{I}_{\text{VCoT}}^{\text{img}}(s, k).$$

This function is 1 if all three predictions for step $k$ are correct, and 0 otherwise.

Now, define the indicator function for the perfect completion of sample $s$:

$$\mathbb{I}_{\text{VCoT}}^{\text{sample\_perfect}}(s) = \begin{cases} 1, & \text{if } \mathbb{I}_{\text{step\_all\_correct}}(s, k) = 1 \\ & \text{for all } k \in \{1, 2, \ldots, K_s\}, \\ 0, & \text{otherwise.} \end{cases}$$

The **acc+$_{\text{VCoT}}$** metric is the proportion of perfectly completed samples:

$$\mathbf{acc+}_{\text{VCoT}} = \frac{1}{|S_{\text{VCoT}}|} \sum_{s \in S_{\text{VCoT}}} \mathbb{I}_{\text{VCoT}}^{\text{sample\_perfect}}(s).$$

## H.4 UNIFY SCORES

The **Unify Score (Unify-S)** is defined as the arithmetic mean of the $\mathbf{acc}_t$ metrics across all subtasks:

$$\text{Unify-S} = \frac{1}{|T|} \sum_{t \in T} \mathbf{acc}_t,$$

## H.5 MME-U SCORE

The MME-U Score is computed as the arithmetic mean of the Understanding Score (US), Generation Score (GS), and Unify Score (Unify-S):

$$\text{MME-U} = \frac{1}{3} \left( \text{US} + \text{GS} + \text{Unify-S} \right).$$

where:

- US is the Understanding Score,
- GS is the Generation Score,
- Unify-S is the Unify Score.

Each component score is calculated as described in their respective sections.

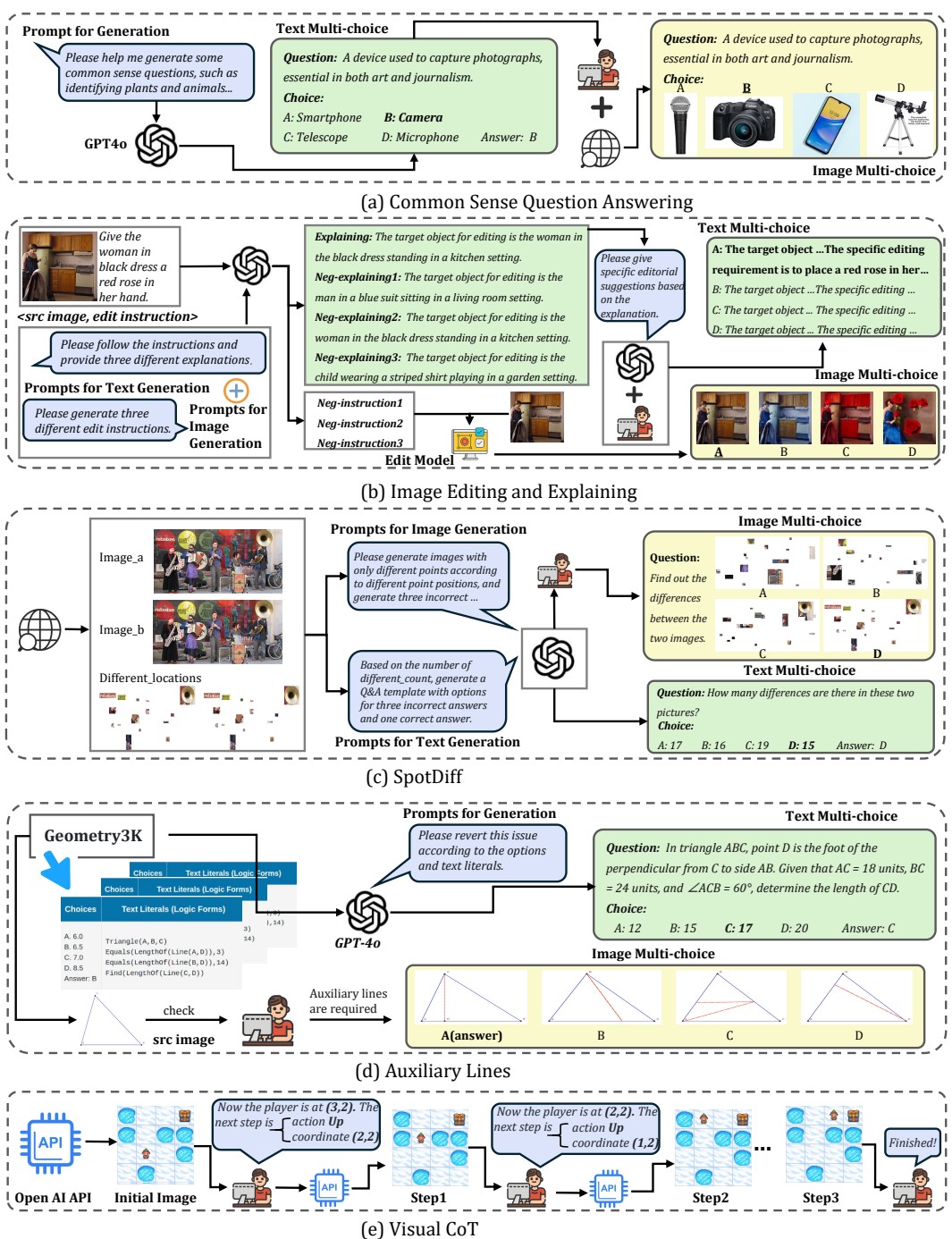

Figure 9: The overall construction process for five unified tasks, which consists of (a) Common Sense Question Answering, (b) Image Editing and Explaining, (c)SpotDiff, (d) Auxiliary Lines, and (e) Visual CoT.

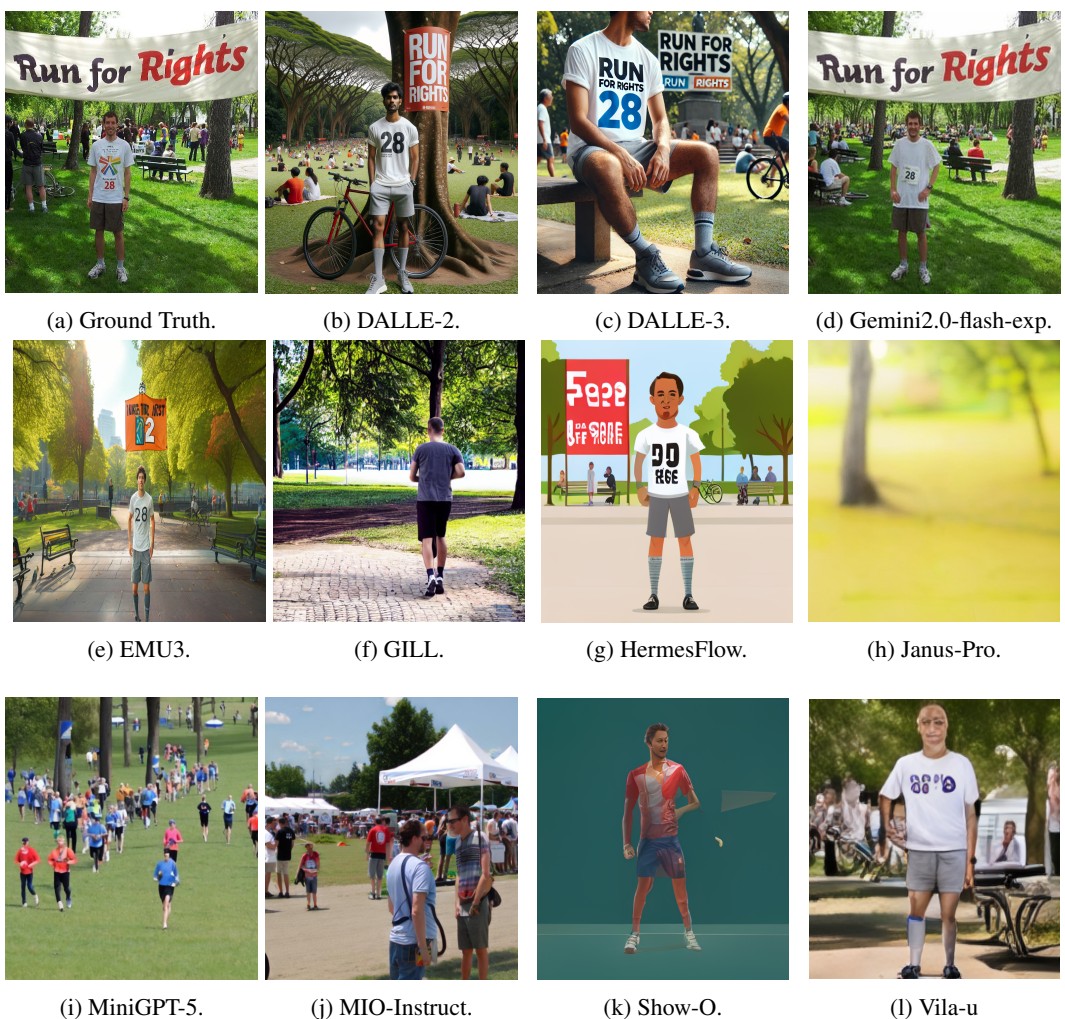

Figure 10: **The generated results from various models in the text-to-image generation task**, based on the following text prompt: *A man is standing in a park with a 'Run for Rights' banner in the background. He is wearing a white t-shirt with the number 28 on it, grey shorts, and grey socks with black shoes. The park is filled with people, some sitting on benches, and there is a bicycle leaning against a tree.*

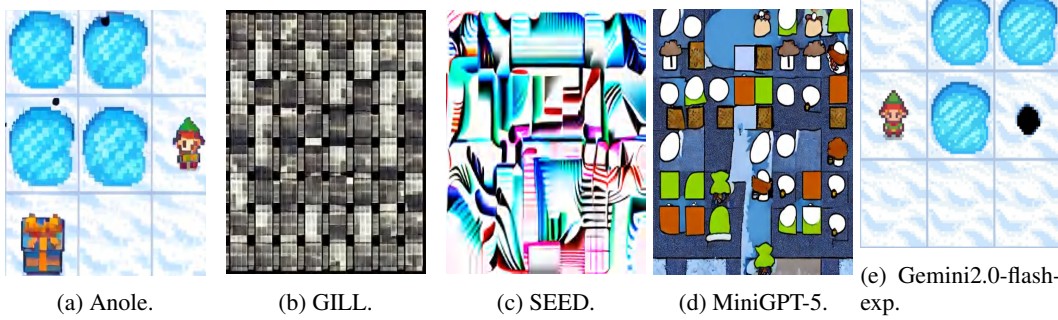

Figure 11: **Intermediate process images generated by different models in VCoT.** The figure illustrates the intermediate outputs of various models in the VCoT (Visual Composition Task), showing distinct approaches in processing and generating visual content. The models shown include (a) Anole, (b) GILL, (c) SEED, (d) MiniGPT-5, and (e) Gemini-2.0-flash-exp, each producing unique visual patterns and compositions.

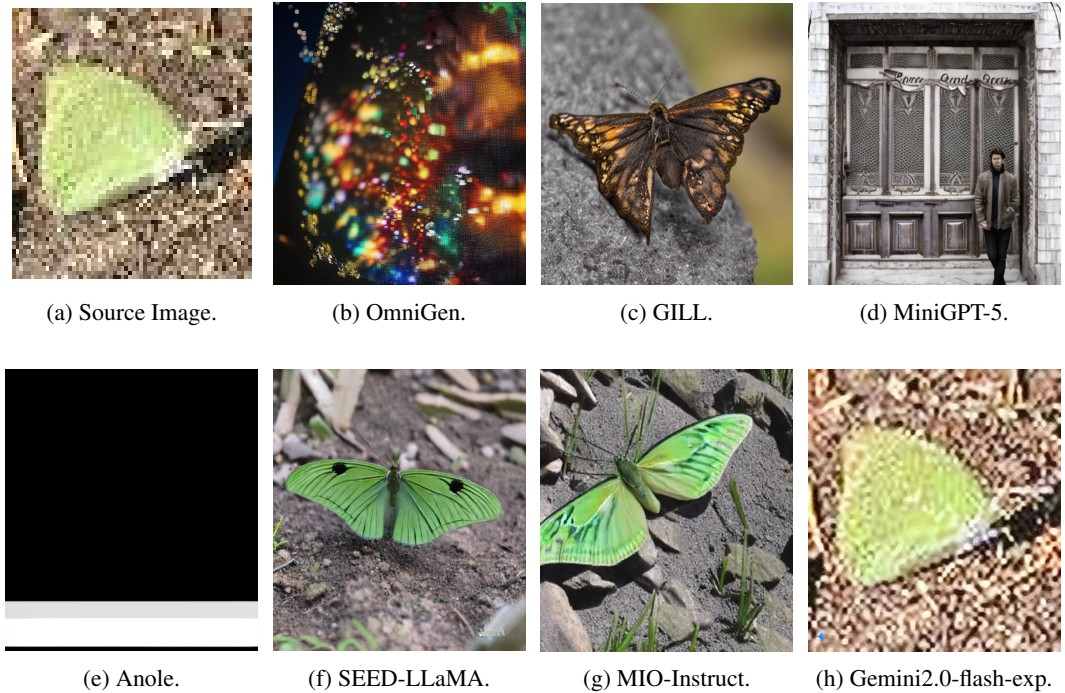

Figure 12: **The generated results from various models in the fine-grained image reconstruction task**, based on the following text prompt: *Reconstruct high-fidelity images from degraded inputs, preserving fine-grained details, textures, and structural integrity with perceptual realism.*

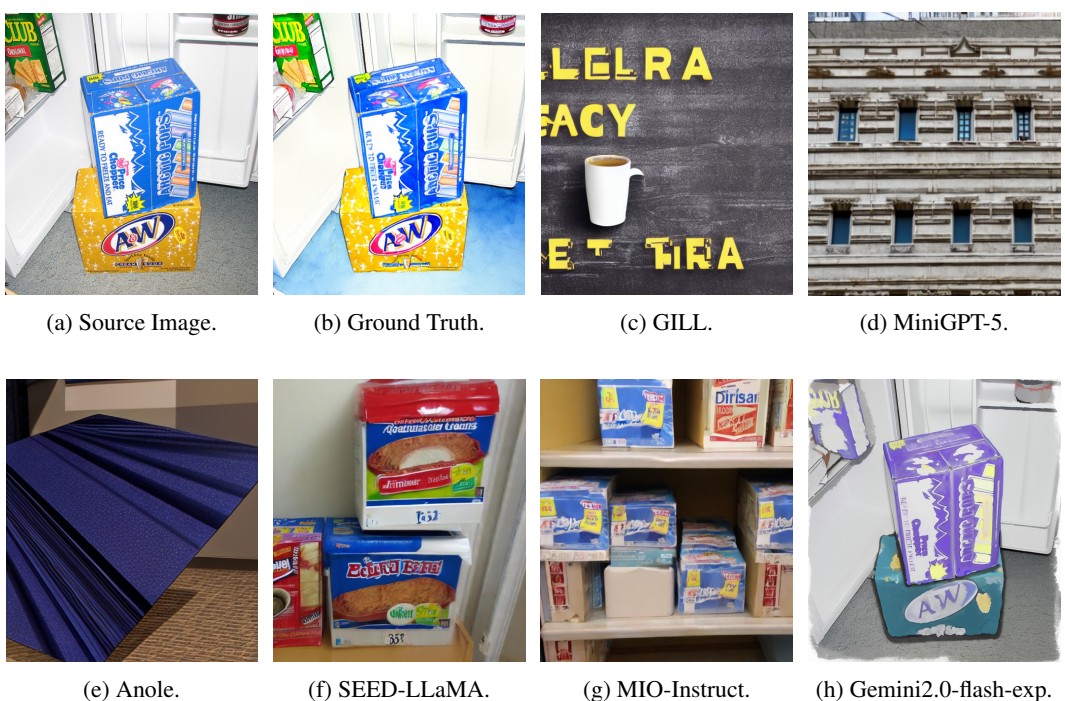

Figure 13: **The generated results from various models in the text-guided image editing task**, based on the following text prompt: *Change this image into a watercolor art.*

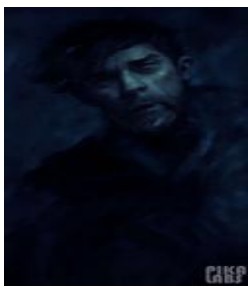

(a) Source Image.

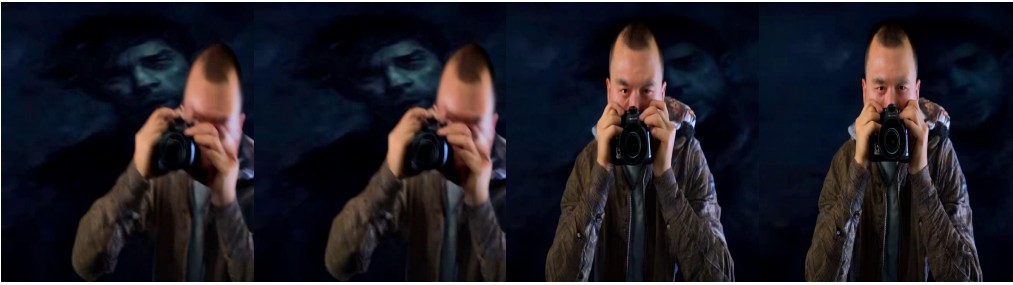

(b) CogVideoX.

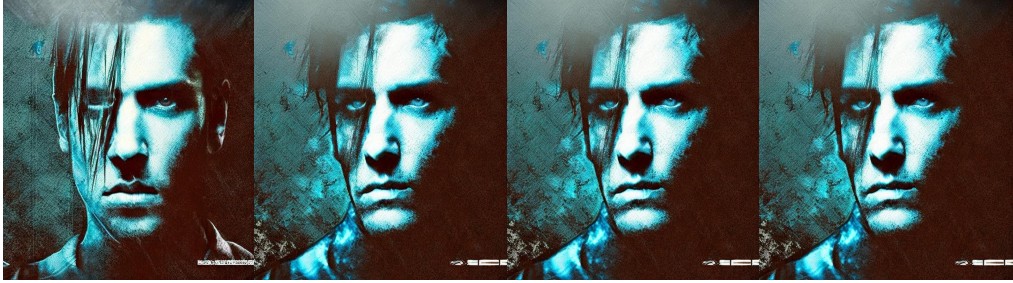

(c) MIO-Instruct.

Figure 14: **The generated results from various models in the conditional image-to-video generation task**, based on the following text prompt: *The man is so tired. -camera zoom in*.

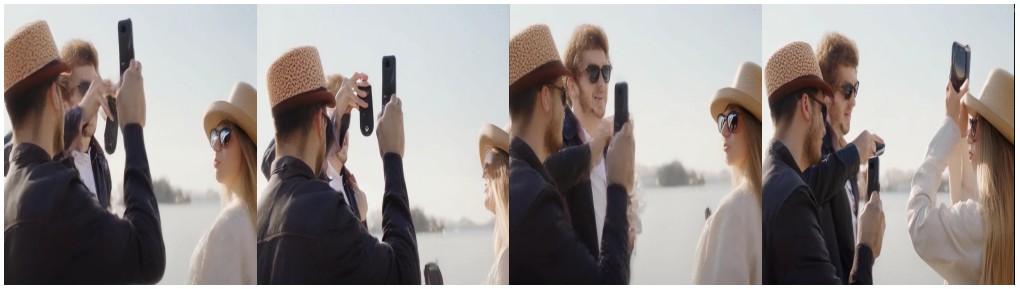

(a) CogVideoX.

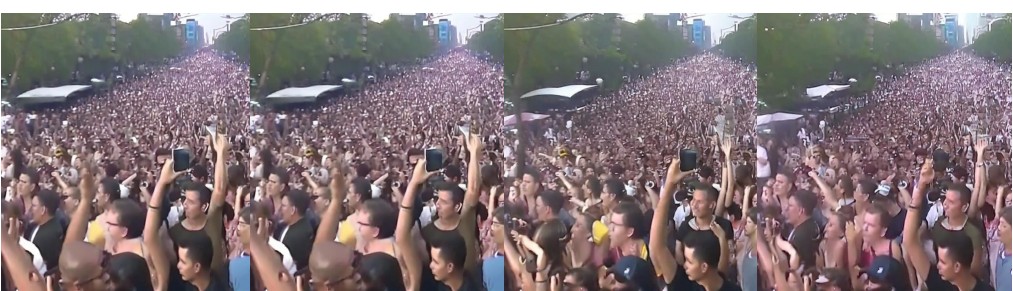

(b) MIO-Instruct.

Figure 15: **The generated results from various models in the text-to-video generation task**, based on the following text prompt: *Men wearing sunglasses and women with hats take photographs of themselves*.

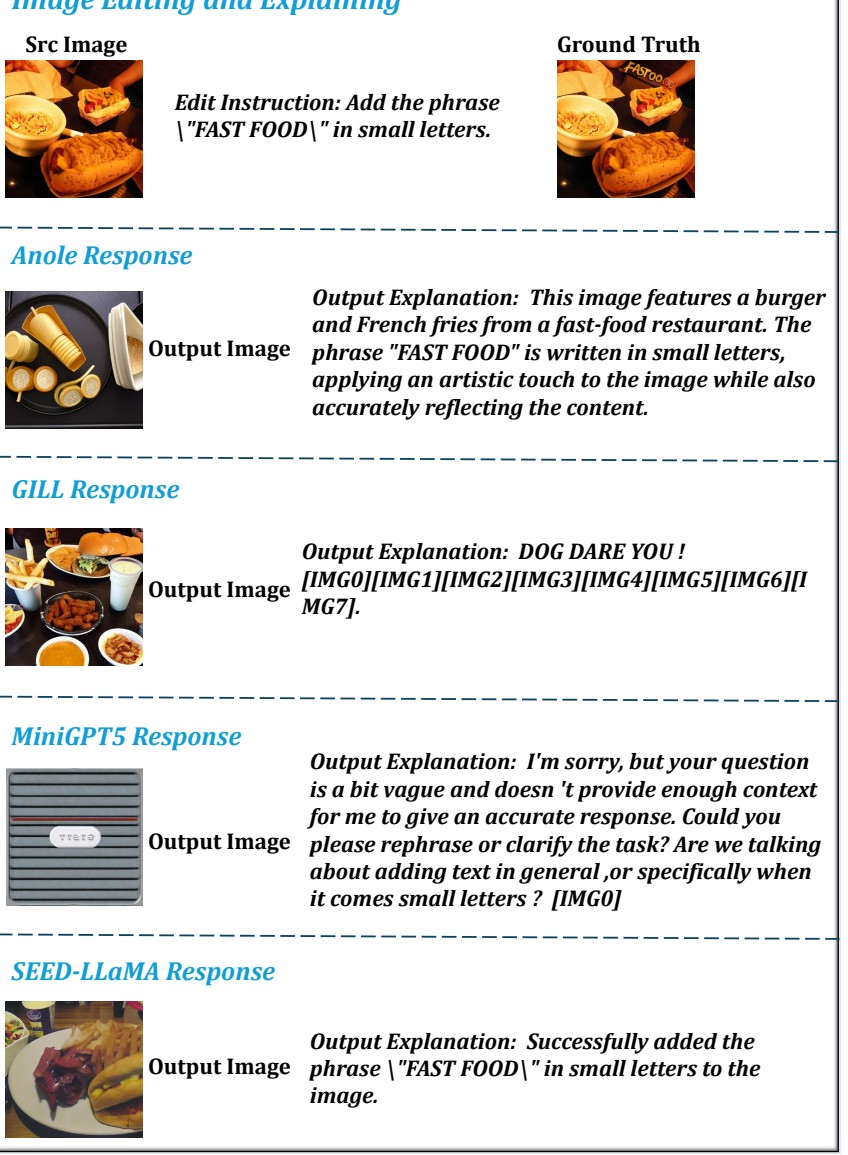

Figure 16: The generated results from various models in the image editing and explaining task.

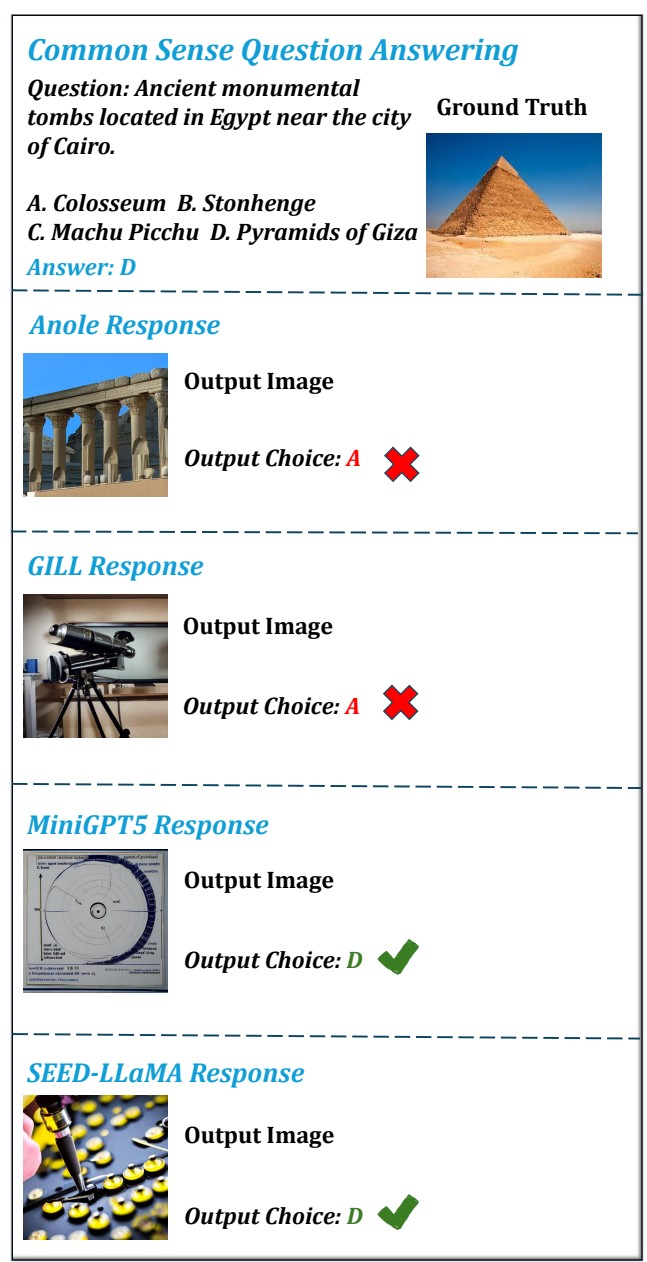

Figure 17: The generated results from various models in the common sense question answering task.

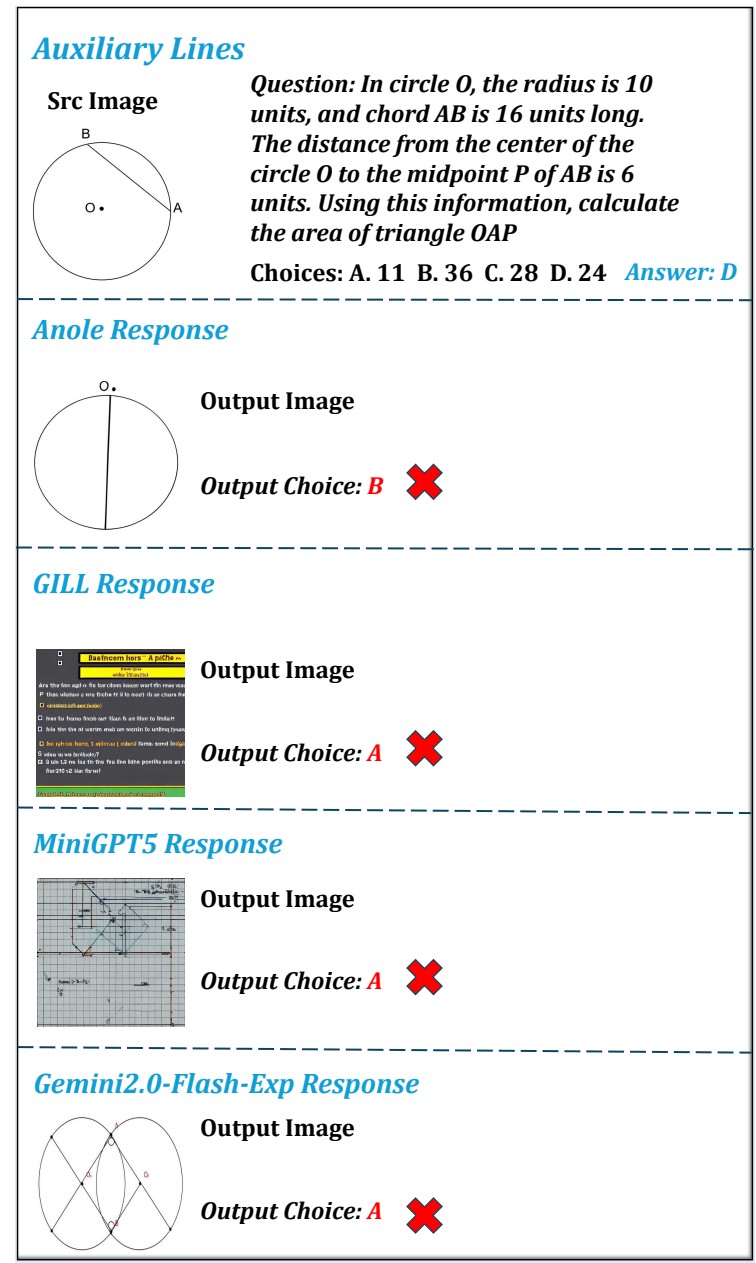

Figure 18: The generated results from various models in the auxiliary lines task.

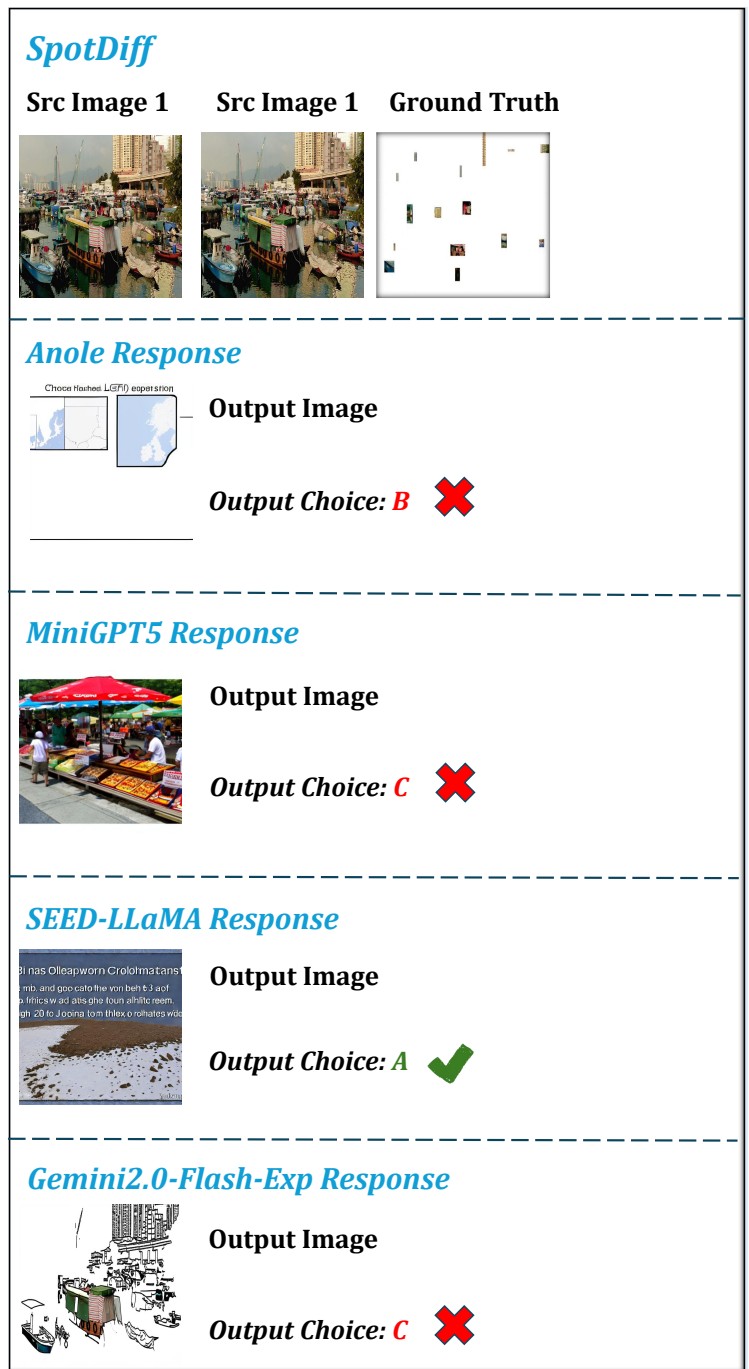

Figure 19: The generated results from various models in the spotdiff task.

# Common Sense Question Answering

**System Prompt:**
You are an AI system that answers common-sense knowledge questions by selecting the correct answer from multiple choices and then generating an image that visually represents the answer.

**Input Data**
Question: A factual question requiring knowledge-based reasoning.
Choice: A set of multiple-choice answers labeled A, B, C, and D.

**Output Requirements**

**Answer Selection:**
Analyze the question and determine the correct answer based on general knowledge.
Output the selected answer in the format: Answer: X (where X is A, B, C, or D).

**Image Generation:**
Generate an image that visually represents the content of the chosen answer.

**Processing Steps**
Understand the Question: Extract key information from the question.
Evaluate the Choices: Compare each option and determine the most accurate answer.
Select the Correct Answer: Output the correct choice in the required format.
Generate the Image: Create an image that correctly depicts the content of the selected answer.
Verify Coherence: Ensure the generated image aligns with the chosen answer.

**Example**
**Input:**
question: "Which planet is known as the Red Planet?",
choice:
  "A: Earth",
  "B: Mars",
  "C: Venus",
  "D: Jupiter"

**Model Output:**
Answer: B
<image> (Generating image of the Mars)

Figure 20: System prompt for Common Sense Question Answering task.

# Image Editing and Explaining

**System Prompt:**
You are an AI-powered image editing assistant. Your task is to modify a provided initial image based on a question instruction and generate a clear visual description of the edited object.

Input Data
Question: A natural language instruction specifying how the image should be modified.
Initial Image: The original image that needs editing.
Output Requirements
Explanation:
Identify the target object or region in the image that needs to be modified.
Provide a concise visual description of the object before and after modification.
Clearly describe how the edit integrates into the scene.
Edited Image:
Generate an image that precisely follows the question instruction while ensuring realism and coherence.
Maintain the original image's quality, lighting, and perspective in the edited version.
Processing Steps
Analyze the Question: Extract key editing instructions (e.g., add, remove, modify, change color, reposition).
Identify the Target Object: Locate the relevant object or scene element that needs modification.
Generate a Visual Description: Clearly describe the object before and after editing, ensuring it aligns with the given instruction.
Apply the Modification: Edit the image accordingly, ensuring seamless integration with existing elements.
Verify Output: Ensure the modification meets the instruction while preserving natural aesthetics.

Example
Input:
Question: Add a fork to the plate.
<image>

Model Output:
Explanation: The target object for editing is the plate containing a steak, potatoes, and mixed vegetables, with a slice of orange for garnish. The specific editing requirement is to add a fork to the plate, ensuring it complements the arrangement of the existing food items.
<edited image>

Figure 21: Systemp prompt for Image Editing and Explaining task

# SpotDiff

**System Prompt:**
You are an AI system designed to analyze two similar images (img_a and img_b) and identify the number of differences between them. Your task is to:

Compare img_a and img_b to find all differences.
Select the correct answer from the provided multiple-choice options.
Extract the different regions from img_a and place them on a white background of the same size.
Input Format
img_a: The first image.
img_b: The second image (similar but not identical to img_a).
choice: Multiple-choice answers indicating different counts of differences, labeled as A, B, C, D.

Example Input:
img_a: "<image_a>",
img_b: "<image_b>",
choice:
  "A: 14",
  "B: 11",
  "C: 19",
  "D: 10"

Output Format
Answer Selection:
Identify the correct number of differences and output the answer in the format:
Answer: X  (where X is A, B, C, or D)
Extracted Difference Image:
Identify regions in img_a that differ from img_b.
Extract these differing regions and place them on a white background of the same size as img_a.
The final image should highlight only the different areas while preserving their original details.

Example Output:
Answer: B
<image> (Extracted difference regions placed on a white background)

Processing Steps
Compare img_a and img_b to identify all differences (object position, shape, color, missing parts, etc.).
Count the total number of differences and match it to the correct multiple-choice answer.
Extract differing regions from img_a and overlay them on a white background of the same size.
Output the selected answer and the processed image.

Key Requirements
Strictly select one answer from A, B, C, D.
Ensure extracted differences are accurately placed on a clean white background.
Maintain the original structure of differing regions (no modifications, just extraction).

Figure 22: System prompt for SpotDiff task.

# Auxiliary Lines

**System Prompt:**
You are an AI system designed to solve junior high school geometry problems. Your task is to:

Analyze the given geometry question, image, and multiple-choice answers.
Draw auxiliary lines on the geometric diagram to assist in problem-solving.
Determine the correct answer based on the problem's conditions.
**Input Data**
Question: A geometry-related word problem describing angles, lengths, or relationships.
Image: A geometric diagram corresponding to the problem statement.
Choice: A set of multiple-choice answers labeled A, B, C, and D.
**Output Requirements**
**Answer Selection:**
Use geometric reasoning to determine the correct answer.
Output the selected answer in the format: Answer: X (where X is A, B, C, or D).
**Image with Auxiliary Lines:**
Draw necessary auxiliary lines (such as perpendiculars, bisectors, or diagonals) on the geometric diagram to facilitate solving.
Ensure the lines are clear and logically placed according to the problem's constraints.
Maintain the original structure of the diagram while highlighting the new construction.
**Processing Steps**
Understand the Problem: Analyze given conditions (parallel lines, angles, lengths, etc.).
Identify Key Geometric Properties: Determine the relationships between elements in the diagram.
Draw Auxiliary Lines: Add necessary constructions to simplify calculations.
Solve for the Answer: Apply geometric theorems and algebraic calculations.
Output Answer and Edited Image: Provide the correct answer and the diagram with auxiliary lines.

**Example**
**Input:**
question: "Given the quadrilateral ABCD, where line segment AB is parallel to line segment DC, the measure of ∠ABC is 60°, and the measure of ∠ADC is 45°. Additionally, the length of BC is 8 units, and the length of AB is 24 units. Determine the perimeter of quadrilateral ABCD.",
choice:
  "A: 26 + 2 \\sqrt { 3 } + 2 \\sqrt { 6 }",
  "B: 26 + 4 \\sqrt { 3 } + 4 \\sqrt { 6 }",
  "C: 52 + 2 \\sqrt { 3 } + 2 \\sqrt { 6 }",
  "D: 52 + 4 \\sqrt { 3 } + 4 \\sqrt { 6 }"

**<image>**(geometry diagram)

**Model Output:**
**Answer: B**
**<image>** (image with auxiliary lines)

Figure 23: System prompt for Auxiliary Lines task.

# Visual CoT

**System Prompt for first step:**

You are given a grid-based puzzle game map where each grid square can either be a safe square (land) or a hole. Your goal is to reach the target while avoiding the holes and using as few moves as possible. You can move in four directions: Left, Right, Up, or Down. The grid is 3×3.The top-left cell is (0,0), the top-right cell is (2,0), the bottom-left cell is (0,2), and so forth.Rows increase downward, and columns increase to the right.

**Game Settings:**
- The grid map is fully observable.
- The player starts at a designated grid square.
- The goal is located elsewhere on the map.
- Each grid square is either safe (land) or contains a hole (non-safe).
- The player must avoid holes, and moving into a hole results in failure.
- The objective is to guide the player to the goal without falling into holes.

**Movement Rules:**
- The player can move left, right, up, or down to an adjacent square, provided it is a safe square.
- The player cannot move more than one square at a time.
- Moving outside the edge of the map has no effect. The player stays in the same position.
- Do not fall into holes.
- The player wins by reaching the goal.

**Your task:**
- Based on the current state of the game, decide the next move for the player.
- Provide the next action: "Left", "Right", "Up", or "Down".
- After selecting the action, specify the coordinates of the player's new location as [x, y].
- Also, output a representation of the grid map after the selected action.

**Output Format:**
Action: [Your move choice]
Location: [x, y]
Image: [Generated Image]

Here is the Initial grid map:
(Shown Initial Figure)

Please choose the next move and give output:

Figure 24: Systemp prompt for Visual CoT task in the first step.

# Visual CoT

**System Prompt after first step:**

You are given a grid-based puzzle game map where each grid square can be either a safe square (land) or a hole. Your goal is to reach the goal while avoiding holes and using as few moves as possible. You can move in four directions: left, right, up, or down. The grid is 3×3. The top-left cell is (0,0), the top-right cell is (2,0), the bottom-left cell is (0,2), and so on. Rows increase downwards and columns increase rightwards.

**Game Setup:**
- The grid map is fully observable.
- The player starts at a designated grid square.
- The goal is somewhere else on the map.
- Each grid square is either safe (land) or contains a hole (non-safe).
- The player must avoid holes, and entering a hole will result in failure.
- The goal is to guide the player to the goal without falling into a hole.

**Movement Rules:**
- The player can move left, right, up, or down to an adjacent square, provided it is a safe square.
- The player cannot move more than one block at a time.
- Moving beyond the edge of the map has no effect. The player remains in the same position.
- Do not fall into a hole.
- The player wins by reaching the goal.

**Your Task:**
- Determine the next move for the player based on the initial grid map, the history information, and the current state of the game.
- Provide the next action: "Left", "Right", "Up", or "Down", and output "Finish" if you think the goal position has been reached
- After selecting an action, specify the coordinates of the player's new position as [x, y].
- Also, output a representation of the grid map after the selected action.

Please provide the action, coordinates and the maze image of the player's new position for next step

This is the initial grid map:
(Showing Initial Map)

Here is the state of the game after last step:
**History Information:**
- Last action (e.g., "Go Right", "Go Down", etc.).
- Current position.
- An image of the grid after the last move.
- Initial grid map:

**Output format:**
Action: [your move selection]
Location: [x, y]
Image: [generated image]

Please select the next step and give the output:

Figure 25: Systemp prompt for Visual CoT task after first step.

