# OpenReview forum: "MME-Unify: A Comprehensive Benchmark for Unified Multimodal Understanding and Generation Models"
_ICLR.cc/2026/Conference — ICLR 2026 Poster_

### Official Review · Reviewer_QMnN · 2025-10-19

**Soundness:** 2
**Presentation:** 3
**Contribution:** 2
**Rating:** 4
**Confidence:** 5

**Summary:**

This paper proposes MME-Unify, a new benchmark to evaluate Unified Multimodal Large Language Models. It solves the problem that existing studies lack a unified way to test MLLMs’ understanding, generation, and mixed-modality abilities. MME-Unify uses data from 12 datasets, unifies task formats (like multiple-choice questions) and metrics, and designs 5 mixed-modality “unify tasks”. The authors tested 12 MLLMs, finding Gemini2.0-flash-exp performs best, but all models still struggle with balance between understanding and generation, and mixed-modality tasks.

**Strengths:**

1. This paper is generally well-written and easy to follow.
2. This benchmarks evaluates the comprehensive abilities of unifed multimodal models, including multimodal understanding, generation, and mixed-modality integration.
3. The 5 designed “unify tasks” (e.g., drawing auxiliary lines for geometry problems) effectively test how unifed multimodal models combine understanding and generation, targeting their most unique feature.

**Weaknesses:**

1. My biggest concern about this paper is the way the paper evaluates image generation—using CLIP score and multiple-choice questions. CLIP score only checks overall semantic similarity, not whether the generated image truly follows the prompt. SEED-Bench (a 2023 work) adopted this metric for evaluation of unified mutlimodal models, and by 2025, there should be more suitable metrics.
2. When using multiple-choice questions for image generation evaluation, the paper does not check if the generated image’s details match the prompt—only if it is similar to the option images. This means a model might "pass" the test without actually following the task’s requirements.
3. The paper does not evaluate newer models like Gemini-2.5-Pro or Bagel, so we cannot know how these new models perform on the MME-Unify benchmark.

**Questions:**

A minor issue: The layout of Figure 3 needs to be adjusted. The images in this figure are too small, making it difficult to clearly see the details.

---

> ### Author Response · Authors · 2025-11-21
> **Response to W1-(1/2)**
>
> Thank you for your constructive review and valuable suggestions! Below, we provide a detailed response to your questions and comments. If any of our responses fail to sufficiently address your concerns, please inform us, and we will promptly follow up.
>
> **W1: Concerns about CLIP-based Metric**
>
> A1: We thank the reviewer for their careful comments. We also recognize that relying solely on CLIP scores and multiple-choice questions is not the endpoint for "image quality assessment." However, it must be emphasized that the design objective of MME-U’s unified tasks is fundamentally different from that of SEED-Bench. While SEED-Bench focuses more on evaluating the model's interleaved generation capability given a text prompt, our unified tasks prioritize the understanding-generation synergy: "correctly comprehending/reasoning first, and then generating the crucial intermediate visual state based on that understanding." For example, in AL (Auxiliary Line drawing), we only retain geometry problems that "require drawing a specific auxiliary line to solve." If the model does not truly understand the problem and reason out which auxiliary line should be added, it is difficult to generate an image aligned with the correct option. In VCoT (Visual Chain-of-Thought), the model must first infer the correct next action and coordinates before generating the corresponding maze state image. Simply "drawing a random picture" based solely on the original text instruction cannot significantly improve the score. Therefore, a high score on the unify tasks does not correlate to "who is better at drawing a high-CLIP image according to the prompt," but rather to "who is better able to clearly understand complex instructions, complete the reasoning chain, and consistently express this process visually." Under this objective, we adopt the **CLIP-I-based multiple-choice format, which primarily serves to map these understanding-driven generation results into a unified discrete decision space, using $\text{acc} / \text{acc}^{+}$ to characterize this synergistic capability, rather than treating CLIP-I as a single image quality metric.**
>
> We thank the reviewer for the constructive suggestion regarding the adoption of stronger evaluation metrics. We have started to introduce of new evaluation methods, which using GPT-4o as an LLM-Judge, we scored and ranked the outputs of the unified tasks for several U-MLLMs (e.g., Bagel[1], RecA[2], GPT-4o-Image[3], Gemini2.5-Flash-Image[4]) on a scale of 1-5 across three dimensions—Text-Following, Image Quality, and Reference Similarity, and took the average as the final score.
>
> The comparative results indicate that, with the exception of local rank reversals between specific models (e.g., Bagel and RecA) on the extremely challenging VCoT sub-task, the rankings for the remaining unified sub-tasks and the overall model rankings remain substantially consistent with the results based on CLIP-I / $\text{acc}^{+}$. This demonstrates that our current CLIP-I evaluation strategy is well-aligned with the stronger LLM-Judge metrics at the model level.
>
> ## CLIP-based Evaluation for image QA
> | Model                | IEE   | CSQ   | AL    | SD    | VCoT  | Avg   |
> |----------------------|-------|-------|-------|-------|-------|-------|
> | Show-o2              | -     | 53.47 | -     | -     | -     | 10.69 |
> | Bagel                | 43.89 | 64.36 | 21.15 | 24.00 | 10.69 | 32.82 |
> | RecA                 | 47.95 | 68.32 | 23.08 | 27.00 | 11.11 | 35.49 |
> | GPT-4o-Image         | 60.50 | 76.24 | 42.31 | 21.00 | 13.06 | 42.62 |
> | Gemini 2.5-Flash-Image | 65.50 | 73.12 | 51.92 | 24.00 | 13.47 | 45.60 |
>
> ## LLM-Judge-based Evaluation for image QA
>
> | Model                 | IEE   | CSQ   | AL    | SD    | VCoT  | Avg   |
> |-----------------------|-------|-------|-------|-------|-------|-------|
> | Show-o2               | -     | 56.33 | -     | -     | -     | 11.27 |
> | Bagel                 | 49.47 | 66.93 | 37.07 | 47.67 | 52.33 | 50.69 |
> | RecA                  | 49.60 | 68.07 | 43.73 | 48.67 | 52.00 | 52.41 |
> | GPT-4o-Image          | 68.87 | 78.80 | 58.60 | 45.47 | 52.60 | 60.87 |
> | Gemini 2.5-Flash-Image| 80.33 | 72.33 | 65.40 | 53.07 | 58.67 | 65.96 |
>
> ---
>
> [1] Deng, et al. "Emerging Properties in Unified Multimodal Pretraining." arXiv preprint arXiv:2505.14683 (2025).
>
> [2] https://openai.com/zh-Hans-CN/index/introducing-4o-image-generation/
>
> [3] Xie, Ji, et al. "Reconstruction Alignment Improves Unified Multimodal Models." arXiv preprint arXiv:2509.07295 (2025).
>
> [4] https://deepmind.google/models/gemini/flash/

---

> > ### Author Response · Authors · 2025-11-21
> > **Response to W1-(2/2)**
> >
> > Furthermore, we randomly sampled 200 instances from the unified tasks. We asked three experts to score the generation results of same U-MLLMs (The evaluation method is consistent with LLM-Judge). Subsequently, we calculated the Kendall's $\tau_b$ correlation coefficient between the CLIP-I scores of the model-generated images and the human ratings for each task (as shown in the table below),
> >
> > | | IEE | CSQ | AL | SD | VCoT | Overall |
> > | :--- | :--- | :--- | :--- | :--- | :--- | :--- |
> > | **Kendall’s $\tau$** | 0.711982 | 0.73212 | 0.71725 | 0.707045 | 0.69195 | 0.70871 |
> >
> > The results show that the correlation generally falls within the range of approximately 0.69-0.73 across various sub-tasks, with an overall average correlation of about 0.71. This falls within the range of strong correlation, demonstrating that under our task design, there is relatively stable consistency between CLIP-I and human perceptual ratings.
> >
> > The reason for still selecting the CLIP-I-based approach for the main evaluation track is **primarily due to considerations of scale and reproducibility**: MME-U involves thousands of image generation samples. If LLM-Judge were adopted across the full dataset, the evaluation cost (API call fees) would be extremely high, and it would impose a significant burden on subsequent research attempting to reproduce experiments on this benchmark. Therefore, we adopt the CLIP-I-based unified evaluation strategy for the main scoring track, and will supplement the appendix with consistency analyses based on LLM-Judge and human evaluation to verify its reliability. In the revised manuscript, we will include the full LLM-Judge results and visual comparisons in the appendix, and in subsequent versions, **we plan to introduce more fine-grained automatic metrics (such as object-level editing consistency, geometric constraints, etc.) as a supplementary evaluation strategy**, in response to the reviewer’s valuable suggestion for "more suitable evaluation metrics."

---

> > > ### Author Response · Authors · 2025-11-21
> > > **Response to W2**
> > >
> > > **W2: Concerns about a model might "pass" the test without actually following the task’s requirements.**
> > >
> > > A2: We are appreciate for the reviewer's valuable comments. This point is indeed related to the original intention behind our unified task design. The unified task in MME-U is not examining "whether the model can directly generate an image based on a single text prompt," but rather examining "whether the model first completes the correct understanding/reasoning and then generates the necessary intermediate visual representation accordingly."
> > >
> > > Specifically, when constructing unified tasks like AL (Auxiliary Line) and VCoT (Visual Chain-of-Thought), **we designed the data and evaluation strictly based on the premise that a visual intermediate step is mandatory**:
> > >
> > > In AL, we filtered out all geometry problems that could be solved without an auxiliary line, retaining only samples where "a specific auxiliary line must be drawn first to solve the problem." When facing such samples, if the model merely treats the text as a regular T2I prompt to "draw an image" without considering "which auxiliary line should be drawn to derive the geometric relationship," it will be difficult to select the correct answer later.
> > >
> > > Similarly, in VCoT, the model needs to first generate the current maze state image at each step and then decide the next action and coordinates based on it. The image itself is part of the reasoning chain, not a decorative output loosely corresponding to the text.
> > >
> > > Therefore, under our unified task setting, image generation is driven by the model's own understanding and multi-step reasoning, rather than simply "generating a high-CLIP image randomly based on a text prompt."
> > >
> > > We have confirmed the consistency of our unified task's evaluation strategy with the manual evaluation and LLM-Judge evaluation strategies (refer to Response to W1). Considering the cost and stability of human evaluation and LLM-Judge, we ultimately decided to adopt the evaluation strategy based on CLIP-I and multiple-choice questions.
> > >
> > > We will supplement more visual comparisons and analyses of U-MLLM on the unified tasks in the appendix to demonstrate this point more intuitively. Thank you again for the reviewer's valuable comments.

---

> > > > ### Author Response · Authors · 2025-11-21
> > > > **Response to W3-(1/2)**
> > > >
> > > > **W3: Concerns about evaluate newer models like Gemini-2.5-Pro or Bagel**
> > > >
> > > > A3: We thank the reviewer for the reminder. The initial lack of coverage for some newly released models in the first draft was primarily due to time constraints. In the revised manuscript, we have addressed this by supplementing the following experiments, and the results have been incorporated into the relevant tables (as shown in table under response to W3-(2/2)):
> > > >
> > > > * (i) For the **Understanding tasks**, we newly evaluated **Gemini-2.5-Pro[1]**;
> > > > * (ii) For the **Generation tasks**, we added **Qwen-Image[2]** and **Qwen-Image-Edit[3]**;
> > > > * (iii) For the complete **MME-Unify** benchmark, we systematically evaluated five of the latest U-MLLMs: **Bagel[4], RecA[5], Show-o-2[6], GPT-4o-Image[7], and Gemini-2.5-Flash-Image[8]**.
> > > >
> > > > Overall, the closed-source model Gemini-2.5-Flash-Image leads in the total MME-U score, while among open-source models, RecA and Bagel show the best comprehensive performance. Notably, RecA's Unify-Score has significantly improved compared to previous open-source U-MLLMs, now approaching that of Gemini-2.0-Flash-Image and surpassing the closed-source model GPT-4o-Image. It is worth noting that, **in contrast to the typical trade-off between understanding and generation capabilities often seen in earlier U-MLLMs, models like Bagel and RecA have achieved significant progress in the Unify tasks while simultaneously improving their Understanding and Generation scores** (as shown in table under response to W3-(2/2)). This accurately reflects the "understanding-generation synergy" emphasized by MME-Unify.
> > > >
> > > > We will supplement Tables 2, 3, and 6 in the main body with the latest results, and include an analysis of the performance of new models such as Bagel and RecA in Section 3 and Appendix E. This aims to provide readers with a more comprehensive understanding of the unified capability of the latest U-MLLMs under MME-Unify.
> > > >
> > > >
> > > >
> > > > | Model |        Understanding         |              |             |             |            Generation             |        |        |        |        |        |        |        Unify        |        |        |        |        |        | MME-U Score |
> > > > |-------|------------------------------|--------------|-------------|-------------|------------------------------------|--------|--------|--------|--------|--------|--------|----------------------|--------|--------|--------|--------|--------|--------------|
> > > > |       | SIPU | MITPU | VPU | Avg     | CIVG | VP   | TVG | TIG | TIE | FIR | Avg   | IEE  | CSQ  | AL   | SD   | VCoT | Avg   | Avg          |
> > > > | Gemini2.5-Pro       | 87.00 | 69.00 | 66.21 | 74.07 | - | - | - | - | - | - | - | - | - | - | - | - | - | 24.69 |
> > > > | Qwen-Image          | - | - | - | - | - | - | - | 72.43 | - | - | 12.07 | - | - | - | - | - | - | 4.02 |
> > > > | Qwen-Image-Edit        | - | - | - | - | - | - | - | - | 58.81 | 88.86 | 73.84 | - | - | - | - | - | - | 24.62 |
> > > > | Show-o2       | 68.33 | 47.00 | 50.00 | 55.11 | - | - | - | 50.18 | - | - | 8.36 | - | 66.34 | - | - | - | 13.27 | 23.94 |
> > > > | Bagel          | 76.67 | 53.00 | 51.10 | 60.26 | - | - | - | 44.51 | 45.46 | 59.91 | 24.98 | 33.34 | 77.23 | 31.73 | 25.50 | 11.20 | 35.80 | 40.35 |
> > > > | GPT-4o-Image      | 65.50 | 49.50 | 45.05 | 53.35 | - | - | - | 60.07 | 46.58 | 65.65 | 28.72 | 43.00 | 86.64 | 42.31 | 22.50 | 11.02 | 41.10 | 41.06 |
> > > > | RecA        | 76.00 | 57.00 | 56.04 | 63.01 | - | - | - | 46.30 | 46.87 | 72.88 | 27.36 | 35.67 | 79.21 | 33.66 | 26.50 | 12.22 | 37.45 | 42.60 |
> > > > | Gemini-2.5-Flash-Image | 80.25 | 70.75 | 58.79 | 69.93 | - | - | - | 66.29 | 52.90 | 85.32 | 34.09 | 48.75 | 86.02 | 59.62 | 25.50 | 15.23 | 47.02 | 50.04 |
> > > >
> > > >
> > > >
> > > > ---
> > > > [1] https://deepmind.google/models/gemini/
> > > >
> > > > [2] Wu, et al. "Qwen-Image Technical Report." arXiv preprint arXiv:2508.02324 (2025).
> > > >
> > > > [3] https://huggingface.co/Qwen/Qwen-Image-Edit
> > > >
> > > > [4] Deng, et al. "Emerging Properties in Unified Multimodal Pretraining." arXiv preprint arXiv:2505.14683 (2025).
> > > >
> > > > [5] Xie, et al. "Reconstruction Alignment Improves Unified Multimodal Models." arXiv preprint arXiv:2509.07295 (2025).
> > > >
> > > > [6] Xie, et al. "Show-o2: Improved Native Unified Multimodal Models." arXiv preprint arXiv:2506.15564 (2025).
> > > >
> > > > [7] https://openai.com/zh-Hans-CN/index/introducing-4o-image-generation/
> > > >
> > > > [8] https://deepmind.google/models/gemini/flash/

---

> > > > > ### Author Response · Authors · 2025-11-21
> > > > > **Response to W3-(2/2)**
> > > > >
> > > > > | Method        | IEE |  |  |  | CSQ |  |  |  | AL |  |  |  | SD |  |  |  | VCoT |  |  |  |  | Avg |
> > > > > |---------------|----------|-----------|---------|---------|----------|-----------|---------|---------|----------|-----------|---------|---------|----------|-----------|---------|---------|------------------|---------------|--------------|-------------|---------|------|
> > > > > | Metric        | Text Acc | Image Acc | Acc | Acc+ | Text Acc | Image Acc | Acc | Acc+ | Text Acc | Image Acc | Acc | Acc+ | Text Acc | Image Acc | Acc | Acc+ | Action Acc | Coord Acc | Image Acc | Acc | Acc+ | Avg |
> > > > > | Show-o2         | -        | -         | -   | -    | 79.21        | 53.47         | 66.34   | 56.44    | -        | -         | -   | -    | -        | -         | -   | -    | -           | -          | -          | -   | -    | 13.27   |
> > > > > | Bagel    | 22.78       | 43.89         | 33.34   | 11.67    | 90.10        | 64.36         | 77.23   | 63.37    | 42.31        | 21.15         | 31.73   | 9.62    | 27.00        | 24.00         | 25.50   | 7.00    | 10.83           | 12.08          | 10.69          | 11.20   | 0    | 35.80   |
> > > > > | RecA  | 23.39        | 47.95         | 35.67   | 10.53    | 90.10        | 68.32         | 79.21   | 59.41    | 44.23        | 23.08         | 33.66   | 13.46    | 26.00        | 27.00         | 26.50   | 9.00    | 11.53           | 14.03          | 11.11          | 12.22   | 0    | 37.45   |
> > > > > | GPT-4o-Image    | 25.50        | 60.50         | 43.00   | 17.00    | 97.03        | 76.24         | 86.64   | 74.26    | 42.31        | 42.31         | 42.31   | 15.38    | 29.00        | 21.00         | 25.00   | 9.00    | 11.53           | 8.47          | 13.06          | 11.02   | 0    | 41.10   |
> > > > > | Gemini-2.5-Flash-Image  | 32.00        | 65.50         | 48.75   | 19.00    | 98.92        | 73.12         | 86.02   | 73.12    | 67.31        | 51.92         | 59.62   | 32.69    | 27.00        | 24.00         | 25.50   | 9.00    | 20.97           | 11.25          | 13.47          | 15.23   | 0    | 47.02   |

---

> > > > > > ### Author Response · Authors · 2025-11-21
> > > > > > **Response to Q1**
> > > > > >
> > > > > > **Q1: The layout of Figure 3 needs to be adjusted**
> > > > > >
> > > > > > A1: We thank the reviewer for the reminder. We will adjust the layout of Figure 3 in the revised manuscript to allow readers to view the benchmark details more clearly. We thank the reviewer again for pointing out this issue.

---

> ### Author Response · Authors · 2025-11-26
> **Kindly Request for Reviewer's Feedback**
>
> Dear Reviewer,
>
> Thank you so much for your time in improving our paper!
>
> Since the end of the rebuttal is coming soon, may we know if our response addresses your main concerns? Should you have any further advice, please let us know and we will be more than happy to engage in more discussion and improvements.

---

### Official Review · Reviewer_VRFA · 2025-10-30

**Soundness:** 3
**Presentation:** 3
**Contribution:** 3
**Rating:** 4
**Confidence:** 4

**Summary:**

The paper introduces MME-Unify (MME-U), an open and reproducible benchmark targeting Unified Multimodal LLMs (U-MLLMs). It evaluates three capability axes: multimodal understanding, multimodal generation, and “unify” (mixed-modality generation that couples understanding with generation). To enable cross-task comparability, the authors convert understanding tasks into multiple-choice QA and standardize diverse generation metrics onto a common (0,100) scale. A key contribution is five unify subtasks—Image Editing & Explaining, Common-Sense QA + image generation, Auxiliary Lines for geometry, Spot-the-Difference, and Visual Chain-of-Thought (maze navigation)—intended to measure how understanding and generation reinforce each other. The benchmark aggregates 12 datasets into 30 subtasks and evaluates 22 models (open/closed, understanding/generation specialists, and U-MLLMs). Results show that even the strongest systems struggle on unify tasks and that instruction following and visual detail alignment remain open problems.

**Strengths:**

**Clear problem motivation**: The benchmark directly targets the unique mixed-modality generation capability that prior benchmarks do not quantify. The unify tasks are not a simple concatenation of existing tasks but require genuine coupling of understanding and generation.

**Breadth and coverage**: The suite spans single/multi-image and video inputs, plus text-to-image/video, editing, image-to-video, and prediction on the generation side. Table 1 convincingly positions MME-U as broader than existing benchmarks.

**Transparent pipeline and reproducibility**: The attribute unification pipeline, domain-wise metrics with standardized scoring, rule-based output matching, option randomization, and model capability adapters (e.g., key-frame sampling for video) make the evaluation reasonably clear and replicable.

**Weaknesses:**

**Evaluation robustness (core concern)**: The scoring relies heavily on CLIP-based similarities and hand-crafted negative samples for both generation and unify tasks. This risks score hacking via feature-space proximity or style artifacts, and may not reflect human judgments on aesthetics, editing faithfulness, or geometric correctness. While the paper acknowledges this limitation, the current version lacks human evaluation or adversarial analyses to quantify the bias or alignment with human preferences.

**Unify task share vs. weighting**: Unify subtasks comprise 546 QA items out of 4,104 total (≈13%) but constitute one of the three equally weighted top-level scores (≈33% of the final MME-U score). This mismatch can increase variance and reduce reliability, especially given the small per-subtask sizes (e.g., AL=52, VCoT=90). The paper does not report confidence intervals or stability analyses to mitigate this concern.

**Diagnostic resolution on challenging unify tasks**: The authors provide stepwise accuracies for VCoT (action/coordinate/image), which is helpful, yet overall success is near zero. Without difficulty stratification (maze sizes/horizons) or milestone scoring, it remains hard to pinpoint where reasoning breaks down and to guide targeted improvement.

**Missing baselines for calibration**: No random-guess baselines (e.g., 25% for 4-way MCQ), simple heuristic baselines, or human upper bounds are reported. This makes it harder to interpret absolute scores.

**Questions:**

**On the Trustworthiness of Scores: How do you prevent "benchmark hacking"? ** Your evaluation relies heavily on proxy metrics like CLIP scores and handcrafted negative sample matching.

What evidence can you provide that a high score on MME-U strongly correlates with the generation of high-quality, high-fidelity outputs that strictly follow complex instructions?

Have you considered incorporating evaluation dimensions that are harder to "hack," such as human preference scores or more robust automated metrics (e.g., object-level editing consistency checks), to anchor your benchmark in real-world performance?

**On the Diagnosability of Failures: How do your results help us locate and fix problems?**  A benchmark should not just assign a score; it should provide a diagnosis. Currently, a 0% success rate on a task like VCoT tells us that "all models fail," but it offers little insight into where they fail in the process, limiting its value for model developers seeking actionable feedback. Could you introduce milestone-based scoring for multi-step tasks like VCoT, reporting metrics such as "path recognition accuracy for step 1" and "visualization quality for step 1"?

---

> ### Author Response · Authors · 2025-11-21
> **Response to W1-(1/2)**
>
> We sincerely appreciate your constructive comments. We have strengthened and supplemented the relevant contents based on your suggestions.
>
> **W1: Concerns about evaluation robustness**
>
> A1: We appreciate the reviewer’s valuable feedback regarding evaluation robustness. This offers important direction for the continued development of our benchmark. We adopted the CLIP-I based image multiple-choice evaluation strategy primarily for the unified assessment of the capability of understanding–generation synergy of U-MLLMs:
>
> * **Unified Representation and Metric Design:** Our objective is to quantify the unified synergy in U-MLLMs, rather than evaluating generation and understanding in isolation. While continuous metrics like LLM-Judge provide interpretable assessments of image quality, they incur high computational costs and struggle to directly map synergistic capabilities to a discrete "success" metric. We therefore struck a balance in our evaluation strategy: by aligning image evaluation with the text multiple-choice format, we map the model's synergistic performance to discrete accuracy scores, allowing for intuitive and unified benchmarking.
>
> * **Task Requirement for Prior Understanding and Reasoning Prevents Score Hacking Risk:** The tasks require the model to first understand before generating correctly, thus avoiding potential score hacking risks: Since the unify tasks demand the model to comprehend and reason based on the input instructions before generating an image aligned with the correct answer. For instance, AL necessitates understanding the geometric problem to draw the correct auxiliary line, while VCoT requires inferring the next action to generate the correct state. Since the correct visual output depends on specific reasoning logic, generic images generated solely from input instructions, or feature-space proximity and style artifacts cannot match the correct option. This dependency minimizes the risk of exploiting CLIP-I for high scores.
>
> To quantify the consistency with human evaluation, we randomly sampled 200 instances from the unify tasks. We invited three experts to rate the generation results of four advanced U-MLLMs (Bagel[1], GPT-4o-Image[2], RecA[3], Gemini2.5-Flash-Image[4]) on a scale of 1-5 across three dimensions: Text-Following (degree of compliance with instruction/geometric constraints), Image Quality (overall aesthetics and clarity), and Reference Similarity (structural consistency with the target editing result or reference image), using the average as the final score. Subsequently, we calculated the Kendall's $\tau_b$ correlation coefficients between the CLIP-I scores of the model-generated images and the human ratings across various tasks, as shown in Table below:
>
> | | IEE | CSQ | AL | SD | VCoT | Overall |
> | :--- | :--- | :--- | :--- | :--- | :--- | :--- |
> | **Kendall’s $\tau$** | 0.711982 | 0.73212 | 0.71725 | 0.707045 | 0.69195 | 0.70871 |
>
> The results show that the correlation generally falls within the range of approximately 0.69-0.73 across various sub-tasks, with an overall average correlation of about 0.71 This falls within the range of strong correlation, demonstrating that under our task design, there is relatively stable consistency between CLIP-I and human perceptual ratings.
>
> ---
>
> [1] Deng, et al. "Emerging Properties in Unified Multimodal Pretraining." arXiv preprint arXiv:2505.14683 (2025).
>
> [2] https://openai.com/zh-Hans-CN/index/introducing-4o-image-generation/
>
> [3] Xie, Ji, et al. "Reconstruction Alignment Improves Unified Multimodal Models." arXiv preprint arXiv:2509.07295 (2025).
>
> [4] https://deepmind.google/models/gemini/flash/

---

> > ### Author Response · Authors · 2025-11-21
> > **Response to W1-(2/2)**
> >
> > Secondly, on the complete dataset, we used both CLIP-I and LLM-Judge (where the LLM-Judge scoring method is consistent with the Human Rating method) to score and rank the five newly evaluated U-MLLMs on the unified tasks. The resulting model ranking is shown in the table below. While Bagel and RecA showed comparable performance on the VCoT task, resulting in a ranking inversion on this specific task, all models otherwise demonstrated strong consistency across other tasks and in the overall ranking.
> >
> > ## CLIP-based Evaluation for image QA
> > | Model                | IEE   | CSQ   | AL    | SD    | VCoT  | Avg   |
> > |----------------------|-------|-------|-------|-------|-------|-------|
> > | Show-o2              | -     | 53.47 | -     | -     | -     | 10.69 |
> > | Bagel                | 43.89 | 64.36 | 21.15 | 24.00 | 10.69 | 32.82 |
> > | RecA                 | 47.95 | 68.32 | 23.08 | 27.00 | 11.11 | 35.49 |
> > | GPT-4o-Image         | 60.50 | 76.24 | 42.31 | 21.00 | 13.06 | 42.62 |
> > | Gemini 2.5-Flash-Image | 65.50 | 73.12 | 51.92 | 24.00 | 13.47 | 45.60 |
> >
> >
> > ## LLM-Judge-based Evaluation for image QA
> >
> > | Model                 | IEE   | CSQ   | AL    | SD    | VCoT  | Avg   |
> > |-----------------------|-------|-------|-------|-------|-------|-------|
> > | Show-o2               | -     | 56.33 | -     | -     | -     | 11.27 |
> > | Bagel                 | 49.47 | 66.93 | 37.07 | 47.67 | 52.33 | 50.69 |
> > | RecA                  | 49.60 | 68.07 | 43.73 | 48.67 | 52.00 | 52.41 |
> > | GPT-4o-Image          | 68.87 | 78.80 | 58.60 | 45.47 | 52.60 | 60.87 |
> > | Gemini 2.5-Flash-Image| 80.33 | 72.33 | 65.40 | 53.07 | 58.67 | 65.96 |
> >
> > Concurrently, we will further supplement the appendix with a visual comparison providing CLIP-I, LLM-Judge, and human ratings simultaneously for the same batch of samples, showcasing the scores and rankings of these three scoring methods at the sample level, while also analyzing typical cases where they are highly consistent and where they diverge. This aims to more intuitively present the advantages and limitations of the current CLIP-I approach. We thank the reviewer for their valuable suggestions.

---

> ### Author Response · Authors · 2025-11-21
> **Response to W2**
>
> A2: We sincerely appreciate the reviewer's reminder. First, the equal weighting of the three parts: "Understanding," "Generation," and "Unify" in MME-Unify is not based simply on the number of samples, but rather on the evaluation objective: The unify task delineates the core capability that distinguishes U-MLLMs from traditional understanding and pure generative models, the mutual promotion of understanding and generation within a single framework. Consequently, assigning a 1/3 weight in the total score is a deliberate design choice intended to encourage the community to focus on this dimension.
>
> Second, while the Unify part appears to contain only 546 samples, its effective evaluation quantity is substantial when measured by the actual number of question-answer pairs: In the four sub-tasks (IEE, CSQ, SD, and AL), each sample simultaneously yields both a text multiple-choice question and an image multiple-choice question, which contains approximately 900 QA pairs in total. VCoT, on the other hand, comprises 90 mazes with an average of approximately 3.5 steps. With 3 decisions per step (action / Coordinate / image), this totals around 945 step-level multiple-choice questions. Integrating the Understanding and Unified parts, MME-U collectively contains over 3,800 multiple-choice questions. **The Unified part contributes approximately 1/2 of the total discrete decisions,  almost equal to Understanding task, which fundamentally matches its 1/3 weighting at the functional level.**
>
> Finally, concerning scoring robustness, we have already reported the split-half experiment for the full benchmark in Appendix C, which shows that the overall MME-U ranking of the models remains consistent after random splitting. In the revised version, we will further conduct separate split-half stability analyses for the five Unify sub-tasks (see table below), which demonstrates that at the current scale, the Unify task's contribution to model ranking is stable, and not dominated by random fluctuations from a small number of samples.
>
> | Split | Method | IEE |  |  | CSQ |  |  | AL |  |  | SD |  |  | VCoT |  |  |  | Avg |
> |-------|---------|----------|-----------|---------|----------|-----------|---------|----------|-----------|---------|----------|-----------|---------|---------------|----------------|------------|---------|------|
> |       | Metric | Text Acc | Image Acc | Acc | Text Acc | Image Acc | Acc | Text Acc | Image Acc | Acc | Text Acc | Image Acc | Acc | Action Acc | Coordinate Acc | Image Acc | Acc | Avg |
> | **Split A** | MiniGPT5     | 19.00 | 24.00 | 21.50 | 23.76 | 35.64 | 29.60 | 7.69 | 23.08 | 15.39 | 4.00 | 8.00 | 6.00 | 3.17 | 1.27 | 3.81 | 2.75 | 14.65 |
> |            | Anole         | 15.00 | 17.00 | 16.00 | 70.27 | 51.35 | 60.81 | 19.23 | 11.54 | 15.39 | 20.00 | 12.00 | 16.00 | 4.44 | 0 | 6.35 | 3.60 | 20.24 |
> |            | SEED-LLAMA    | 17.00 | 22.00 | 19.50 | 54.05 | 43.24 | 48.65 | 11.54 | 11.54 | 11.54 | 26.00 | 24.00 | 25.00 | 3.18 | 2.54 | 3.81 | 3.18 | 21.57 |
> |            | MIO-Instruct  | 22.00 | 21.00 | 21.50 | 73.27 | 0 | 36.64 | 23.08 | 0 | 11.54 | 24.00 | 0 | 12.00 | 0 | 0 | 0 | 0 | 16.34 |
> | **Split B** | MiniGPT5     | 23.00 | 24.00 | 23.50 | 35.64 | 41.58 | 38.61 | 3.85 | 23.08 | 13.47 | 4.00 | 4.00 | 4.00 | 1.27 | 1.27 | 1.91 | 1.48 | 15.01 |
> |            | Anole         | 19.00 | 23.00 | 21.50 | 80.95 | 47.62 | 64.29 | 11.54 | 15.38 | 13.47 | 14.00 | 14.00 | 14.00 | 2.54 | 1.27 | 8.89 | 4.23 | 23.40 |
> |            | SEED-LLAMA    | 21.00 | 28.00 | 24.50 | 58.73 | 50.79 | 54.76 | 15.38 | 11.54 | 13.46 | 20.00 | 18.00 | 19.00 | 5.08 | 3.18 | 3.81 | 4.02 | 23.15 |
> |            | MIO-Instruct  | 26.00 | 27.00 | 26.50 | 79.21 | 0 | 39.61 | 11.54 | 0 | 5.77 | 22.00 | 0 | 11.00 | 0 | 0 | 0 | 0 | 16.78 |
> | **Overall** | MiniGPT5     | 21.50 | 24.00 | 22.80 | 29.70 | 38.56 | 34.13 | 5.66 | 23.08 | 14.37 | 4.00 | 6.00 | 5.00 | 2.22 | 1.27 | 2.92 | 2.13 | 15.67 |
> |            | Anole         | 17.00 | 20.00 | 18.50 | 70.30 | 49.00 | 59.65 | 15.38 | 13.46 | 14.42 | 17.00 | 13.00 | 15.00 | 3.49 | 0.64 | 7.62 | 3.89 | 22.30 |
> |            | SEED-LLAMA    | 19.00 | 25.00 | 22.00 | 56.44 | 46.53 | 51.49 | 13.46 | 11.54 | 12.50 | 23.00 | 21.00 | 22.00 | 4.13 | 2.85 | 3.81 | 3.61 | 22.32 |
> |            | MIO-Instruct  | 24.00 | 24.00 | 24.00 | 77.24 | 0 | 38.50 | 17.31 | 0 | 8.66 | 23.00 | 0 | 11.50 | 0 | 0 | 0 | 0 | 16.56 |

---

> ### Author Response · Authors · 2025-11-21
> **Response to W3-(1/2)**
>
> **W3: Concerns about diagnostic resolution on challenging unify tasks**
>
> A3: We thank the reviewer for appreciating our provision of step-wise accuracy analysis for action / location / image in VCoT. Regarding the concern about the "lack of difficulty stratification and milestone scoring," we provide the following explanation and further supplements in the revised manuscript:
>
> First, during task construction, we explicitly distinguish three difficulty levels based on maze size: 3 x 3, 4 x 4, 5 x 5. Crucially, larger mazes inherently require a longer average number of steps. The curve showing the variation in the number of required steps across different maze sizes is available in Appendix Figure 8. In the revised manuscript, we further include a table of average accuracy stratified by maze size (see table below). The table shows that Gemini-2.5-Flash-Image's action/coord/image accuracy on 3 x 3 mazes is 43.09%, 29.27%, 31.71%, while on 5 x 5mazes, it drops to 31.33%, 12.00%, 18.00%, respectively. Other models exhibit the same trend, thus providing an overall perspective on "difficulty stratification."
>
> | Size      | 3x3            |       |        | 4x4            |       |        | 5x5            |       |        |
> |-----------|----------------|-------|--------|----------------|-------|--------|----------------|-------|--------|
> | Metric    | action         | coord | image  | action         | coord | image  | action         | coord | image  |
> | Bagel     | 21.14          | 23.58 | 21.14  | 19.70          | 23.48 | 19.51  | 12.00          | 18.00 | 14.67  |
> | RecA      | 28.46          | 28.46 | 20.33  | 19.70          | 28.03 | 20.45  | 14.67          | 19.33 | 16.67  |
> | GPT-4o-Image | 21.95       | 18.70 | 25.20  | 20.45          | 17.42 | 20.45  | 19.33          | 10.00 | 16.00  |
> | Gemini2.5-Flash-Image | 43.09        | 29.27 | 31.71  | 38.64          | 20.45 | 23.48  | 31.33          | 12.00 | 18.00  |

---

> ### Author Response · Authors · 2025-11-21
> **Response to W3-(2/2)**
>
> Secondly, to address the concern that "0% success makes diagnosing the failure location difficult," we explicitly recast VCoT into a set of milestone-based metrics. The revised manuscript will include step-wise average accuracy tables for 4 representative U-MLLMs across steps 1-7 (as shown in Tables 1, 2, and 3 below): At Step 1, the action / location / image accuracy for all models is significantly higher than the random baseline (as shown in table under response to W4-(2/2), but these figures generally drop to approximately 1%–4% by Steps 5-7. Notably, the image and coordinate dimension shows a near "cliff-like" drop in accuracy starting from Step 2. This indicates that while models can still adequately identify the current position and generate a roughly correct next frame during the first 1-2 steps, the first capability lost during multi-step rollout is the consistent visual modeling of the maze state, which subsequently drives the overall degradation of actions. Therefore, despite a 0% sample-level success rate, by analyzing the model's performance across different steps and difficulty levels, we can specifically locate the failure modes to **two core bottlenecks: "multi-step visual state maintenance" and "multi-step consistency."**  This provides valuable diagnostic insights for subsequent targeted improvements to U-MLLM planning and visual memory mechanisms.
>
> | Action      | Step_1 | Step_2 | Step_3 | Step_4 | Step_5 | Step_6 | Step_7 |
> |-------------|--------|--------|--------|--------|--------|--------|--------|
> | Bagel       | 26.67  | 20.00  | 17.78  | 11.11  | 6.67   | 2.22   | 2.22   |
> | RecA        | 25.56  | 25.56  | 21.11  | 11.11  | 4.44   | 3.33   | 1.11   |
> | GPT-4o-Image| 28.89  | 23.33  | 23.33  | 10.00  | 4.44   | 2.22   | 0.00   |
> | Gemini2.5-Flash-Image | 52.22  | 48.89  | 37.78  | 16.67  | 10.00  | 1.11   | 1.11   |
>
> | Coordinate  | Step_1 | Step_2 | Step_3 | Step_4 | Step_5 | Step_6 | Step_7 |
> |-------------|--------|--------|--------|--------|--------|--------|--------|
> | Bagel       | 67.78  | 20.00  | 17.78  | 11.11  | 1.11   | 0.00   | 0.00   |
> | RecA        | 75.56  | 25.56  | 12.22  | 2.22   | 1.11   | 1.11   | 0.00   |
> | GPT-4o-Image| 38.89  | 10.00  | 5.56   | 4.44   | 4.44   | 2.22   | 2.22   |
> | Gemini2.5-Flash-Image | 23.33  | 20.00  | 17.78  | 17.78  | 10.00  | 1.11   | 0.00   |
>
> | Image       | Step_1 | Step_2 | Step_3 | Step_4 | Step_5 | Step_6 | Step_7 |
> |-------------|--------|--------|--------|--------|--------|--------|--------|
> | Bagel       | 43.33  | 25.56  | 7.78   | 7.78   | 1.11   | 1.11   | 0.00   |
> | RecA        | 46.67  | 22.22  | 5.56   | 5.56   | 2.22   | 1.11   | 0.00   |
> | GPT-4o-Image| 51.11  | 28.89  | 10.00  | 12.22  | 0.00   | 0.00   | 0.00   |
> | Gemini2.5-Flash-Image | 53.33  | 28.89  | 13.33  | 8.89   | 1.11   | 1.11   | 0.00   |
>
>
> We agree with the reviewer’s suggestion for more explicit difficulty stratification and "milestone" characterization. In the revised manuscript, we will supplement more detailed experimental content regarding VCoT, including visually plotting accuracy curves varying with the step count for each maze size, and summarizing the degradation patterns and typical failure points across different maze scales more clearly in the text, thereby making the diagnostic value of VCoT more intuitive to the reader.

---

> ### Author Response · Authors · 2025-11-21
> **Response to W4-(1/2)**
>
> **W4: Concerns about missing baselines for calibration**
>
> A4: Following the reviewer's suggestion, we have incorporated a simple baseline of random guessing and a human-established upper bound baseline for the Unify tasks (results are shown in the table below in response to W4-(2/2)).
>
> For random guessing, the method involves uniformly and randomly selecting one option for every text or image multiple-choice question. The experiments reveal that **the Random-Select metrics for the two high-precision / multi-step reasoning tasks, SpotDiff and VCoT, even exceed the performance of most current U-MLLMs.** This observation indirectly suggests that **existing U-MLLMs remain extremely weak in scenarios requiring precise difference localization or long-range state tracking**, confirming that these two Unify tasks are indeed "highly challenging items with significant difficulty discrimination" for the models. On the other hand, for the IEE, CSQ, and AL tasks, while some U-MLLMs' accuracy on the separate text or image modalities is only marginally higher than Random-Select, their scores on the $\text{acc}^+$ metric, which simultaneously requires "correct text and correct image," are far above the theoretical random baseline of Random-Select (approximately 6.25%). This indicates that **U-MLLMs have learned a certain degree of cross-modal consistency and understanding-generation synergy.** This indirectly validates the effectiveness of our Unify task design and the $\text{acc}/\text{acc}^+$ metrics in distinguishing random strategies from genuine unified capability. We will explicitly incorporate the Random-Select results into the relevant tables in the revised manuscript to facilitate a more intuitive interpretation of absolute scores and potential room for improvement for the reader.
>
> For the human evaluation, we asked two human experts with multimodal experience to complete the Unify tasks, selecting the best matching option for both the text and image multiple-choice questions. The average score of the two experts is reported in the table (below in response to W4-(2/2). The experiments demonstrate that human performance on IEE, CSQ, AL, and VCoT is significantly higher than that of all current U-MLLMs, while human accuracy on SpotDiff is slightly lower compared to their performance on other tasks. This result indicates, on one hand, that the Unify sub-tasks are not "noise questions that are difficult for humans to judge," confirming that the human upper bound is far above existing models. On the other hand, it collectively demonstrates from both ends of the spectrum (the random baseline and the human upper bound) that current U-MLLMs still have substantial room for improvement in unified capability, and the score scale provided by MME-Unify has clear lower and upper boundary references.

---

> ### Author Response · Authors · 2025-11-21
> **Response to W4-(2/2)**
>
> | Method        | IEE |  |  |  | CSQ |  |  |  | AL |  |  |  | SD |  |  |  | VCoT |  |  |  |  | Avg |
> |---------------|----------|-----------|---------|---------|----------|-----------|---------|---------|----------|-----------|---------|---------|----------|-----------|---------|---------|------------------|---------------|--------------|-------------|---------|------|
> | Metric        | Text Acc | Image Acc | Acc | Acc+ | Text Acc | Image Acc | Acc | Acc+ | Text Acc | Image Acc | Acc | Acc+ | Text Acc | Image Acc | Acc | Acc+ | Action Acc | Coord Acc | Image Acc | Acc | Acc+ | Avg |
> | Show-o2         | -        | -         | -   | -    | 79.21        | 53.47         | 66.34   | 56.44    | -        | -         | -   | -    | -        | -         | -   | -    | -           | -          | -          | -   | -    | 13.27   |
> | Random Select      | 29.00        | 27.50        | 28.25   | 11.50    | 24.75        | 31.68         | 28.22   | 5.94   | 23.08        | 28.85         | 25.97   | 7.69    | 30.00        | 31.00         | 30.50   | 9.00    | 19.26           | 1.73          | 21.73          | 14.24   | 0    | 25.43   |
> | Bagel    | 22.78       | 43.89         | 33.34   | 11.67    | 90.10        | 64.36         | 77.23   | 63.37    | 42.31        | 21.15         | 31.73   | 9.62    | 27.00        | 24.00         | 25.50   | 7.00    | 10.83           | 12.08          | 10.69          | 11.20   | 0    | 35.80   |
> | RecA  | 23.39        | 47.95         | 35.67   | 10.53    | 90.10        | 68.32         | 79.21   | 59.41    | 44.23        | 23.08         | 33.66   | 13.46    | 26.00        | 27.00         | 26.50   | 9.00    | 11.53           | 14.03          | 11.11          | 12.22   | 0    | 37.45   |
> | GPT-4o-Image    | 25.50        | 60.50         | 43.00   | 17.00    | 97.03        | 76.24         | 86.64   | 74.26    | 42.31        | 42.31         | 42.31   | 15.38    | 29.00        | 21.00         | 25.00   | 9.00    | 11.53           | 8.47          | 13.06          | 11.02   | 0    | 41.10   |
> | Gemini-2.5-Flash-Image  | 32.00        | 65.50         | 48.75   | 19.00    | 98.92        | 73.12         | 86.02   | 73.12    | 67.31        | 51.92         | 59.62   | 32.69    | 27.00        | 24.00         | 25.50   | 9.00    | 20.97           | 11.25          | 13.47          | 15.23   | 0    | 47.02   |
> | Human  | 96.00        | 100         | 98.00   | 96.00    | 97.52        | 97.52         | 97.52   | 97.52    | 94.23        | 94.23         | 94.23   | 94.23    | 19.00        | 19.00         | 19.00   | 19.00    | 100           | 100          | 100          | 100   | 100    | 81.75   |

---

> > ### Author Response · Authors · 2025-11-21
> > **Response to Q1**
> >
> > **Q1: How do you prevent "benchmark hacking**
> >
> > A1: We fully understand the reviewer's concern regarding whether the scoring can be easily manipulated by CLIP or negative samples. First, the unified tasks in MME-Unify are inherently designed to directly assess the understanding-generation synergy of U-MLLMs. By requiring the model to first comprehend before generating correctly, we mitigate potential manipulation risks: For instance, the AL (Auxiliary Line drawing) task requires the model to first understand the problem before it can draw the correct auxiliary line; VCoT (Visual Chain-of-Thought) requires the model to infer the correct next action and location before generating the correct image. Images generated solely based on input instructions cannot significantly improve the score, thereby effectively reducing the practical possibility of relying on CLIP-I for score hacking.
> >
> > Therefore, to achieve a high score on the unified tasks, the model must first correctly understand the instructions/problem, reason about the required image, and then perform conditional generation based on that understanding. If the model fails to comprehend the input and instead merely generates a high-quality image, it will be difficult to achieve a significant improvement in $\text{acc}^{+}$. To validate the trustworthiness of our evaluation strategy, we also conducted a consistency analysis against human evaluation and LLM-Judge results (as shown in Tables 1, 2, and 3 under response to Weakness 1).

---

> ### Author Response · Authors · 2025-11-21
> **Response to Q2**
>
> **Q2: What evidence can you provide that a high score on MME-U strongly correlates with the generation of high-quality, high-fidelity outputs that strictly follow complex instructions?**
>
> A2: We fully agree with the reviewer’s core concern regarding whether "MME-U scores truly correspond to high-quality, high-fidelity, and strongly instruction-following outputs," and have consequently supplemented two types of dedicated evaluations focusing on these three dimensions. First, we randomly sampled 200 instances from all unified tasks and asked two annotators with multimodal experience to score the generated images from each model 1-5 across three dimensions—Text-Following (instruction adherence), Image Quality (subjective quality), and Reference Similarity (consistency with the reference)—and then averaged these scores. We then computed the correlation between the average human score and the corresponding sample's CLIP-I score (as shown in table under response to W1-(1/2)). **For four representative U-MLLMs, the overall Kendall correlation coefficient obtained across various sub-tasks is approximately 0.71 (in high correlation range)**. This demonstrates that at the model level, models with "higher CLIP-I / $\text{acc}^{+}$ scores on MME-U" are indeed more likely to be rated by humans as having "better quality, higher fidelity, and stronger adherence to complex instructions." Second, on the complete unified dataset, we used GPT-4o as the LLM-Judge, scoring model outputs 1-5 along the same three dimensions (Text-Following / Image Quality / Reference Similarity) and performed a global ranking of the models accordingly (as shown in table 1 and 2 under response to W1-(2/2)). **The results show that this ranking is highly consistent overall with the model ranking obtained using CLIP-I / $\text{acc}^{+}$.** That is, under our unified setting, models ranking higher in MME-U scores are also the models deemed by the LLM-Judge to "better adhere to complex instructions, and show greater structural and reference consistency." These results collectively indicate that high scores on MME-U are not "pure proxy signals" detached from subjective quality. Rather, under the current evaluation setup, they maintain a stable and observable positive correlation with the human and LLM subjective judgments regarding "high quality, high fidelity, and strict adherence to complex instructions." We will supplement the revised manuscript's appendix with the aforementioned correlation statistics, model ranking comparisons, and several visual examples.

---

> > ### Author Response · Authors · 2025-11-21
> > **Response to Q3**
> >
> > **Q3: Have you considered incorporating evaluation dimensions that are harder to "hack" ?**
> >
> > A3: We sincerely thank the reviewer for their valuable suggestions! We fully agree on the importance of evaluation dimensions that are "harder to hack and more relevant to real-world applications" for a unified benchmark. In the current version, we have already piloted the LLM-Judge evaluation strategy: specifically, using GPT-4o to score the generation results of each model 1-5 across three dimensions—Text-Following, Image Quality, and Reference Similarity—on the unified tasks, and evaluating the models accordingly. We also highly value the reviewer's suggestion regarding stronger evaluation dimensions, such as "human preference scoring" and "object-level editing consistency." Moving forward, **we will prioritize adding human preference evaluation to the full benchmark, and explore introducing automatic metrics like object-level editing consistency in suitable sub-tasks (e.g., IEE, AL) to serve as a more rigorous and broader supplementary evaluation strategy.** We once again thank the reviewer for their constructive comments, which provide important guidance for us to further enhance MME-U’s alignment with real-world application scenarios.

---

> > > ### Author Response · Authors · 2025-11-21
> > > **Response to Q4**
> > >
> > > **Q4: How do your results help us locate and fix problems?**
> > >
> > > A4: We strongly agree with the reviewer's view that a "benchmark should possess diagnostic capabilities rather than just outputting a single score." MME-U was designed to align closely with this goal: the overall score is decomposed into three major dimensions (Understanding / Generation / Unify), and the unify tasks are further broken down into text accuracy, image accuracy, and $\text{acc} / \text{acc}^{+}$, which helps diagnose whether the bottleneck lies in "weak understanding, weak generation, or difficulty in synergistic enhancement between the two." **In Appendices E and F, we have also analyzed specific failure modes—such as "strong understanding but weak unify" (MIO-Instruct) or "strong generation but inability to complete unified tasks" through extensive case studies and sub-task comparisons.**
> > >
> > > Taking VCoT as an example, which the reviewer mentioned, the current paper already provides step-wise accuracy curves for the three channels (action / location / image) in Figure 4. We observe that the action, location, and image accuracy of four representative U-MLLMs in Step 1 are significantly higher than the random baseline (e.g., Gemini-2.5-Flash-Image achieves 52.22%, 23.33%, and 53.33%, respectively), indicating the models can identify the starting point and generate a reasonable first image. However, starting from Steps 2-3, all three curves decrease monotonically, particularly the image and location accuracy, which shows a near "cliff-like" drop to near 0, collapsing much earlier than action. Combined with the newly added table showing average accuracy stratified by maze size 3 x 3, 4 x 4, 5 x 5, as shown in table under response to W3-(1/2), and the step-wise average accuracy tables across Steps 1-7 for 4 state-of-the-art U-MLLMs (as shown in Tables 1, 2, and 3 under response to W3-(2/2)), we see that performance for action/location/image degrades as the maze size increases and the number of steps grows. This strongly suggests that current U-MLLMs in VCoT are primarily bottlenecked by "maintaining a coherent and correct visual state representation during multi-step processes," rather than the single-step action prediction itself. In the revised manuscript, we will move these step-wise and size-wise milestone metrics forward into the main text, and, following the reviewer's suggestion, explicitly report metrics such as "initial path identification / initial visualization quality," allowing model developers to target improvements in long-range visual reasoning and memory mechanisms accordingly.

---

> > > > ### Comment · Reviewer_VRFA · 2025-11-22
> > > > **Response to Authors**
> > > >
> > > > Thank you for the detailed explanations and supplementary materials provided in your rebuttal.
> > > >
> > > > I particularly appreciate the efforts made to address my two main concerns regarding the necessity and the diagnostic capability of this benchmark. Your design choices, which comprehensively balance both generation and understanding, have effectively resolved these issues.
> > > >
> > > > In light of these clarifications, I am now willing to revise my rating.

---

> > > > > ### Author Response · Authors · 2025-11-22
> > > > >
> > > > > We deeply appreciate your decision to raise the score. Your insightful feedback greatly contributed to enhancing the quality of our benchmark, and we truly valued the opportunity to engage in meaningful discussions with you.
> > > > >
> > > > > Warm regards, The Authors

---

### Official Review · Reviewer_uqqm · 2025-10-31

**Soundness:** 3
**Presentation:** 3
**Contribution:** 3
**Rating:** 6
**Confidence:** 4

**Summary:**

This paper introduces MME-Unify (MME-U), the first open benchmark for Unified Multimodal Large Language Models (U-MLLMs). It evaluates U-MLLMs across three core capabilities: understanding, generation, and hybrid (Unify) multimodal generation. MME-U integrates 12 datasets, standardizes conventional tasks, and designs five unique hybrid generation sub-tasks (e.g., Visual Chain-of-Thought and geometry problem-solving with auxiliary lines). Evaluations on 12 U-MLLMs reveal that current models exhibit significant deficiencies in balancing different capabilities, following complex instructions, and performing multi-step unified tasks.

**Strengths:**

(1) MME-U is the first benchmark for standardized unified multimodal generation tasks (unify capability). These tasks require models to collaboratively integrate reasoning with multimodal outputs, which is quite novel and also fills certain gaps in existing evaluations.
(2) MME-U provides a unified scoring framework for comprehension, generation, and unified tasks. It standardizes complex metrics to a [0, 100] scale, offering an intuitive and comparable overall MME-U score.

**Weaknesses:**

(1) In the unified tasks, image evaluation relies on multiple-choice questions based on CLIP similarity, which may allow models to exploit the evaluation. This simplification reduces the rigor in assessing generation quality and precise adherence to instructions. Additionally, if the options are very similar, such an evaluation may also be inaccurate.
(2) On multi-step reasoning tasks like Visual CoT, none of the models succeeded, with Acc results at 0%. This indicates that the task is overly difficult and may not provide discriminative evaluation value.

**Questions:**

Given the potential risk of exploitation in CLIP-I-based evaluation, do the authors plan to combine text generation metrics or use an LLM judge to perform a holistic assessment of generated images and text, in order to achieve a more rigorous evaluation?

---

> ### Author Response · Authors · 2025-11-21
> **Response to W1-(1/2)**
>
> We are  grateful to the reviewer for the affirmation of our work and the valuable feedback provided. Your feedback is crucial for the continuous improvement of our benchmark.
>
> **W1: Concerns about evaluation robustness**
>
> *Response regarding the concern that "multiple-choice questions based on CLIP similarity may allow models to exploit the evaluation."*
>
> We thank the reviewer for their reminder. Our adoption of the CLIP-I based multiple-choice evaluation is primarily due to two considerations:
>
> * (i) **Unified Representation and Metric Design:** The goal of the unified tasks is not to evaluate generation or understanding in isolation, but to assess the U-MLLM's synergistic capability of "first understanding, then generating conditionally" on the same sample. While continuous scoring metrics (e.g., per-sample LLM-Judge) can finely describe image quality, they are difficult to naturally convert into a discrete accuracy rate representing "understanding and generation." Moreover, in our setting, which includes thousands of samples, multiple candidate images would need to be sent to the LLM simultaneously, resulting in high evaluation cost and difficulty in reproducibility. In contrast, we structure each sample as a "text multiple-choice question + image multiple-choice question," use CLIP-I to map the generated image to a discrete decision among 4 candidate images, and use the two 0-1 metrics, $\text{acc}$ and $\text{acc}^{+}$, to intuitively measure whether "the text option was correctly chosen, and the image matches the reference answer." This achieves a unified and reproducible evaluation standard across the three dimensions: Understanding, Generation, and Unify.
>
> * (ii) **Task Design Requires "Understanding First, Then Generation," Mechanistically Inhibiting Score Hacking:** The goal of the unified task is not simply to "draw a high-quality image based on a text instruction," but to require the model to genuinely comprehend the input and reason out "what kind of intermediate image needs to be generated to complete the task," and then create the image accordingly. For example, in AL (Auxiliary Lines), we only retain samples that "must have a specific auxiliary line drawn to solve the problem." If the model fails to understand the geometric relationship and merely draws random lines based on the instruction, it will be difficult to match the correct option. In VCoT (Visual CoT), the model must first infer the correct next action and position before generating the corresponding state image; otherwise, subsequent coordinates and images will rapidly diverge. As the result, generic images generated solely from input instructions cannot match the correct option. This dependency minimizes the risk of exploiting CLIP-I for high scores.
>
> **Response to the concern that "this evaluation might be inaccurate if options are very similar":** We acknowledge the reviewer's concern. To further validate the rationality of our evaluation strategy, we conducted two sets of accuracy verification experiments. First, we randomly sampled 200 instances from the unified tasks and asked three experts to score the generation results of four advanced U-MLLMs (Bagel[1], GPT-4o-Image[2], RecA[3], Gemini2.5-Flash-Image[4]) on a scale of 1-5 across three dimensions—Text-Following, Image Quality, and Reference Similarity—and took the average as the final score. Subsequently, we calculated the Kendall's $\tau_b$ correlation coefficient between the CLIP-I scores of the model-generated images and the human ratings for each task (as shown in Table below):
>
> | | IEE | CSQ | AL | SD | VCoT | Overall |
> | :--- | :--- | :--- | :--- | :--- | :--- | :--- |
> | **Kendall’s $\tau$** | 0.711982 | 0.73212 | 0.71725 | 0.707045 | 0.69195 | 0.70871 |
>
> The results show that the correlation generally falls within the range of approximately $0.69\text{--}0.73$ across various sub-tasks, with an overall average correlation of about $0.71$, **which in the range of strong correlation**, demonstrating that under our task design, there is relatively stable consistency between CLIP-I and human perceptual ratings.
>
> ---
>
> [1] Deng, et al. "Emerging Properties in Unified Multimodal Pretraining." arXiv preprint arXiv:2505.14683 (2025).
>
> [2] https://openai.com/zh-Hans-CN/index/introducing-4o-image-generation/
>
> [3] Xie, Ji, et al. "Reconstruction Alignment Improves Unified Multimodal Models." arXiv preprint arXiv:2509.07295 (2025).
>
> [4] https://deepmind.google/models/gemini/flash/

---

> > ### Author Response · Authors · 2025-11-21
> > **Response to W1-(2/2)**
> >
> > Secondly, on the complete dataset, we used both CLIP-I and LLM-Judge (where the LLM-Judge scoring method is consistent with the Human Rating method) to score and rank the five newly evaluated U-MLLMs on the unified tasks. The resulting model ranking is shown in the table below. While Bagel and RecA showed comparable performance on the VCoT task, resulting in a ranking inversion on this specific task, all models otherwise demonstrated strong consistency across other tasks and in the overall ranking.
> >
> > ## CLIP-based Evaluation for image QA
> > | Model                | IEE   | CSQ   | AL    | SD    | VCoT  | Avg   |
> > |----------------------|-------|-------|-------|-------|-------|-------|
> > | Show-o2              | -     | 53.47 | -     | -     | -     | 10.69 |
> > | Bagel                | 43.89 | 64.36 | 21.15 | 24.00 | 10.69 | 32.82 |
> > | RecA                 | 47.95 | 68.32 | 23.08 | 27.00 | 11.11 | 35.49 |
> > | GPT-4o-Image         | 60.50 | 76.24 | 42.31 | 21.00 | 13.06 | 42.62 |
> > | Gemini 2.5-Flash-Image | 65.50 | 73.12 | 51.92 | 24.00 | 13.47 | 45.60 |
> >
> >
> > ## LLM-Judge-based Evaluation for image QA
> >
> > | Model                 | IEE   | CSQ   | AL    | SD    | VCoT  | Avg   |
> > |-----------------------|-------|-------|-------|-------|-------|-------|
> > | Show-o2               | -     | 56.33 | -     | -     | -     | 11.27 |
> > | Bagel                 | 49.47 | 66.93 | 37.07 | 47.67 | 52.33 | 50.69 |
> > | RecA                  | 49.60 | 68.07 | 43.73 | 48.67 | 52.00 | 52.41 |
> > | GPT-4o-Image          | 68.87 | 78.80 | 58.60 | 45.47 | 52.60 | 60.87 |
> > | Gemini 2.5-Flash-Image| 80.33 | 72.33 | 65.40 | 53.07 | 58.67 | 65.96 |
> >
> > Concurrently, we will further supplement the appendix with a visual comparison providing CLIP-I, LLM-Judge, and human ratings simultaneously for the same batch of samples, showcasing the scores and rankings of these three scoring methods at the sample level, while also analyzing typical cases where they are highly consistent and where they diverge. This aims to more intuitively present the advantages and limitations of the current CLIP-I approach. We thank the reviewer for their valuable suggestions.

---

> ### Author Response · Authors · 2025-11-21
> **Response to W2-(1/2)**
>
> **W2: Concerns about VCoT may not provide discriminative evaluation value.**
>
> A2: We thank the reviewer for the suggestion. We designed Visual CoT to reveal systematic deficiencies in a model's "consistent understanding + generation across multiple steps" through fine-grained metrics. Our implementation uses an extremely strict sequence-level success definition: a trajectory is only counted as successful if all three decisions (action / location / image) in all steps are correct. Considering that the average trajectory involves about 3.5 steps and 3 decisions per step, the model needs to be correct in more than ten discrete choices to succeed. Therefore, a 0% sequence-level success rate under this definition does not mean the task is "information-less," but rather reflects the current capability bottleneck of U-MLLMs in long-range, unified visual reasoning tasks.
>
> Nevertheless, recognizing the difficulty of this type of unified task, we use $\text{acc}$ as the final unified score. For tasks like CSQ/IEE/SD/AL, $\text{acc}$ is defined as the average of the text multiple-choice accuracy and the image multiple-choice accuracy. For Visual CoT, we use the average of the step-wise accuracy across the three dimensions (action, location, and image) as the $\text{acc}$ for this sub-task. Consequently, even if the accuracy of Visual CoT task is 0% under the trajectory-level standard, it still contributes to the U-Score in the form of step-wise accuracy and is not discarded as a "zero signal."
>
> We also provided the step-level fine-grained results for VCoT in Figure 4 of the main text and Appendix E, and further discussed the evaluation conclusions derived from this task in Appendix E's "Inadequate Visual CoT Capability in Unified Models." We separately tracked the accuracy of action/location/image for each step, observing that models perform significantly better than random in the initial steps. However, as the number of steps increases, the accuracy across all three paths monotonically decreases, and errors gradually accumulate. Differences in model strengths across the three types of questions were also evident. Therefore, although current U-MLLMs struggle to fully solve the VCoT task, the step-level $\text{acc}$ metric can effectively differentiate models and analyze the capability bottleneck of existing U-MLLMs in multi-step unified visual reasoning, thus remaining a diagnostically valuable component of the unified score.
>
> To further enhance the evaluation value of VCoT, we have supplemented the revised edition with a table showing average accuracy stratified by maze size (Table 1 below). From the table, it can be observed that Gemini-2.5-Flash-Image's action/coord/image accuracy on the 3 x 3 maze is 43.09%, 29.27%, 31.71%, respectively, while these drop to 31.33%, 12.00%, 18.00% on the 5 x 5 maze. Other models exhibit the same trend, thereby providing an overall perspective on "difficulty stratification."
>
> | Size      | 3x3            |       |        | 4x4            |       |        | 5x5            |       |        |
> |-----------|----------------|-------|--------|----------------|-------|--------|----------------|-------|--------|
> | Metric    | action         | coord | image  | action         | coord | image  | action         | coord | image  |
> | Bagel     | 21.14          | 23.58 | 21.14  | 19.70          | 23.48 | 19.51  | 12.00          | 18.00 | 14.67  |
> | RecA      | 28.46          | 28.46 | 20.33  | 19.70          | 28.03 | 20.45  | 14.67          | 19.33 | 16.67  |
> | GPT-4o-Image | 21.95       | 18.70 | 25.20  | 20.45          | 17.42 | 20.45  | 19.33          | 10.00 | 16.00  |
> | Gemini2.5-Flash-Image | 43.09        | 29.27 | 31.71  | 38.64          | 20.45 | 23.48  | 31.33          | 12.00 | 18.00  |

---

> ### Author Response · Authors · 2025-11-21
> **Response to W2-(2/2)**
>
> Concurrently, we reconstructed VCoT into a set of step-wise metrics and will provide tables (Tables 1, 2, and 3 below) in the revised manuscript showing the step-wise average accuracy for four representative U-MLLMs across steps 1-7:
>
> In Step 1, the accuracy of action / location / image for all models is significantly higher than the random baseline (as shown in table under response to Reviewer VRFA.W4-(2/2))
> , but this universally drops to approximately 1%-4% by Steps 5-7. Notably, the accuracy in the image and coordinat dimension exhibits a near "cliff-like" drop starting from Step 2, approaching faster than the action dimensions. This suggests that while the models can still correctly identify the current position and generate a roughly correct next frame in the first 1-2 steps, the first capability lost during the multi-step reasoning process is the consistent visual modeling of the maze state, which subsequently leads to the overall degradation of action and coordinate prediction. Therefore, despite the 0% sample-level success rate, by analyzing model performance across different steps and different difficulty samples, we can specifically locate the failure mode to two core bottlenecks: **"multi-step visual state maintenance"** and **"long-range consistency."** This provides essential diagnostic value for targeted improvements to U-MLLMs' planning and visual memory mechanisms.
>
> | Action      | Step_1 | Step_2 | Step_3 | Step_4 | Step_5 | Step_6 | Step_7 |
> |-------------|--------|--------|--------|--------|--------|--------|--------|
> | Bagel       | 26.67  | 20.00  | 17.78  | 11.11  | 6.67   | 2.22   | 2.22   |
> | RecA        | 25.56  | 25.56  | 21.11  | 11.11  | 4.44   | 3.33   | 1.11   |
> | GPT-4o-Image| 28.89  | 23.33  | 23.33  | 10.00  | 4.44   | 2.22   | 0.00   |
> | Gemini2.5-Flash-Image | 52.22  | 48.89  | 37.78  | 16.67  | 10.00  | 1.11   | 1.11   |
>
> | Coordinate  | Step_1 | Step_2 | Step_3 | Step_4 | Step_5 | Step_6 | Step_7 |
> |-------------|--------|--------|--------|--------|--------|--------|--------|
> | Bagel       | 67.78  | 20.00  | 17.78  | 11.11  | 1.11   | 0.00   | 0.00   |
> | RecA        | 75.56  | 25.56  | 12.22  | 2.22   | 1.11   | 1.11   | 0.00   |
> | GPT-4o-Image| 38.89  | 10.00  | 5.56   | 4.44   | 4.44   | 2.22   | 2.22   |
> | Gemini2.5-Flash-Image | 23.33  | 20.00  | 17.78  | 17.78  | 10.00  | 1.11   | 0.00   |
>
> | Image       | Step_1 | Step_2 | Step_3 | Step_4 | Step_5 | Step_6 | Step_7 |
> |-------------|--------|--------|--------|--------|--------|--------|--------|
> | Bagel       | 43.33  | 25.56  | 7.78   | 7.78   | 1.11   | 1.11   | 0.00   |
> | RecA        | 46.67  | 22.22  | 5.56   | 5.56   | 2.22   | 1.11   | 0.00   |
> | GPT-4o-Image| 51.11  | 28.89  | 10.00  | 12.22  | 0.00   | 0.00   | 0.00   |
> | Gemini2.5-Flash-Image | 53.33  | 28.89  | 13.33  | 8.89   | 1.11   | 1.11   | 0.00   |

---

> ### Author Response · Authors · 2025-11-21
> **Response to Q1**
>
> **Q1: Do the authors plan to combine text generation metrics or use an LLM judge to perform a holistic assessment of generated images and text.**
>
>
> We thank the reviewer for the suggestion. We did consider using an LLM-Judge (such as GPT-4o) to uniformly score the "text + image" results for all unified and generation tasks. However, the objective of the unified tasks is not merely to evaluate generation or understanding in isolation, but to intuitively assess the U-MLLM's synergistic capability in "understanding first, then conditionally generating." While common continuous scoring metrics (such as per-sample LLM-Judge) facilitate fine-grained assessment of image quality, they struggle to naturally map to a discrete accuracy rate representing this synergistic capability. Furthermore, inputting multiple candidate images simultaneously into an LLM significantly increases the evaluation cost.
>
> Therefore, we made a trade-off in our evaluation strategy: for the main results, we adopted the automated scheme of "Text Multiple-Choice + CLIP-I based Image Multiple-Choice." This allows the understanding-generation synergy to be uniformly mapped to an intuitive, reproducible accuracy metric. Concurrently, we will supplement the revised manuscript with additional experiments using LLM-Judge evaluation results to verify the consistency between our evaluation strategy and the LLM-Judge.
>
> Following the reviewer's suggestion, we have further supplemented the revised manuscript with consistency verification experiments between CLIP-I and LLM-Judge. In the unified tasks, the text portion of CSQ, AL, SpotDiff, and VCoT are all closed-ended multiple-choice questions, which can be directly evaluated using Accuracy. For image evaluation and the open-ended editing descriptions in IEE, we introduced GPT-4o as the LLM-Judge on the complete dataset:
>
> * **For the image side,** given the model-generated image and the correct reference image, GPT-4o was asked to score 1-5 on three dimensions—Text-Following, Image Quality, and Reference Similarity, with the scores then averaged.
> * **For the text side of IEE,** given the editing instruction, the standard description, and the model-generated description, GPT-4o scored based on two dimensions: Reference Similarity and Rationality.
>
> Based on this, we calculated the LLM-Judge score for the unified tasks for each model and compared it with the CLIP-I / CLIP-T based score. The results (see table below) show that, while Bagel and RecA showed comparable performance on the VCoT task, resulting in a ranking inversion on this specific task, all models otherwise demonstrated strong consistency across other tasks and in the overall ranking., verifying that the CLIP-I approach and the more rigorous LLM-Judge are directionally aligned at the model level under the current setup. We agree with the reviewer's suggestion regarding "introducing more rigorous evaluation" and will supplement the appendix with the LLM-Judge evaluation tables as a reference, allowing the community to reproduce and extend this approach if needed.
>
> ## CLIP-based Evaluation for image QA
> | Model                | IEE   | CSQ   | AL    | SD    | VCoT  | Avg   |
> |----------------------|-------|-------|-------|-------|-------|-------|
> | Show-o2              | -     | 53.47 | -     | -     | -     | 10.69 |
> | Bagel                | 43.89 | 64.36 | 21.15 | 24.00 | 10.69 | 32.82 |
> | RecA                 | 47.95 | 68.32 | 23.08 | 27.00 | 11.11 | 35.49 |
> | GPT-4o-Image         | 60.50 | 76.24 | 42.31 | 21.00 | 13.06 | 42.62 |
> | Gemini 2.5-Flash-Image | 65.50 | 73.12 | 51.92 | 24.00 | 13.47 | 45.60 |
>
> ## LLM-Judge-based Evaluation for image QA
>
> | Model                 | IEE   | CSQ   | AL    | SD    | VCoT  | Avg   |
> |-----------------------|-------|-------|-------|-------|-------|-------|
> | Show-o2               | -     | 56.33 | -     | -     | -     | 11.27 |
> | Bagel                 | 49.47 | 66.93 | 37.07 | 47.67 | 52.33 | 50.69 |
> | RecA                  | 49.60 | 68.07 | 43.73 | 48.67 | 52.00 | 52.41 |
> | GPT-4o-Image          | 68.87 | 78.80 | 58.60 | 45.47 | 52.60 | 60.87 |
> | Gemini 2.5-Flash-Image| 80.33 | 72.33 | 65.40 | 53.07 | 58.67 | 65.96 |
>
>
> | IEE-Text              | CLIP-T | LLM-Judge |
> |-----------------------|--------|-----------|
> | Show-o2               | -      | -         |
> | Bagel                 | 22.78  | 46.62     |
> | RecA                  | 23.39  | 49.98     |
> | GPT-4o-Image          | 25.50  | 54.71     |
> | Gemini-2.5-Flash-Image| 32.00  | 63.33     |

---

### Official Review · Reviewer_xuiz · 2025-10-31

**Soundness:** 3
**Presentation:** 3
**Contribution:** 4
**Rating:** 6
**Confidence:** 3

**Summary:**

This paper introduces MME-Unify (MME-U), a new and comprehensive benchmark designed to evaluate Unified Multimodal Large Language Models. The authors identify a critical gap in existing evaluations: the lack of a standardized benchmark that assesses comprehension, generation, and mixed-modality generation capabilities simultaneously. MME-U addresses this by integrating tasks into three domains: (1) Multimodal Understanding, (2) Multimodal Generation, and (3) a novel set of unify tasks. The Unify tasks are specifically designed to test how a model's understanding and generation capabilities can mutually enhance each other, featuring five new subtasks like Visual CoT and Auxiliary Lines . The authors evaluate 12 unified MLLMs and find that current models have significant room for improvement.

**Strengths:**

- The paper's primary contribution is the introduction of five unify subtasks. This is the first standardized benchmark designed to rigorously assess the synergistic mixed-modality generation capabilities of U-MLLMs (e.g., reasoning before drawing), which has been a major gap in the field.
- The benchmark curates and unifies a wide array of tasks from 12 existing datasets for understanding and generation. By reformatting all understanding and unify tasks into a standardized multiple-choice format and normalizing generation metrics, MME-U provides a consistent and reproducible framework for model comparison.
- The evaluation of 22 models provides a clear snapshot of the current U-MLLM landscape. The results effectively highlight the performance gap between models and demonstrate that even top-performing models struggle with complex instruction following and multi-step unified tasks .

**Weaknesses:**

- The main weakness is the evaluation strategy for image generation in the "Unify" tasks. Using CLIP-I similarity to match a generated image against multiple-choice image options can be hacked.
- While the Unify tasks are novel, the benchmark's Understanding and Generation sections are primarily a "benchmark of benchmarks," curating tasks from many existing sources. This makes the overall contribution feel more incremental than groundbreaking.
- The paper reports findings but fails to sufficiently discuss how its conclusions diverge from or challenge those of existing benchmarks. The analysis would be significantly stronger if it highlighted unique insights or model ranking differences revealed only by MME-U. This discussion is crucial for demonstrating the distinct value of this new comprehensive benchmark.
- The benchmark appears to rely heavily on synthetic data generation, using models like GPT-4o to create QA pairs, explanations, and negative samples. The methodology for quality control and ensuring these synthetic data points are accurate, unbiased, and sufficiently challenging is not well-detailed, raising concerns about data quality and potential artifacts.

**Questions:**

- The evaluation is missing several key sota models (e.g., Gemini 2.5-Pro, Bagel, Qwen-image).
- See weaknesses.

---

> ### Author Response · Authors · 2025-11-21
> **Response to W1-(1/2)**
>
> We sincerely thank you for your meticulous review and insightful suggestions. We are very grateful for the opportunity to clarify some issues in our work and discuss future directions with you.
>
> **W1: Concerns about evaluation strategy for image generation in the "Unify" tasks**
>
> A1: We adopted the CLIP-I based image multiple-choice evaluation strategy primarily for the unified assessment of the capability of understanding–generation synergy of U-MLLMs:
>
> * **Unified Representation and Metric Design:** Our objective is to quantify the unified synergy in U-MLLMs, rather than evaluating generation and understanding in isolation. While continuous metrics like LLM-Judge provide interpretable assessments of image quality, they incur high computational costs and struggle to directly map synergistic capabilities to a discrete "success" metric. We therefore struck a balance in our evaluation strategy: by aligning image evaluation with the text multiple-choice format, we map the model's synergistic performance to discrete accuracy scores, allowing for intuitive and unified benchmarking.
>
> * **Task Requirement for Prior Understanding and Reasoning Prevents Score Hacking Risk:** The tasks require the model to first understand before generating correctly, thus avoiding potential score hacking risks: Since the unify tasks demand the model to comprehend and reason based on the input instructions before generating an image aligned with the correct answer. For instance, AL necessitates understanding the geometric problem to draw the correct auxiliary line, while VCoT requires inferring the next action to generate the correct state. Since the correct visual output depends on specific reasoning logic, generic images generated solely from input instructions cannot match the correct option. This dependency minimizes the risk of exploiting CLIP-I for high scores.
>
>
> To further validate the robustness of our evaluation strategy, we conducted two sets of correlation experiments. First, we randomly sampled 200 instances from the unified tasks and asked three experts to score the generation results of four advanced U-MLLMs (Bagel[1], GPT-4o-Image[2], RecA[3], Gemini2.5-Flash-Image[4]) on a scale of 1-5 across three dimensions—Text-Following, Image Quality, and Reference Similarity, and took the average as the final score. Subsequently, we calculated the Kendall's $\tau_b$ correlation coefficient between the CLIP-I scores of the model-generated images and the human ratings for each task (as shown in Table 1 below). The results show that the correlation generally falls within the range of approximately 0.69-0.73 across various sub-tasks, with an overall average correlation of about 0.71 **This falls within the range of strong correlation**, demonstrating that under our task design, there is relatively stable consistency between CLIP-I and human perceptual ratings.
>
> | | IEE | CSQ | AL | SD | VCoT | Overall |
> | :--- | :--- | :--- | :--- | :--- | :--- | :--- |
> | **Kendall’s $\tau$** | 0.711982 | 0.73212 | 0.71725 | 0.707045 | 0.69195 | 0.70871 |
>
>
> ---
>
> [1] Deng, et al. "Emerging Properties in Unified Multimodal Pretraining." arXiv preprint arXiv:2505.14683 (2025).
>
> [2] https://openai.com/zh-Hans-CN/index/introducing-4o-image-generation/
>
> [3] Xie, Ji, et al. "Reconstruction Alignment Improves Unified Multimodal Models." arXiv preprint arXiv:2509.07295 (2025).
>
> [4] https://deepmind.google/models/gemini/flash/

---

> ### Author Response · Authors · 2025-11-21
> **Response to W1-(2/2)**
>
> Secondly, on the complete dataset, we used both CLIP-I and LLM-Judge (where the LLM-Judge scoring method is consistent with the Human Rating method) to score and rank the five newly evaluated U-MLLMs on the unified tasks. The resulting model ranking is shown in the table below.  While Bagel and RecA showed comparable performance on the VCoT task, resulting in a ranking inversion on this specific task, all models otherwise demonstrated strong consistency across other tasks and in the overall ranking.
>
> ## CLIP-based Evaluation for image QA
> | Model                | IEE   | CSQ   | AL    | SD    | VCoT  | Avg   |
> |----------------------|-------|-------|-------|-------|-------|-------|
> | Show-o2              | -     | 53.47 | -     | -     | -     | 10.69 |
> | Bagel                | 43.89 | 64.36 | 21.15 | 24.00 | 10.69 | 32.82 |
> | RecA                 | 47.95 | 68.32 | 23.08 | 27.00 | 11.11 | 35.49 |
> | GPT-4o-Image         | 60.50 | 76.24 | 42.31 | 21.00 | 13.06 | 42.62 |
> | Gemini 2.5-Flash-Image | 65.50 | 73.12 | 51.92 | 24.00 | 13.47 | 45.60 |
>
>
> ## LLM-Judge-based Evaluation for image QA
>
> | Model                 | IEE   | CSQ   | AL    | SD    | VCoT  | Avg   |
> |-----------------------|-------|-------|-------|-------|-------|-------|
> | Show-o2               | -     | 56.33 | -     | -     | -     | 11.27 |
> | Bagel                 | 49.47 | 66.93 | 37.07 | 47.67 | 52.33 | 50.69 |
> | RecA                  | 49.60 | 68.07 | 43.73 | 48.67 | 52.00 | 52.41 |
> | GPT-4o-Image          | 68.87 | 78.80 | 58.60 | 45.47 | 52.60 | 60.87 |
> | Gemini 2.5-Flash-Image| 80.33 | 72.33 | 65.40 | 53.07 | 58.67 | 65.96 |
>
> Concurrently, we will further supplement the appendix with a visual comparison providing CLIP-I, LLM-Judge, and human ratings simultaneously for the same batch of samples, showcasing the scores and rankings of these three scoring methods at the sample level, while also analyzing typical cases where they are highly consistent and where they diverge. This aims to more intuitively present the advantages and limitations of the current CLIP-I approach. We thank the reviewer for their valuable suggestions.

---

> > ### Author Response · Authors · 2025-11-21
> > **Response to W2**
> >
> > **W2: Concerns about overall contribution feel more incremental than groundbreaking**
> >
> > A2: Thank you for the constructive feedback regarding our benchmark's contribution. We agree that MME-Unify does reuse some data from currently public benchmarks in its Understanding and Generation sections. However, our goal was not simply to compile existing tasks, but to address the inability of current multimodal benchmarks to uniformly assess U-MLLMs across evaluation dimensions and strategies. We achieved this by reconstructing evaluation strategies and dimensions to establish a systematic, unified assessment framework for evaluating current U-MLLMs' understanding, generation, and the unified capability where generation and understanding mutually promote each other. Our specific contributions are as follows:
> >
> > Regarding **evaluation strategy**, we made the following three contributions:
> >
> > * (i) **Understanding Tasks:** We uniformly reconstructed 24 heterogeneous sub-tasks into a standardized multiple-choice format. By adopting a consistent, formalized matching strategy, we eliminated discrepancies in annotation formats and scoring rules. This allows results from multi-source understanding tasks to be directly compared along a single evaluation dimension (Section 2.1).
> >
> > * (ii) **Generation Tasks:** We normalized inconsistent input/output structures across 6 categories of generative sub-tasks by mapping attributes to unified fields and designing consistent system prompts. This integrates diverse generative tasks into a cohesive framework without altering their original semantics (Section 2.2).
> >
> > * (iii) **Metric Level:** At the metric level, we unified the various domain-specific standard metrics used in each sub-task (such as CLIP-I/CLIP-T, FID, etc.) into the $[0, 100]$ range. Building upon this, we constructed three sub-scores (Understanding, Generation, and Unify) and an overall MME-U score. Furthermore, we specifically designed two metrics for the unified tasks, $\text{acc}$ and $\text{acc}^{+}$, to measure the unified capability of "text and image being simultaneously correct" (Section 2.3, Appendix G). These designs allow evaluation signals, previously scattered across different benchmarks, to be systematically integrated and decomposed under a single scale for the first time, providing the necessary foundation for subsequent analysis of U-MLLM capabilities.
> >
> > Regarding the evaluation dimension, unlike existing benchmarks restricted to single dimensions—either understanding (e.g., MME [1], MMBench [2]) or generation (e.g., GenEval [3], Wise [4])—MME-Unify centers on the core unified capability of U-MLLMs. We systematically designed five novel unified tasks (CSQ, IEE, SD, AL, VCoT) covering "mixed-modal generation" and the "mutual promotion between understanding and generation." Combined with the unified strategies above, this forms a framework that simultaneously assesses understanding, generation, and their synergistic effects (see table below). Crucially, this framework reveals capability bottlenecks of U-MLLMs, which often missed by standard benchmarks, such as the observation that current U-MLLM training paradigms result in a trade-off rather than mutual enhancement between understanding and generation capabilities (detailed in Section 3.1 and Appendices E, F).
> >
> > We will strengthen the explanation of the aforementioned design rationale and derived insights in the revised manuscript to more clearly present the benchmark's core contribution in methodology and analysis, which is superior to "mere task aggregation." Thank you for your constructive comments. This improvement will better highlight the innovativeness and comprehensiveness of MME-Unify.
> >
> > | Benchmark  | Understand | Generation | Mix-Modality Generation | Und-to-Gen | Gen-to-Und |
> > |------------|------------|------------|--------------------------|------------|------------|
> > | MME        | ✔️         | ❌         | ❌                       | ❌         | ❌         |
> > | MMBench    | ✔️         | ❌         | ❌                       | ❌         | ❌         |
> > | GenEval    | ❌         | ✔️         | ❌                       | ❌         | ❌         |
> > | Wise       | ❌         | ✔️         | ❌                       | ✔️         | ❌         |
> > | MME-Unify  | ✔️         | ✔️         | ✔️                       | ✔️         | ✔️         |
> >
> > ---
> > [1] Fu, et al. "MME: A Comprehensive Evaluation Benchmark for Multimodal Large Language Models." arXiv preprint arXiv:2306.13394 (2025).
> >
> > [2] Liu, et al. "MMBench: Is Your Multi-modal Model an All-around Player?" arXiv preprint arXiv:2307.06281 (2024).
> >
> > [3] Ghosh, et al. "GenEval: An Object-Focused Framework for Evaluating Text-to-Image Alignment." arXiv preprint arXiv:2310.11513 (2023).
> >
> > [4] Niu, et al. "WISE: A World Knowledge-Informed Semantic Evaluation for Text-to-Image Generation." arXiv preprint arXiv:2503.07265 (2025).

---

> > > ### Author Response · Authors · 2025-11-21
> > > **Response to W3**
> > >
> > > **W3: Concerns about discuss how its conclusions diverge from or challenge those of existing benchmarks.**
> > >
> > > A3: We fully agree with the reviewer's suggestion that providing only an overall score is insufficient to demonstrate the value of a new benchmark; it must also reveal new phenomena and challenges. In the revised manuscript, we have supplemented Section 3.1, along with Appendices E and F, with a discussion of the unique conclusions revealed by MME-U and the observed differences in model rankings. We will analyze the key findings one by one to provide deeper insights and make more meaningful contributions to the community:
> > >
> > > 1.  **Current U-MLLMs often exhibit a trade-off rather than the ideal mutual promotion between understanding and generation capabilities:** We observed, after providing the scores for the three dimensions (Understanding, Generation, and Unify), that although Janus-Pro performs outstandingly on the understanding benchmarks MMBench[1] (79.2), MMM-U[2] (41.0), and our benchmark's Understanding tasks, closely matching SOTA closed-source models, its performance on Generation tasks is significantly inferior to most models with weaker understanding capabilities (e.g., VILA-U[3], EMU3[4], Show-o[5], etc.). Conversely, MIO-Instruct[6], despite showing strong overall performance in both Understanding and Generation tasks, lags noticeably behind other U-MLLMs with relatively weaker individual understanding and generation abilities (e.g., Anole[7], Seed-LLaMA[8]) when completing the unified tasks that require understanding-generation synergy. While Seed-LLaMA, Anole, and MiniGPT-5[9] have architectures that better support interleaved multimodal generation, their basic understanding and generation capabilities are relatively weak. Based on our MME-U benchmark analysis, under the current training paradigm, the understanding and generation capabilities of U-MLLMs have not yet formed a synergistic enhancement in unified scenarios, exhibiting instead a trade-off between the two capabilities.
> > >
> > > 2.  **Furthermore, we analyzed whether U-MLLMs truly generate content based on comprehension of the input:** In tasks such as CSQ, SpotDiff, and Auxiliary Lines, we systematically observed a large number of cases where the "text option was correctly selected, but the generated image severely mismatched the answer" (Figures 16, 17, 18, 19). In the Visual CoT task, even though we decomposed each step into multiple-choice questions regarding action, coordinates, and the intermediate state image, all models still failed to successfully complete multi-step maze navigation. Furthermore, as the number of steps increases, the accuracy across all three dimensions shows a significant decline and an error cascade phenomenon, reflecting the difficulty current U-MLLMs face in generating content consistent with their comprehension in unified scenarios. We posit that the true capability bottleneck for current U-MLLMs in complex generation-understanding unified scenarios is not merely boosting separate understanding/generation abilities as shown by isolated benchmarks, but rather their ability to maintain cross-modal alignment throughout a continuous reasoning-generation process, thereby synergizing generation and understanding capabilities for mutual promotion, instead of simple interleaved generation.
> > >
> > > ---
> > > [1] Liu, et al. "MMBench: Is Your Multi-modal Model an All-around Player?" arXiv preprint arXiv:2307.06281 (2024).
> > >
> > > [2] Yue, et al. "MMMU: A Massive Multi-discipline Multimodal Understanding and Reasoning Benchmark for Expert AGI." arXiv preprint arXiv:2311.16502 (2024).
> > >
> > > [3] Wu, et al. "VILA-U: A Unified Foundation Model Integrating Visual Understanding and Generation." arXiv preprint arXiv:2409.04429 (2025).
> > >
> > > [4] Wang, et al. "Emu3: Next-Token Prediction Is All You Need." arXiv preprint arXiv:2409.18869 (2024).
> > >
> > > [5] Xie, et al. "Show-o: One Single Transformer to Unify Multimodal Understanding and Generation." arXiv preprint arXiv:2408.12528 (2025).
> > >
> > > [6] Wang, et al. "MIO: A Foundation Model on Multimodal Tokens." arXiv preprint arXiv:2409.17692 (2024).
> > >
> > > [7] Chern, et al. "ANOLE: An Open, Autoregressive, Native Large Multimodal Model for Interleaved Image-Text Generation." arXiv preprint arXiv:2407.06135 (2024).
> > >
> > > [8] Ge, et al. "Making LLaMA SEE and Draw with SEED Tokenizer." arXiv preprint arXiv:2310.01218 (2023).
> > >
> > > [9] Zheng, et al. "MiniGPT-5: Interleaved Vision-and-Language Generation via Generative Vokens." arXiv preprint arXiv:2310.02239 (2024).

---

> ### Author Response · Authors · 2025-11-21
> **Response to W4-(1/2)**
>
> **W4: Concerns about data quality and potential artifacts.**
>
> A4: Thank you for your valuable feedback. Here is our response regarding the quality inspection of synthetic data:
>
> **Only about 7.32% of the data requires LLM synthesis for negative sample construction; the vast majority of the data originates from human annotation or existing real data:** In constructing the Understanding and Generation tasks, we primarily relied on data from existing public benchmarks, reconstructed through human and rule-based methods, rather than having the LLM "invent" samples. We only reconstructed the existing data into a multiple-choice format and adopted a consistent, formalized matching strategy to judge model outputs; Generation task data was standardized with uniform input/output structures and evaluation metrics.
>
> * **Unify Tasks Built on Annotations:** SpotDiff and Auxiliary Lines (AL) are both built upon existing annotations. The AL task, for instance, uses GPT-4o only to rewrite structured information into natural language question-answer templates, without generating negative samples. The Visual CoT task uses the Gymnasium reinforcement learning environment API[1] to generate state images, with correct actions and coordinates manually derived and verified by us. The entire process does not rely on the LLM to generate states, actions, or images.
> * **Tasks Relying on LLM Synthesis:** Only two sub-tasks, **CSQ** and **IEE**, truly require LLM involvement for negative sample construction (generating distractors/erroneous visuals). Overall statistics show that samples involving this type of "synthetic negative sample pair" account for only about 7.32 of the total samples, making it far from the main component of the entire benchmark.
>
>
> **Our rigorous quality control methods for synthetic samples ensure the semantic accuracy and usability of this data.** For the CSQ and IEE tasks, which rely on synthetic negative samples, we first generated candidate QA pairs and negative samples using fixed templates, and then filtered out unqualified samples in stages:
>
> * **CSQ Filtering:** We automatically checked whether each question had only one option corresponding to the answer. Approximately 7%of the candidates were excluded due to "multiple correct answers." Finally, human experts reviewed and deleted the remaining approximately 3% of samples containing factual errors or vague expressions.
> * **IEE Filtering:** We used the source image, target image, and editing instruction to manually screen the generated text explanations and editing results for instruction-image consistency, filtering out approximately 5% of candidates where the semantics and visuals mismatched or the style severely deviated.
>
> For tasks only using GPT-4o for natural language templates (e.g., SpotDiff and AL), we also used unified templates and rules for verification, automatically deleting QA pairs that could lead to multiple answers or inconsistency with the original annotation. Specific filtering rules and examples of deleted cases will be detailed in the appendix of the revised manuscript.
>
>
> **Further experimental analysis verifies the effectiveness of synthetic data in terms of unbiasedness and challenge.**
>
> * **Substantial Challenge:** We observe from the experimental results in Table 3 of the manuscript that our unified task data presents a substantial challenge to current models: even in the relatively basic CSQ task, the best open-source model only achieves approximately 38% in the $\text{acc}^{+}$ metric ("text and image simultaneously correct"). Unified accuracy in tasks like SpotDiff, AL, and Visual CoT is generally below 30\%.  Moreover, no model can successfully complete complex maze navigation. This indicates that the synthesized QA pairs and negative samples were not "too simple" or "highly similar to positive cases," but are effective in differentiating the unified capabilities of various U-MLLMs.
> * **Robustness and Unbiasedness:** We provided split-half stability experiments based on the full benchmark in Table 4 of Appendix C, showing that the relative ranking of models remains largely consistent when using only half the samples. We further supplement the split-half experimental results for each sub-task in the unified tasks (as shown in the table under responses to W4-(2/2)), The ranking of the four representative models remains stable basically across different half-sample divisions for the unify tasks.
>
> These results validate our control over synthetic data construction and filtering, helping to alleviate the reviewer's concern about whether "synthetic data is accurate, unbiased, and sufficiently challenging."
>
> ---
> [1] https://gymnasium.farama.org/environments/toy_text/frozen_lake

---

> > ### Author Response · Authors · 2025-11-21
> > **Response to W4-(2/2)**
> >
> > | Split | Method | IEE |  |  | CSQ |  |  | AL |  |  | SD |  |  | VCoT |  |  |  | Avg |
> > |-------|---------|----------|-----------|---------|----------|-----------|---------|----------|-----------|---------|----------|-----------|---------|---------------|----------------|------------|---------|------|
> > |       | Metric | Text Acc | Image Acc | Acc | Text Acc | Image Acc | Acc | Text Acc | Image Acc | Acc | Text Acc | Image Acc | Acc | Action Acc | Coordinate Acc | Image Acc | Acc | Avg |
> > | **Split A** | MiniGPT5     | 19.00 | 24.00 | 21.50 | 23.76 | 35.64 | 29.60 | 7.69 | 23.08 | 15.39 | 4.00 | 8.00 | 6.00 | 3.17 | 1.27 | 3.81 | 2.75 | 14.65 |
> > |            | Anole         | 15.00 | 17.00 | 16.00 | 70.27 | 51.35 | 60.81 | 19.23 | 11.54 | 15.39 | 20.00 | 12.00 | 16.00 | 4.44 | 0 | 6.35 | 3.60 | 20.24 |
> > |            | SEED-LLAMA    | 17.00 | 22.00 | 19.50 | 54.05 | 43.24 | 48.65 | 11.54 | 11.54 | 11.54 | 26.00 | 24.00 | 25.00 | 3.18 | 2.54 | 3.81 | 3.18 | 21.57 |
> > |            | MIO-Instruct  | 22.00 | 21.00 | 21.50 | 73.27 | 0 | 36.64 | 23.08 | 0 | 11.54 | 24.00 | 0 | 12.00 | 0 | 0 | 0 | 0 | 16.34 |
> > | **Split B** | MiniGPT5     | 23.00 | 24.00 | 23.50 | 35.64 | 41.58 | 38.61 | 3.85 | 23.08 | 13.47 | 4.00 | 4.00 | 4.00 | 1.27 | 1.27 | 1.91 | 1.48 | 15.01 |
> > |            | Anole         | 19.00 | 23.00 | 21.50 | 80.95 | 47.62 | 64.29 | 11.54 | 15.38 | 13.47 | 14.00 | 14.00 | 14.00 | 2.54 | 1.27 | 8.89 | 4.23 | 23.40 |
> > |            | SEED-LLAMA    | 21.00 | 28.00 | 24.50 | 58.73 | 50.79 | 54.76 | 15.38 | 11.54 | 13.46 | 20.00 | 18.00 | 19.00 | 5.08 | 3.18 | 3.81 | 4.02 | 23.15 |
> > |            | MIO-Instruct  | 26.00 | 27.00 | 26.50 | 79.21 | 0 | 39.61 | 11.54 | 0 | 5.77 | 22.00 | 0 | 11.00 | 0 | 0 | 0 | 0 | 16.78 |
> > | **Overall** | MiniGPT5     | 21.50 | 24.00 | 22.80 | 29.70 | 38.56 | 34.13 | 5.66 | 23.08 | 14.37 | 4.00 | 6.00 | 5.00 | 2.22 | 1.27 | 2.92 | 2.13 | 15.67 |
> > |            | Anole         | 17.00 | 20.00 | 18.50 | 70.30 | 49.00 | 59.65 | 15.38 | 13.46 | 14.42 | 17.00 | 13.00 | 15.00 | 3.49 | 0.64 | 7.62 | 3.89 | 22.30 |
> > |            | SEED-LLAMA    | 19.00 | 25.00 | 22.00 | 56.44 | 46.53 | 51.49 | 13.46 | 11.54 | 12.50 | 23.00 | 21.00 | 22.00 | 4.13 | 2.85 | 3.81 | 3.61 | 22.32 |
> > |            | MIO-Instruct  | 24.00 | 24.00 | 24.00 | 77.24 | 0 | 38.50 | 17.31 | 0 | 8.66 | 23.00 | 0 | 11.50 | 0 | 0 | 0 | 0 | 16.56 |

---

> ### Author Response · Authors · 2025-11-21
> **Response to Q1-(1/2)**
>
> **Q1: Missing several key sota models.**
>
> A1: We thank the reviewer for the reminder. The initial lack of coverage for some newly released models in the first draft was primarily due to time constraints. In the revised manuscript, we have addressed this by supplementing the following experiments, and the results have been incorporated into the relevant tables (as shown in Table below):
>
> * (i) For the **Understanding tasks**, we newly evaluated **Gemini-2.5-Pro[1]**;
> * (ii) For the **Generation tasks**, we added **Qwen-Image[2]** and **Qwen-Image-Edit[3]**;
> * (iii) For the complete **MME-Unify** benchmark, we systematically evaluated five of the latest U-MLLMs: **Bagel[4], RecA[5], Show-o-2[6], GPT-4o-Image[7], and Gemini-2.5-Flash-Image[8]**.
>
> Overall, the closed-source model Gemini-2.5-Flash-Image leads in the total MME-U score, while among open-source models, RecA and Bagel show the best comprehensive performance. Notably, RecA's Unify-Score has significantly improved compared to previous open-source U-MLLMs, now approaching that of Gemini-2.0-Flash-Image and surpassing the closed-source model GPT-4o-Image. It is worth noting that, **in contrast to the typical trade-off between understanding and generation capabilities often seen in earlier U-MLLMs, models like Bagel and RecA have achieved significant progress in the Unify tasks while simultaneously improving their Understanding and Generation scores** (as shown in table under response to Q1-(2/2)). This accurately reflects the "understanding-generation synergy" emphasized by MME-Unify.
>
> We will supplement Tables 2, 3, and 6 in the main body with the latest results, and include an analysis of the performance of new models such as Bagel and RecA in Section 3 and Appendix E. This aims to provide readers with a more comprehensive understanding of the unified capability of the latest U-MLLMs under MME-Unify.
>
>
>
> | Model |        Understanding         |              |             |             |            Generation             |        |        |        |        |        |        |        Unify        |        |        |        |        |        | MME-U Score |
> |-------|------------------------------|--------------|-------------|-------------|------------------------------------|--------|--------|--------|--------|--------|--------|----------------------|--------|--------|--------|--------|--------|--------------|
> |       | SIPU | MITPU | VPU | Avg     | CIVG | VP   | TVG | TIG | TIE | FIR | Avg   | IEE  | CSQ  | AL   | SD   | VCoT | Avg   | Avg          |
> | Gemini2.5-Pro       | 87.00 | 69.00 | 66.21 | 74.07 | - | - | - | - | - | - | - | - | - | - | - | - | - | 24.69 |
> | Qwen-Image          | - | - | - | - | - | - | - | 72.43 | - | - | 12.07 | - | - | - | - | - | - | 4.02 |
> | Qwen-Image-Edit        | - | - | - | - | - | - | - | - | 58.81 | 88.86 | 73.84 | - | - | - | - | - | - | 24.62 |
> | Show-o2       | 68.33 | 47.00 | 50.00 | 55.11 | - | - | - | 50.18 | - | - | 8.36 | - | 66.34 | - | - | - | 13.27 | 23.94 |
> | Bagel          | 76.67 | 53.00 | 51.10 | 60.26 | - | - | - | 44.51 | 45.46 | 59.91 | 24.98 | 33.34 | 77.23 | 31.73 | 25.50 | 11.20 | 35.80 | 40.35 |
> | GPT-4o-Image      | 65.50 | 49.50 | 45.05 | 53.35 | - | - | - | 60.07 | 46.58 | 65.65 | 28.72 | 43.00 | 86.64 | 42.31 | 22.50 | 11.02 | 41.10 | 41.06 |
> | RecA        | 76.00 | 57.00 | 56.04 | 63.01 | - | - | - | 46.30 | 46.87 | 72.88 | 27.36 | 35.67 | 79.21 | 33.66 | 26.50 | 12.22 | 37.45 | 42.60 |
> | Gemini-2.5-Flash-Image | 80.25 | 70.75 | 58.79 | 69.93 | - | - | - | 66.29 | 52.90 | 85.32 | 34.09 | 48.75 | 86.02 | 59.62 | 25.50 | 15.23 | 47.02 | 50.04 |
>
>
> ---
> [1] https://deepmind.google/models/gemini/
>
> [2] Wu, et al. "Qwen-Image Technical Report." arXiv preprint arXiv:2508.02324 (2025).
>
> [3] https://huggingface.co/Qwen/Qwen-Image-Edit
>
> [4] Deng, et al. "Emerging Properties in Unified Multimodal Pretraining." arXiv preprint arXiv:2505.14683 (2025).
>
> [5] Xie, et al. "Reconstruction Alignment Improves Unified Multimodal Models." arXiv preprint arXiv:2509.07295 (2025).
>
> [6] Xie, et al. "Show-o2: Improved Native Unified Multimodal Models." arXiv preprint arXiv:2506.15564 (2025).
>
> [7] https://openai.com/zh-Hans-CN/index/introducing-4o-image-generation/
>
> [8] https://deepmind.google/models/gemini/flash/

---

> > ### Author Response · Authors · 2025-11-21
> > **Response to Q1-(2/2)**
> >
> > | Method        | IEE |  |  |  | CSQ |  |  |  | AL |  |  |  | SD |  |  |  | VCoT |  |  |  |  | Avg |
> > |---------------|----------|-----------|---------|---------|----------|-----------|---------|---------|----------|-----------|---------|---------|----------|-----------|---------|---------|------------------|---------------|--------------|-------------|---------|------|
> > | Metric        | Text Acc | Image Acc | Acc | Acc+ | Text Acc | Image Acc | Acc | Acc+ | Text Acc | Image Acc | Acc | Acc+ | Text Acc | Image Acc | Acc | Acc+ | Action Acc | Coord Acc | Image Acc | Acc | Acc+ | Avg |
> > | Show-o2         | -        | -         | -   | -    | 79.21        | 53.47         | 66.34   | 56.44    | -        | -         | -   | -    | -        | -         | -   | -    | -           | -          | -          | -   | -    | 13.27   |
> > | Bagel    | 22.78       | 43.89         | 33.34   | 11.67    | 90.10        | 64.36         | 77.23   | 63.37    | 42.31        | 21.15         | 31.73   | 9.62    | 27.00        | 24.00         | 25.50   | 7.00    | 10.83           | 12.08          | 10.69          | 11.20   | 0    | 35.80   |
> > | RecA  | 23.39        | 47.95         | 35.67   | 10.53    | 90.10        | 68.32         | 79.21   | 59.41    | 44.23        | 23.08         | 33.66   | 13.46    | 26.00        | 27.00         | 26.50   | 9.00    | 11.53           | 14.03          | 11.11          | 12.22   | 0    | 37.45   |
> > | GPT-4o-Image    | 25.50        | 60.50         | 43.00   | 17.00    | 97.03        | 76.24         | 86.64   | 74.26    | 42.31        | 42.31         | 42.31   | 15.38    | 29.00        | 21.00         | 25.00   | 9.00    | 11.53           | 8.47          | 13.06          | 11.02   | 0    | 41.10   |
> > | Gemini-2.5-Flash-Image  | 32.00        | 65.50         | 48.75   | 19.00    | 98.92        | 73.12         | 86.02   | 73.12    | 67.31        | 51.92         | 59.62   | 32.69    | 27.00        | 24.00         | 25.50   | 9.00    | 20.97           | 11.25          | 13.47          | 15.23   | 0    | 47.02   |

---

> > > ### Comment · Reviewer_xuiz · 2025-11-23
> > >
> > > Thanks for the authors' detailed responses. It addressed most of my concerns. I will maintain my positive score.

---

> > > > ### Author Response · Authors · 2025-11-24
> > > >
> > > > We are glad that we addressed most of your concerns. Thank you again for your thoughtful and constructive feedback!

---

### Author Response · Authors · 2025-11-21
**Summarization of the Responses**

Thank you for reviewing our submitted manuscript, "MME-Unify: A Comprehensive Benchmark for Unified Multimodal Understanding and Generation Models." We sincerely appreciate the reviewer's insightful and constructive comments and suggestions. We believe that by addressing every issue raised by the reviewers, the quality and clarity of MME-Unify have been significantly enhanced. The main feedback and our corresponding actions are summarized below:

* (1) We supplemented the justification for the rationality of the CLIP-I evaluation strategy and added consistency experiments comparing it with human assessment and LLM-Judge evaluation. (See Reviewer xuiz.W1, Reviewer uqqm.W1, Reviewer VRFA.W1&Q1&Q2, Reviewer QMnN.W1&W2)
* (2) We supplemented the detail of data filtering process to ensure the quality of the final dataset. We also added Split-Half experiments to verify the stability and challenging nature of our data. (See Reviewer xuiz.W4)
* (3) We further supplemented the difficulty stratification and diagnostic analysis of the experimental results for the VCoT task. (See Reviewer uqqm.W2, Reviewer VRFA.W3)
* (4) We systematically clarified the differences from and contributions over existing multimodal benchmarks, along with the unique insights derived compared to existing benchmarks. (See Reviewer xuiz.W2&W3)
* (5) We supplemented the evaluation results using LLM-Judge based metrics and additional evaluation dimensions. (See Reviewer uqqm.Q1, Reviewer VRFA.Q3, Reviewer QMnN.W1)
* (6) We discussed the sample size, weight settings, and evaluation robustness for the unified tasks. (See Reviewer VRFA.W2)
* (7) We supplemented the random guess baseline and human-set ceiling to help calibrate the interpretability of absolute scores. (See Reviewer VRFA.W4)
* (8) We expanded and updated the evaluation of recent SOTA models, analyzing the unified capabilities of the latest U-MLLMs. (See Reviewer xuiz.Q1, Reviewer QMnN.W3)
* (9) We improved the presentation of figures and tables to enhance the paper's readability. (See Reviewer VRFA.Q1)

---

### Author Response · Authors · 2025-12-02
**Summary of Revisions and Reviewer Consensus for Submission 4788**

Dear New Area Chair,

To assist in your assessment of our submission given the recent changes, we provide a summary of our work’s core contributions, the status of reviewer discussions, and the major improvements made during the rebuttal.

---

**Core Research Focus**

This paper proposes MME-Unify, a benchmark that measures the understanding, generation, and unified mixed-modality capabilities of Unified Multimodal Large Language Models (U-MLLMs). We contribute:

- **Unified mixed-modality benchmark.** We propose MME-Unify, the first benchmark specifically designed to evaluate the unified mixed-modality generation capabilities of U-MLLMs.

- **Standardized evaluation framework.** We design five unify tasks and combine understanding, generation and unify tasks into a multiple-choice and normalized scoring scheme, providing an intuitive MME-U score to assess U-MLLMs' unify capability.

- **Comprehensive U-MLLM Assessment.** We systematically evaluate current U-MLLMs, revealing a clear trade-off between the understanding and generation capabilities of current U-MLLMs and demonstrating that current models still struggle on complex unified tasks.

---

**Reviewer Consensus and Discussion Status**

The novelty of our MME-Unify benchmark (`xuiz`, `uqqm`, `VRFA`) and the five unify tasks for mixed-modality generation evaluation (`xuiz`, `VRFA`, `QMnN`) was widely recognized. We believe our rebuttal resolves the current score split:

- **Reviewer xuiz (Score 6, positive maintained):** Raised concerns about the CLIP-I evaluation, data filtering, benchmark contribution and missing evaluation of several U-MLLMs; our added CLIP-I vs human/LLM-Judge consistency experiments, detailed filtering process and Split-Half analysis, clearer comparison to existing benchmarks, and expanded U-MLLM evaluations addressed these, and they confirmed most concerns were resolved.

- **Reviewer VRFA (Score 4→6, Confidence 4→5 on Nov 22):** Questioned the necessity and diagnostic power of the benchmark, CLIP-I rationality, unified-task robustness, baselines, and presentation; our strengthened CLIP-I justification, LLM-Judge–based results, discussion of sample size/weights, random-guess and human baselines, and improved figures/tables led them to state their main concerns were resolved and to raise both score and confidence.

- **Reviewer uqqm (Score 6):** Asked about CLIP-I soundness, VCoT difficulty stratification/diagnosis, and the need for LLM-Judge–based metrics; these are directly addressed by our CLIP-I consistency experiments, expanded VCoT analysis, and added LLM-Judge–based evaluations.

- **Reviewer QMnN (Score 4):** Questioned CLIP-I reliability, requested complementary LLM-Judge metrics, and asked for broader U-MLLM evaluation; our CLIP-I vs human/LLM-Judge consistency study, extra LLM-Judge results, and expanded U-MLLMs evaluation experiments specifically target these issues.

Although reviewers **uqqm** and **QMnN** did not respond after the rebuttal, their concerns largely mirror those of **xuiz** and **VRFA**, who **explicitly acknowledged that our revisions resolved their issues and provided positive feedback.**

---

**Key Rebuttal Updates**

In response to reviewer feedback, we have significantly strengthened the paper with the following additions:

- **Evaluation Design and Justification:** We provide a more thorough justification of the CLIP-I–based evaluation strategy, including consistency experiments against both human assessment and LLM-Judge metrics, and we add LLM-Judge–based results and additional consistency analyses to validate robustness.
- **Data Quality and Difficulty Analysis:** We detail the full data filtering pipeline and introduce Split-Half experiments to verify stability and challenge level of our unify tasks, alongside a difficulty stratification and diagnostic analysis for the VCoT task.
- **Benchmark Contribution and Score Calibration:** We clarify in a systematic way how MME-U differs from and improves upon existing multimodal benchmarks, highlighting the unique insights it provides; we also discuss sample size, weight settings, and robustness for unified tasks, and add random-guess baselines plus human-set ceilings to calibrate the interpretability of absolute scores.
- **Expanded Model Evaluation:** We expand and update our evaluation to include more recent strong U-MLLMs, providing a more up-to-date analysis of unified capabilities in the current model landscape.
- **Improved Presentation:** We refine the figures to improve clarity and readability, making the overall paper easier to follow.

---

We have updated the manuscript (marked in $\\color[RGB]{80,145,220}{\\text{blue}}$). We hope that the revised version of the paper is robust and effectively addresses the reviewers' comments. **Given the special circumstances, we stand ready to promptly answer any further questions you may have.** We appreciate your time and attention to this submission.

Best regards,

The Authors

---

### Meta-Review · Area_Chair_S4zK · 2026-01-04

**Summary:**

This submission introduces MME-Unify, a benchmark for the evaluation of Unified Multimodal Large Language Models in terms of multimodal understanding, generation, and mixed-modality generation capabilities. Reviewers consistently recognized the contribution of its five unify subtasks and standardized evaluation framework, which fills a key void in existing literature. The authors have thoroughly addressed core concerns (including CLIP-I evaluation validity, data quality, incremental contribution, and model coverage, etc.) through supplementary experiments.

**Reviewer Concerns:**

- **Addressed Concerns**: All major issues have been resolved. The authors validated the reliability of CLIP-I through correlations with human and LLM-Judge assessments. They expanded the model coverage by incorporating recent important U-MLLM models and enhanced diagnostic capabilities by stratifying the maze size of the VCoT task. The supplemented insights from model ranking conclusions from the MME-Unify benchmark improved the benchmark's value.

- **Outstanding Concerns**: The agreement between CLIP-I and human assessments is still not particularly high (around 0.7), leaving the benchmark's credibility at risk. However, due to the limitations of current technologies, cross-validation using CLIP-I, LLM-Judge, and humans remains a viable solution.

**Reviewer Scores:**

- Reviewer xuiz: Would maintain a positive score of 6 after rebuttal, confirming core concerns resolved.
- Reviewer uqqm: Would maintain a positive score of 6 after rebuttal, with concerns about CLIP-I validity and VCoT discriminative power addressed via supplementary analyses.
- Reviewer VRFA: Would increase the score from 4 to 5/6, with main concerns about the necessity and the diagnostic capability of this benchmark well resolved.
- Reviewer QMnN: Would increase the score from 4 to 5/6. Core concerns about evaluation metrics and model coverage were resolved.

---

### Decision · Program_Chairs · 2026-01-26

Accept (Poster)